# Climate's firm grip on glacier ablation in the Cordillera Darwin Icefield, Tierra del Fuego

**Franziska Temme** [1] ✉, **Christian Sommer** [1], **Marius Schaefer** [2], **Ricardo Jaña** [3], **Jorge Arigony-Neto** [4], **Inti Gonzalez** [5,6], **Eñaut Izagirre** [7,8], **Ricardo Giesecke** [9,10], **Dieter Tetzner** [11] & **Johannes J. Fürst** [1]

The Cordillera Darwin Icefield (CDI) in Tierra del Fuego is one of the largest temperate ice bodies in the Southern Hemisphere. We simulate the climatic energy and mass balance of its glaciers (2000–2023), which are sensitive indicators of climatic changes in the Southern Hemisphere's higher mid-latitudes. Year-round westerly winds cause strong climatic gradients across the mountain range, reflected in the energy and mass fluxes. Our results reveal a significant increase in surface melt (+0.18 m w.e. yr⁻¹ per decade) over the past two decades. We also present the first estimate of dynamically controlled mass loss into adjacent fjords and lakes by frontal ablation, amounting to $1.44 \pm 0.94$ Gt yr⁻¹ (26 % of the total CDI mass loss). Frontal losses are mainly channelized through few marine-terminating glaciers. While frontal ablation is important for predicting the fate of individual glaciers, for the CDI as a whole, atmospheric conditions exert the main control on the current glacier evolution.

The Cordillera Darwin Icefield (CDI) is one of the largest icefields in the Southern Hemisphere[1], holding a substantial mass of ice that is at least twice as large as the mass of all glaciers in the European Alps[2,3]. The main continuous icefield covers the Cordillera Darwin mountain range and is extended by few smaller adjacent ice bodies separated by fjords (Fig. 1), such as the Mount Sarmiento Massif in the west, summing up to a total glaciated area of 2356 km² in 2022[4]. Glaciers in the CDI descend from up to 2500 m a.s.l. down to sea level. This large altitudinal range is possible due to the extreme climatic conditions inducing high mass input at the highest elevations. Tierra del Fuego, located at the southernmost end of South America (Fig. 1a), is the closest continental land mass to Antarctica. Being situated between the subtropical anticyclone and the subpolar low-pressure trough, the area is exposed to strong, year-round westerly winds. Within this so-called storm track, frontal systems continuously transport moist maritime air masses towards the continent[5]. Orographic

uplift of air masses causes abundant precipitation along the western slopes of the Cordillera Darwin while lee-side effects result in more arid conditions in the east[6]. The strength and position of the Southern Hemisphere westerlies impacts not only the formation of clouds and precipitation but also the global ocean circulation[7]. In the past few decades, the storm track has shifted poleward due to an intensification of the subtropical high in the southeast Pacific which is partly ascribed to human-induced climate change[8]. The southward shifting together with an intensification of the westerlies is projected to continue, at least, until the end of the 21st century under high emission scenarios[7]. Southern Patagonia is the only continental land mass disrupting the Southern Hemisphere westerly wind belt. Since glaciers are susceptible indicators of climate change, the glacier evolution of the CDI provides valuable insights into climatic changes in this region.

In the last decades, the CDI experienced strong ice loss[1,9,10], contributing about 5% of the total loss in South America between 2000

¹Institut für Geographie, Friedrich-Alexander-Universität Erlangen-Nürnberg, Erlangen, Germany. ²Instituto de Ciencias Físicas y Matemáticas, Universidad Austral de Chile, Valdivia, Chile. ³Departamento Científico, Instituto Antártico Chileno, Punta Arenas, Chile. ⁴Instituto de Oceanografia, Universidade Federal do Rio Grande, Rio Grande, Brazil. ⁵Centro de Estudios del Cuaternario de Fuego-Patagonia y Antártica, Punta Arenas, Chile. ⁶Programa Doctorado Ciencias Antárticas y Subantárticas, Universidad de Magallanes, Punta Arenas, Chile. ⁷Hydro-Environmental Processes Research Group, University of the Basque Country UPV/EHU, Leioa, Spain. ⁸Basque Centre for Climate Change BC3, Leioa, Spain. ⁹Instituto de Ciencias Marinas y Limnológicas, Universidad Austral de Chile, Valdivia, Chile. ¹⁰Centro FONDAP de Investigación en Dinámica de Ecosistemas Marinos de Altas Latitudes (IDEAL), Valdivia, Chile. ¹¹Ice Dynamics and Palaeoclimate, British Antarctic Survey, Cambridge, UK. ✉e-mail: franziska.temme@fau.de

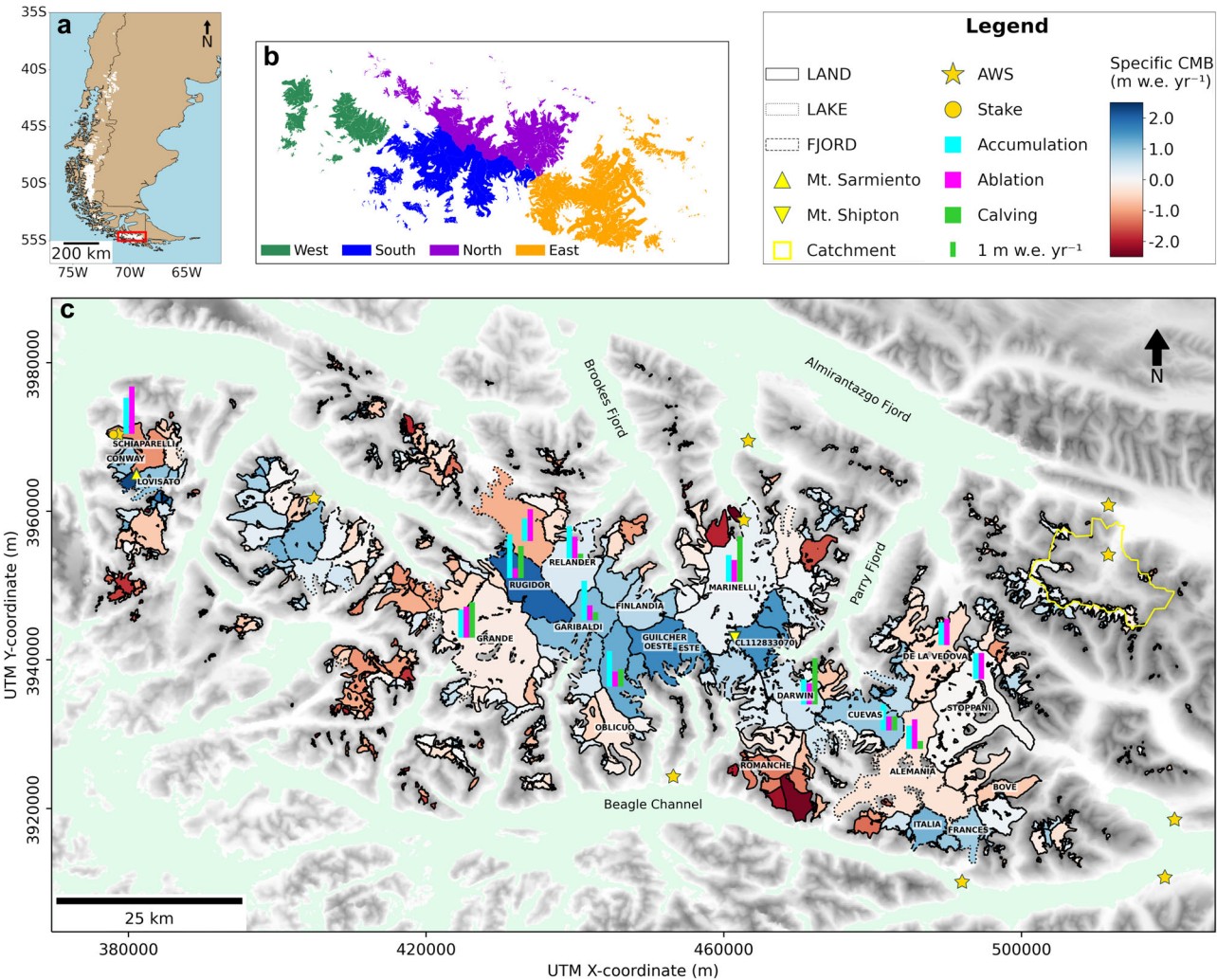

**Fig. 1 | Climatic mass balance of the Cordillera Darwin Icefield (CDI) over the period 2000–2023. a** Overview panel of the study site (made with Natural Earth). **b** Subregions of the CDI defined within this study. **c** Glacier-specific climatic mass balance (CMB) (color scheme), termination type of the glaciers (outline style, outlines mark 2000 extent), observations from automatic weather stations (AWSs) and ablation stakes in the region, and glacier-specific mass fluxes from accumulation, ablation and calving (2000–2013) for selected glaciers. Triangles mark Mount Shipton (2568 m) and Mount Sarmiento (2207 m). The river catchment of Río Betbeder (eastern edge) is shown in yellow. The digital elevation map in the background is taken from Jarvis et al.[83].

and 2011/2014[9]. Estimates of mass loss rates for Tierra del Fuego range between 1.02 ± 0.11 Gt yr⁻¹ (2000–2011/14)[9] and 1.9 ± 1.1 Gt yr⁻¹ (2000–2018)[10]. Despite the general retreat pattern, individual glaciers are stable or even advancing, mostly in the central region of the CDI[11]. Such advance continues to this day as confirmed in glacier inventories covering the last two decades (Fig. S1). The largest advance of around 2 km was observed for Garibaldi Glacier (Fig. S1). The advancing behavior is all the more remarkable when compared to the extreme retreat of Marinelli Glacier[12,13], the largest glacier of the CDI, which is located in close vicinity at the northern slope of Mount Shipton (Fig. 1c). From 1945–2005, Marinelli Glacier experienced an extreme recession of 12.2 km[14], explained by warming and fast retreat along over-deepened fjord bathymetry[12].

Around half of the CDI area consists of marine- or lake-terminating (MALT) glaciers (35 % and 13 %, respectively). Those glaciers do not only lose mass on their surface in contact with the atmosphere, but also at the ocean/lake interface via iceberg calving, which is controlled by ice dynamics and fracturing, and subaqueous melting, collectively known as frontal ablation[15]. The ice losses and the contrasting behavior observed for individual CDI glaciers can possibly be explained by both climatic and ice-dynamic changes. However, the attribution of ice loss to climatic or ice-dynamic forcing is still unknown. Most mass balance

estimates have been obtained applying a geodetic method[1,9,10] which comprises both of these loss terms ultimately quantifying the total mass change. An attribution is possible for individual glaciers if one of the two ice-loss terms is known. In the mass budgeting approach, one of these two terms is determined as a residual by subtracting either the climatic or frontal ablation loss from the total mass change[16]. However, due to the harsh climatic conditions and the inaccessibility of this region, the CDI remains poorly studied[13,14,17,18] and neither of these two loss terms can be quantified reliably. The climatic mass balance (CMB) has been studied in the Cordon Martial (east of the study region, located in Argentina)[19] and in the Mount Sarmiento Massif[20,21] (western CDI), but these local efforts are insufficient and a systematic CDI-wide estimate of the CMB is needed. With this dataset, attribution of the CDI would be possible and, additionally, vital information on current trends and shifts in atmospheric conditions of the Southern Hemisphere's higher mid-latitudes can be produced. Geodetic techniques in satellite remote sensing allow for operational inference of total glacier mass changes worldwide[9,10,22]. Thus, with the climatic mass balance term being quantified, we assume that the residual with respect to the geodetic mass budget is primarily explained by frontal ablation – an ice-dynamically controlled loss term. An alternative approach is the direct quantification of frontal losses with a flux gate approach[15,23].

Here, the frontal ablation is estimated based on the ice flux through a gate upstream of the glacier front. Due to the lack of ice thickness observations in the CDI producing high uncertainties on reconstruction products (e.g., refs. [2],[3]), results are, however, inaccurate. First estimates of frontal ablation in the CDI are limited to two glaciers: At Schiaparelli Glacier frontal ablation calculated with a mass budgeting approach[20] agrees well with inferred values from time-lapse camera observations[24]. At Marinelli Glacier, Koppes et al.[18] find a calving flux of around 0.4 Gt yr⁻¹ (2000–2005) applying an ice budget model.

The two primary objectives of this study are to (1) quantify the unknown CMB of the CDI and (2) attribute CDI ice loss to climatic or frontal ablation. We simulate the CMB over a 23 year period, which provides insights in climatic and glaciological trends, and increases our process understanding of glacier response to climate. The energy and mass balance is examined using the physically based COupled Snowpack and Ice surface energy and mass balance model in PYthon (COSIPY)[25], which combines a surface energy and mass balance model with a subsurface multi-layer snow and ice model (Methods section). The fully distributed model allows an analysis of the spatial and temporal variability in the surface energy and mass fluxes across the CDI. Together with geodetic observations, we use the CMB to close the mass budget of all glaciers in the CDI and provide an estimate of their frontal losses. The unprecedented attribution of the observed mass loss to climatic and ice-dynamic forcing plus the fully distributed CMB model allow conclusions to be drawn on the contrasting glacier behavior in the last two decades.

## Results

### Climatological characteristics of the CDI

High-resolution (200 m spatial and 3 hourly temporal) atmospheric forcing was created by observation-informed downscaling of ERA5 reanalysis data (Methods section). Our multi-method downscaling relies on quantile mapping[26] as well as modeling of solar radiation[27] and orographic precipitation[28–30] for the period of 04/1999-03/2023. Variables comprise near-surface air temperature, relative humidity, air pressure, wind velocities, cloud cover, incoming solar radiation and precipitation (Methods section). Evaluation of atmospheric variables shows overall good agreement with observations from automatic weather stations (Table S1, Table S2). A comparison between a firn core in the central Cordillera Darwin and modeled precipitation at the closest grid point shows an overestimation (mean bias of +0.63 m w.e. yr⁻¹) while catchment-wide precipitation about 50 km downwind measured with a stream gauge indicates an underestimation (mean bias of -0.43 m w.e. yr⁻¹) (Fig. 1, Fig. S2). The location of the former firn core is in an exposed saddle position where the local wind field and snowdrift are not resolved at the process level, while the latter catchment comparison suffers from neglecting water storage. As our aim is a first climatic mass balance estimate on regional scale, consistent with regional geodetic measurements, local deviations are acceptable. The CDI is divided into four subregions (Fig. 1b) to analyze spatial variability of climatic characteristics and energy and mass fluxes across the study region. Significance of trends over the study period are formulated following the IPCC guidance for communication of confidence (Table S3)[31]. These different significance levels are marked in italic font.

The study region is characterized by temperate maritime climate. Downscaled annual mean air temperatures close to sea level (2 m above ground) lie around 5.2 °C with moderate interannual variability (±3 to 4 °C). Air temperatures show a positive trend (*virtually certain*) over the study period of +0.41 °C per decade with an intensification from west to east. Conditions are overall humid. While air temperatures at sea level exhibit no clear west-east gradient, annual average relative humidity shows a drying towards the east (Fig. S3b). Relative humidity is on average higher in winter than summer, with the amplitude increasing from around 7 % in the west to around 12 % in the

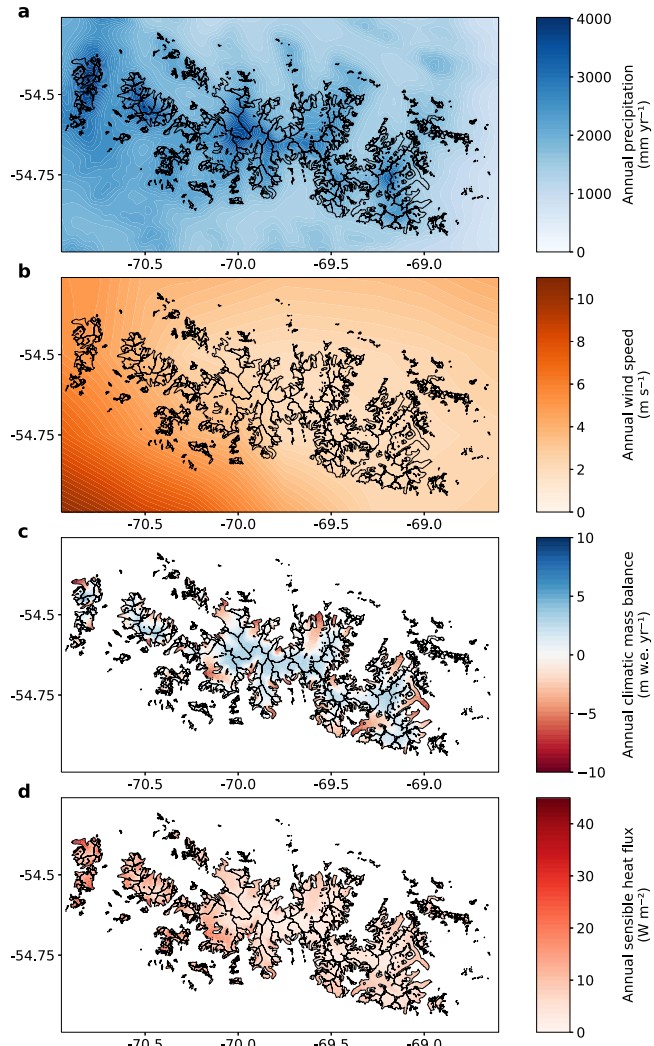

**Fig. 2 | Climatological and mass and energy balance characteristics.** Panels show mean annual (**a**) precipitation (downscaled with the orographic precipitation model), (**b**) wind speed (downscaled from ERA5 reanalysis data), (**c**) climatic mass balance (simulated with the COSIPY model) and (**d**) sensible heat flux (simulated with the COSIPY model) over the Cordillera Darwin Icefield (2000–2023). Black outlines display the glacier extent in 2000 from ref. 59.

east. Within the westerly wind belt, frontal systems move from the southern Pacific Ocean towards southern Patagonia causing high precipitation amounts due to orographic uplift. Highest amounts are reached in the westernmost edge of the Cordillera Darwin, going up to 4000 mm yr⁻¹ at Mount Sarmiento (Fig. 2a). As the air masses move eastwards over the cordillera, fallout of precipitation causes a drying effect towards the east of the CDI (Fig. 2a, Fig. S3e). Precipitation amounts peak in the summer months, related to the increased wind velocities. The seasonality is forced by a southward shift of the westerly wind belt during summer[32]. Precipitation and wind both show an increasing trend (*likely*) over the study period (+70.8 mm yr⁻¹ per decade and +0.1 m s⁻¹ per decade, respectively). Wind velocities are high throughout the year especially in the westernmost edge of the CDI where annual averages go up to 5.5 m s⁻¹, and decrease towards the central and eastern part of the CDI (Fig. 2b, Fig. S3d). High amounts of annual precipitation are accompanied by an average cloud cover of over 84 %. Such extensive cloud cover strongly limits direct solar radiation (Fig. S3c, Fig. S8a). Regional radiation differences between north and south are mainly explained by the orientation to the sun and the aspect of the slopes.

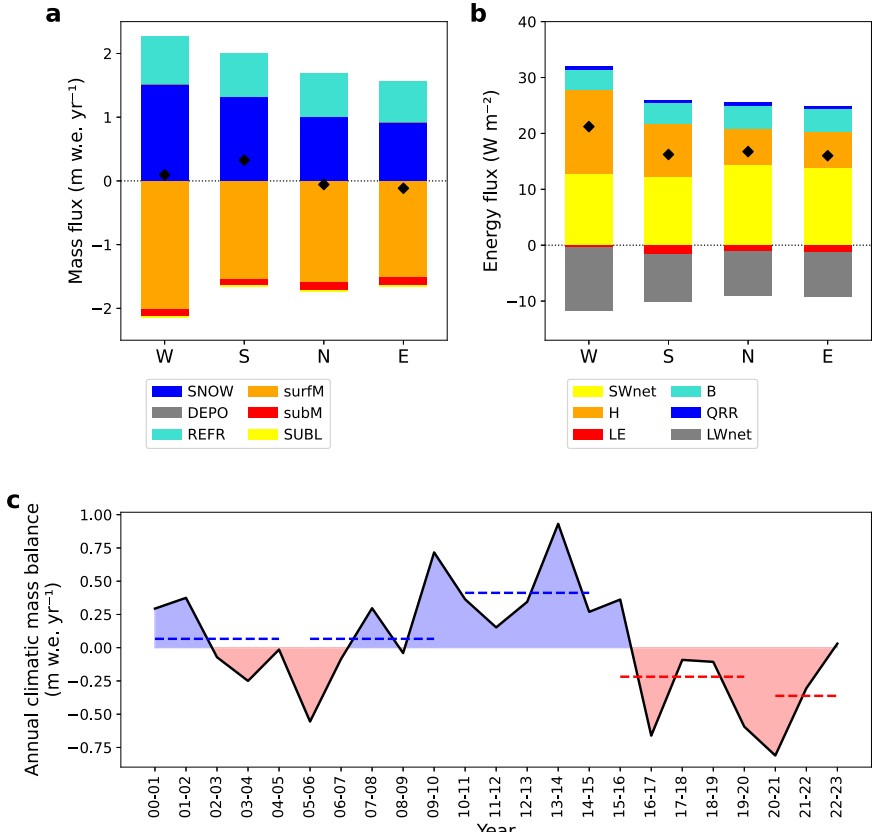

**Fig. 3 | Climatic energy and mass fluxes together with the resulting mass balance of the Cordillera Darwin Icefield (CDI).** CDI-wide (land-, lake- and marine-terminating glaciers) average annual climatic (**a**) mass and (**b**) energy balance components for the four subdomains (W west, S south, N north, E east): Snowfall (SNOW), deposition (DEPO), refreezing (REFR), surface melt (surfM), subsurface melt (subM), sublimation (SUBL), net shortwave radiation (SWnet), sensible (H), latent (LE) and glacier heat flux (B), heat flux from rain (QRR) and net longwave radiation (LWnet). The black diamonds give the resulting (**a**) climatic mass balance and (**b**) energy available for melting, respectively. The lower panel (**c**) displays the CDI-wide annual average climatic mass balance with shading indicating positive (blue) and negative (red) years. Dashed lines give 5 year averages. Source data are provided as a Source Data file.

## Climatic energy and mass balance of the CDI

The climatic energy and mass balance is simulated with COSIPY[25], a fully-distributed surface energy and mass balance model coupled to a multi-layer subsurface snow and ice model (Methods section). For model evaluation, simulation results are compared to geodetic mass balances obtained from elevation change observations (2000–2013) based on the results by Braun et al.[9] for glaciers with no frontal losses (Methods section) (Fig. S4). Glacier-wide specific mass balance is used for the comparison, in the following denoted as glacier-specific mass balance. Based on this performance, we infer a glacier-specific CMB uncertainty of ±0.62 m w.e. yr⁻¹. The simulated icefield-wide average CMB of all land-terminating glaciers (-0.23 m w.e. yr⁻¹) agrees well with the geodetic reference dataset of Braun et al.[9] (-0.27 m w.e. yr⁻¹) (Fig. S5).

The CMB of the entire CDI (including marine-, lake- and land-terminating glaciers) is nearly balanced for the study period (2000–2023) with +0.02 m w.e. yr⁻¹. Temporal variability is high with annual values ranging between -0.81 and +0.93 m w.e. yr⁻¹ (Fig. 3c). Across the CDI, glacier-specific values show a high spatial variability (Fig. 1c), spanning -5.47 to +2.43 m w.e. yr⁻¹. We relate this diversity to hypsometry and aspect of the respective glacier catchments. Positive CMBs dominate especially in the central high-elevated part of the CDI (e.g., Rugidor, Garibaldi, Guilcher Oeste and Este), whereas lower-elevated glaciers or glaciers with large outlet tongues show more pronounced mass loss (e.g., Schiaparelli, Romanche, Alemania and many of the small, unnamed glaciers at the icefield margin) (Fig. 1c). The altitudinal gradient of the CMB is steep with strong mass losses on the glacier tongues at low elevation and high mass gain towards the mountain peaks (Fig. 2c). The largest range is present in the western part of the CDI, where the annual CMB reaches nearly -10 m w.e. yr⁻¹ at the lowest point of Schiaparelli Glacier and nearly +10 m w.e. yr⁻¹ at the highest peak (Fig. 2c), resulting in a gradient of -0.92 m w.e. yr⁻¹ per 100 m.

Mass is mainly gained from snowfall (63 %) at the surface and from refreezing within the snowpack (36 %) (Fig. 3a). Deposition of water vapor at the glacier surface contributes around 1 % to the accumulation. Mass loss is dominated by melt at the surface (92 %) and subsurface (7 %), with sublimation contributing only around 1 % to the total ablation. The average annual CMB is positive in the southern and western part of the CDI ( + 0.33 and +0.09 m w.e. yr⁻¹, respectively), while it is just below zero in the north and east (-0.06 and -0.12 m w.e. yr⁻¹, respectively) (Fig. 3a). These differences mainly stem from the contributions of snowfall and surface melt. Snowfall amounts strongly reduce from +1.50 m w.e. yr⁻¹ in the west to +0.92 m w.e. yr⁻¹ in the east of the CDI (Fig. 3a). In the western part, the high accumulation is partly balanced by enhanced surface melt (-2.00 m w.e. yr⁻¹). Over the remaining CDI, we find very similar surface melt rates of around -1.54 m w.e. yr⁻¹. The decreasing snowfall towards the east also results in CMB decreases (Fig. 3a). Regional differences are also reflected in the equilibrium line altitudes (ELAs) - the elevation at which surface mass gain equals surface mass loss over 1 year - in the CDI. Simulated average ELA values increase from the west (759 m a.s.l.) over the center (~810 m a.s.l.) to the east (893 m a.s.l.). The CMB shows a pronounced seasonality (Fig. S6). In austral summer

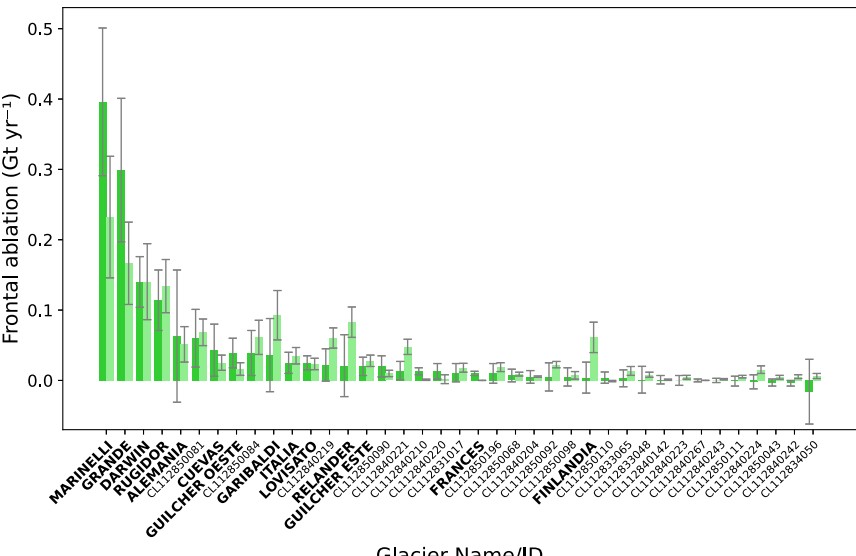

**Fig. 4 | Frontal ablation.** Mean annual frontal ablation for marine- and lake-terminating glaciers, calculated with a mass budgeting approach (2000–2013) (dark green) and a flux gate approach (2013) (light green). The respective uncertainty as defined in the Methods section is given in gray whiskers. Source data are provided as a Source Data file.

(December, January, February), it is strongly negative with the highest losses in the western (-0.77 m w.e. yr⁻¹) and the smallest losses in the southern (-0.51 m w.e. yr⁻¹) part of the CDI (Fig. S6a). This spatial difference is primarily explained by spatial variability in the surface melt. In austral winter (June, July, August), the CMB is positive all over the CDI with reduced melting (Fig. S6c). Regional differences are mainly linked to the snowfall gradient.

To understand the variability in surface melt, the energy fluxes at the surface and subsurface are analyzed. On average, the largest energy input comes from the net shortwave radiation (51 %) and the sensible heat flux (32 %) (Fig. 3b). The glacier heat flux (energy generated from penetrating shortwave radiation and refreezing that is transported to the surface via heat conduction) brings on average 15% of energy, and is mostly constrained to the higher reaches of the glaciers (Fig. S9b). The heat flux from rain is limited to the glacier tongues (Fig. S9c), contributing around 2 % in total. Energy loss at the surface is dominated by the net longwave radiation (88 %), followed by the latent heat flux (12 %) (Fig. 3b), which is an energy source on the glacier tongues but a sink on the spatial average (Fig. S9a). The largest amount of energy available for melting is found in the western part of the CDI (Fig. 3b, Fig. S9d). The energy surplus mainly stems from the sensible heat flux that is more than twice as high (14.97 W m⁻²) as compared to the northern and eastern part (around 6.43 W m⁻²) of the CDI (Fig. 2d). Subsequently, the sensible heat flux is the primary energy source (47 %) in the west (Fig. 3b). The enhanced values there are related to the higher wind velocities prevailing over the south-western part of the CDI (Fig. 2b). The slightly reduced net shortwave radiation in the south-west is caused on the one hand by reduced incoming radiation due to orientation and topographic shading (Fig. S8a), on the other hand by higher average surface albedo (around 0.83 in the south-west, around 0.81 in the north-east) related to the enhanced snowfall. The latent heat flux is an energy sink for the south, north and east part of the CDI (Fig. 3b). Its importance is reduced in the west. Due to the moister conditions in the west (Fig. S3b), water vapor transport towards the glacier surface is more common generating deposition and condensation instead of sublimation and evaporation, which are favored in the other sub-regions (Fig. S7a-b, Fig. S9a). In summer, the latent heat flux temporarily turns into an energy source in the west, which is not seen

over the rest of the icefield (Fig. S6b). The energy balance shows a strong seasonality with highest melt energy in summer (Fig. S6b) and strongly reduced energy in winter (Fig. S6d). In winter, energy availability is limited due to the minimized solar energy together with reduced wind velocities and lower air temperatures.

Our results show that the inter-annual variability of the CMB (Fig. 3c) is mainly influenced by air temperature (Fig. S3a) (Pearson's correlation $r = -0.71$) and by precipitation (Fig. S3e) dictating accumulation ($r = 0.73$). The air temperature strongly influences the energy available for melting ($r = 0.85$). Melt energy and CMB are also linked to the annual average incoming solar radiation ($r = 0.54$ and $r = -0.46$, respectively) as well as the albedo ($r = -0.68$ and $r = 0.69$, respectively). The importance of wind speed over the CDI is highlighted by a significant correlation with the sensible heat flux ($r = 0.62$) as well as accumulation ($r = 0.52$).

Over the 23 year study period, we see an increasing trend (*extremely likely*) in the surface melt (+0.18 m w.e. yr⁻¹ per decade) that translates into a *likely* decreasing trend for mass balance (-0.18 m w.e. yr⁻¹ per decade), which is caused by the more negative annual CMB towards the end of the study period (Fig. 3c, Fig. S10a). The increasing surface melt is mainly caused by an *extremely likely* increasing trend in the sensible heat flux (+0.79 W m⁻² per decade) and net longwave radiation (+0.62 W m⁻² per decade) (Fig. S10b), which we primarily relate to increasing air temperature (both variables) and wind velocity (sensible heat flux). Additionally, we observe an *extremely likely* increasing net shortwave radiation (+0.78 W m⁻² per decade) (Fig. S10b), which we explain with an albedo feedback (albedo decrease of -0.01 per decade, *extremely likely*). Altogether, an *extremely likely* increasing trend in available melt energy (+1.93 W m⁻² per decade) is obtained. The trend in the CMB is more pronounced for lower elevation bins. Below 600 m elevation, the CMB shows an *extremely likely* decreasing trend (up to -0.50 m w.e. yr⁻¹ per decade). Snowfall or accumulation trends are not significant and currently unable to compensate for the increased surface melt.

## Estimation of frontal ablation in the CDI
Mass budgeting of the geodetic and climatic mass balance gives information on frontal ablation of the 39 marine- and lake-terminating (MALT) glaciers of the CDI (Methods section). This way, we are able to

assess the direct ice flux into the fjords. An independent estimate for frontal ablation relies on the flux gate approach requiring surface velocities and near-frontal ice thickness[3] (Methods section). The latter approach implies elevated uncertainties due to the largely unconstrained thickness maps of the CDI in absence of ground-truth data.

With the mass budgeting approach, the total frontal ablation of the CDI is estimated to $1.44 \pm 0.94$ Gt yr$^{-1}$ between 2000–2013. About half of the total flux is channelized through Marinelli Glacier with $0.40 \pm 0.10$ Gt yr$^{-1}$ and Grande Glacier with $0.30 \pm 0.10$ Gt yr$^{-1}$ (Fig. 4). Another 20 % of the frontal ablation is explained by two other prominent marine-terminating glaciers: Darwin with $0.14 \pm 0.04$ Gt yr$^{-1}$, and Rugidor with $0.11 \pm 0.04$ Gt yr$^{-1}$. The flux-gate approach gives an almost identical total frontal ablation with $1.49 \pm 0.53$ Gt yr$^{-1}$ for the year 2013 and shows the overall same pattern (Fig. 4). Largest contributors are confirmed, however, in this case they are responsible for only around half of the total flux. For both approaches, the majority of MALT glaciers shows only low frontal ablation (<0.03 Gt yr$^{-1}$) (Fig. 4) accounting on average for <1% of the total flux. For a few glaciers, the flux-gate approach shows clearly elevated values - though with large uncertainties. In general, calving into fjords explains the main share of the frontal ablation (around 90 %) while these glaciers drain 75 % of the surface area of all MALT glaciers. Calving into lakes therefore plays a secondary role.

The availability of geodetic datasets covering multiple observation periods (e.g., ref. 22.) allows an analysis of changes in frontal ablation rates over the study period (from 2000–2010 to 2010–2020). Bias-corrected estimates based on mass budgeting using these datasets and the CMB, reveal a total frontal ablation of $1.55 \pm 0.71$ Gt yr$^{-1}$ (equaling $1.34 \pm 0.61$ m w.e. yr$^{-1}$) for the first period, and a slightly reduced value of $1.42 \pm 0.75$ Gt yr$^{-1}$ (equaling $1.25 \pm 0.66$ m w.e. yr$^{-1}$) for the second period. Associated uncertainties do not allow any reliable statement to be made about changes over the study period. The derived values suggest, however, that the frontal ablation has changed less over the last 20 years (-0.05 m w.e. yr$^{-1}$ per decade) than the CMB.

### Disentangling climatical and dynamical control on observed glacier changes

With the CMB and the frontal ablation available, we can disentangle the glacier mass changes observed from remote sensing into climatical (CMB) and dynamical (frontal ablation) forcing. For few glaciers, the mass budgeting suggests frontal ablation as the major contributor to mass loss. The frontal ablation exceeds twice the climatic ablation for glaciers Rugidor, Marinelli, Darwin, Italia and two unnamed glaciers (CL112833070 and CL112840210) (Fig. 1c, Table S4). For these glaciers, the mass loss is primarily dictated by ice dynamics instead of climate. For about 15% of CDI glaciers, the contribution of climatical and frontal ablation is rather even (e.g., Grande, Cuevas, Lovisato, Frances) (Table S4). However, for more than half of the MALT glaciers, the climatic losses at the surface and within the snowpack largely exceed the frontal ablation, meaning that the mass loss of these glaciers is dominated by atmospheric processes at the glacier surface rather than ice dynamics.

Glaciers that have been advancing over the study period (Garibaldi, Finlandia, Guilcher Oeste and Este) are in general characterized by a strongly positive CMB and by medium to low frontal ablation (Fig. 1c). These glaciers do not only advance, but also show thickening further upstream (thus, gaining mass), suggesting a primary climatic control.

Overall, ablation in the CDI is primarily controlled by atmospheric conditions. The climatic ablation of the entire CDI on average amounts to $4.17 \pm 1.48$ Gt yr$^{-1}$ (74 % of total ablation) while the frontal ablation on average amounts to $1.44 \pm 0.94$ Gt yr$^{-1}$ (26 % of total ablation). For around one third of the MALT glaciers, the mass loss is mainly controlled by frontal ablation. For six individual glaciers covering around 15 % of the CDI area, the frontal overtakes the climatical contribution to

ablation by a factor of two. While trends in CMB impose an increasing mass loss associated with climatic changes, frontal ablation remains without significant change over the study period.

## Discussion

The climatic conditions during our study period (1999–2023) show strong zonal gradients across the Cordillera Darwin, specifically for precipitation, wind velocities and relative humidity (Fig. S3). These gradients were already reported for the 20th century climate[11]. The highest precipitation amounts are located in the north-northwestern part of the CDI (Fig. 2a), as previously reported[33]. With regard to atmospheric trends we confirm that wind velocities and precipitation are increasing over Tierra del Fuego (e.g., ref. 6). The warming trend for the beginning of the 21$^{st}$ century found in this study (+0.41 °C per decade) exceeds warming rates reported for the 20th century in Punta Arenas (+0.21 °C per decade)[34] as well as the projected warming in the Magellan region until mid of this century under a high-emission scenario (RCP8.5) (total warming of +0.5 °C until 2050)[35].

The moister, snow-rich conditions in the south-western part of the CDI have been proposed to cause less thinning and retreat, while the opposite is true for the drier north-eastern part[1]. Our results support the fact that climatic conditions force more negative mass balances in the north-eastern part of the CDI compared to the south-west (Fig. 3a), which is also reflected in the regionally different average ELAs. At Schiaparelli Glacier in the west, an average ELA of $730 \pm 50$ m a.s.l. (2000–2017) has been reported[21], which is in very good agreement with our results (average ELA of 734 m a.s.l.). ELAs reported for Grande Glacier with around $640 \pm 200$ m a.s.l[1]. to 650 m a.s.l. (for 2011)[13] based on single year end-of-summer snowline altitudes are distinctly lower than our long-term average of 790 m a.s.l. 2011 was however exceptional with rather positive CMB values (Fig. 3c). For Marinelli Glacier, we calculate an average ELA around 800 m a.s.l. for the 23 year study period, which falls in between previous estimates ranging from around 600 m a.s.l. in the year 2000[12] to 1100 m a.s.l. in 2011[13]. Melkonian et al.[1] derive an ELA of $650 \pm 200$ m a.s.l. from single year end-of-summer snowlines at Garibaldi Glacier, which is lower than the average ELA of 760 m a.s.l. we found in this study. Discrepancies between end-of-summer snowline and mean long-term ELA are explained by inter-annual CMB variability and the possibility of snow fall events during the summer months[36], which do impose large uncertainties on ELA detected from satellite imagery.

Glacier-specific CMB across the CDI is highly variable (Fig. 1c) and depends strongly on glacier hypsometry. MALT glaciers show on average a more positive CMB than land-terminating glaciers (Fig. S11) as their catchment area generally extends to higher altitudes (above 1200 m). In contrast to the overall mass loss and retreat of glaciers in the CDI, few glaciers have advanced during the first two decades of the 21st century. While these glaciers are all marine terminating, they have medium to low frontal ablation which, in addition to a thickening further upstream, suggests a primarily climatic control. We ascribe these climatological favorable conditions to the glaciers' exposition, geometry and aspect. All four glaciers have their origin at the high-elevated central plateau of the CDI (above 2000 m) and a steep topography resulting in high snowfall amounts and an above-average accumulation-area-ratio (between 0.72 and 0.88 compared to the CDI average of 0.61). Glaciers with high accumulation-area-ratio are often less sensitive to changes in ELA and, thus, warming due to the small and steep ablation area[15]. Southward exposures of Garibaldi, Guilcher Oeste and Guilcher Este glaciers further reduce surface ablation. Altogether, these characteristics ultimately lead to high snowfall amounts together with reduced surface ablation (Fig. 2c, Fig. S9d), favoring mass gain.

Snowfall and surface melt are determined as the main contributors to the CMB at Schiaparelli Glacier (western CDI) and Martial Este (east of CDI), reflected in a strong correlation with air temperature

and precipitation[21,37], which is in agreement with our results (Fig. 3a). Similar to our results for the western CDI (Fig. 3b), the energy input at Schiaparelli Glacier is dominated by incoming radiation and sensible heat flux, while the largest energy losses are attributed to the outgoing radiation and the energy consumed by melting[20,21]. The importance of the sensible heat flux as an energy source has been highlighted in previous energy balance studies at the Southern Patagonian Icefield[38–42] and the Gran Campo Nevado[43]. Similar to our results, Bravo et al. found high spatial variability in the latent heat fluxes at a west-east transect of the Southern Patagonian Icefield[41]. The pronounced altitudinal mass balance gradient observed in the CDI (Fig. 2c) is a typical characteristic of Patagonian glaciers (e.g., refs. 39,42,44), as is the high inter-annual variability[39]. Periods of more negative/positive CMB (Fig. 3c) are in agreement with findings of geodetic mass balance from Dussaillant et al.[10] reporting stronger losses in the period 2000–2006 compared to the period 2012–2016 for the Fuegian Andes, although these values suffer from poor data coverage in that region (see supplementary material in ref. 10). The more positive or negative periods are strongly linked to snowfall amounts and surface melt (Fig. S10a). A distinct relation to the Southern Annular Mode, which is proposed as the main mode of climate variability in Tierra del Fuego[32], cannot reliably be confirmed from our results. This finding is consistent with a recent study focused on the Patagonian Icefields[45]. Seasonality in the energy fluxes over the CDI (Fig. S6) is in general agreement with findings at Perito Moreno Glacier, located at the Southern Patagonian Icefield[42]: The energy input from the sensible heat flux exceeds the input from net shortwave radiation during austral winter, latent heat flux has a minimum value during austral spring, and the glacier (conductive) heat flux peaks during austral winter, when the glacier surface gets cooled by the atmosphere. Schaefer et al.[40] found latent heat flux as an energy source for two Patagonian glaciers during the ablation season, which confirms our findings for the western region of the CDI, where latent heat flux turns from energy sink to energy source during summer.

A trend analysis over the study period clearly demonstrates an *extremely likely* increase in surface melt, which is in agreement with simulations of surface melt for the Northern and Southern Patagonian Icefield estimating a positive trend of +0.30 m w.e. yr$^{-1}$ per decade (1975–2005)[36]. This rise in surface melt is the main driver of a *likely* trend towards more negative mass balance in the CDI. Congruently, Bravo et al.[36] present negative trends in the surface mass balance of the Northern and Southern Patagonian Icefield (1975–2005). However, contrary results are presented in refs. 44,46,47. Despite the *likely* positive trend in precipitation amounts in the CDI, we do not find an increase in snowfall amounts. In general, glaciers in the CDI that are covering a lower elevation range (<600 m) already experience more pronounced mass losses with an average CMB trend of -0.43 m w.e. yr$^{-1}$ per decade (*extremely likely*). At Perito Moreno Glacier some 400 km north of the CDI, Minowa et al.[42] found a *virtually certain* decreasing trend in the surface mass balance over the ablation zone of -0.90 ± 0.3 m w.e. yr$^{-1}$ per decade (1996–2020). This decreasing trend is in general agreement with our findings for areas at similar altitudes in the CDI (-0.50 m w.e. yr$^{-1}$ per decade). If the recent trends continue, the CDI will get further out of balance and glaciers will trace a path of accelerating mass loss as already indicated by the current trend in the CMB (Fig. 3c). Projections for the Northern and Southern Patagonian Icefield predict just such a pathway of enhanced surface melt causing decreasing surface mass balance until 2050[36].

Frontal ablation is to this day largely unexplored in the CDI. An exception is Marinelli Glacier. Koppes et al.[18] found 0.40 Gt yr$^{-1}$ for the beginning of the 21$^{st}$ century, which is in agreement with our estimate of 0.40 ± 0.10 Gt yr$^{-1}$ by mass budgeting. Frontal ablation at the CDI (1.44 ± 0.94 Gt yr$^{-1}$) is a similar order of magnitude as for the Northern Patagonian Icefield (2.5 ± 0.5 Gt yr$^{-1}$ in 2000–2019), which covers almost twice the area[15]. Like the Northern Patagonian Icefield, ice-

dynamic losses at the front are also channelized through only a few prominent outlet glaciers (Fig. 4). Yet, the fraction of frontal ablation to total ablation (26 %) for the entire CDI is substantial. Considering the MALT glaciers only, this fraction (48 %) is similar to the MALT glaciers of the Southern Patagonian Icefield (48 %)[15]. Apart from these few glaciers, however, ablation is primarily controlled by atmospheric conditions. Frontal ablation calculated by two different methods in this study overall agrees well (Fig. 4). Discrepancies between the two approaches are within the error ranges for most glaciers. For a few glaciers (e.g., Finlandia, Garibaldi) the flux gate method produces higher frontal ablation, which can be explained by a possible over-estimation of ice thickness and/or velocity, or underestimation of modeled CMB, as well as frontal advances that are neglected in the flux gate approach.

A major limitation of this study is the scarcity of in-situ observations, making model calibration and validation a challenge. Combining in-situ and remotely sensed observations, we constrain the CMB satisfactorily. However, due to a lack of observations, most energy fluxes are not directly verifiable. Thus, absolute values should be interpreted with care, while relative comparison and trends are considered more reliable. During model calibration and analysis, we found a strong sensitivity of modeled latent heat flux to relative humidity, a model input variable which is prone to uncertainties. Furthermore, the considerable refreezing rates predicted by the model are a surprising result for a supposedly temperate icefield in a maritime climate setting. Using a 3 hourly timestep, we assume that the sub-daily melt-refreeze cycle is resolved properly. Thus, refreezing rates are expected to be higher than in studies using a daily resolution[48]. Veldhuijsen et al.[49] found that reducing the temporal model resolution from hourly to daily causes a strong underestimation of refreezing (84 %). Due to the high amounts of rainfall and melt water, refreezing is expected to be important over parts of the CDI. Our results suggest that about 23 % of percolated water (rainfall plus melt water) refreezes within the snowpack with the largest proportion in spring and autumn. These values lie in-between numbers for mid-latitude glaciers (-10 %)[50,51] and the Antarctic (Peninsula) (-70–95 %)[52,53]. To better quantify the refreezing in this region, we recommend the acquisition of observations of the firn/melt layer structure by firn cores or high frequency radar in future studies. A minor limitation is related to the mass budgeting approach: Geodetic elevation change products cannot measure subaqueous ice losses when glaciers are retreating. However, an assessment of these losses demonstrates that they are distinctly smaller compared to iceberg calving and lie within the reported uncertainty. For Marinelli Glacier, which experienced the strongest retreat over the study period, we estimate a maximum subaqueous ice loss of roughly 0.05 Gt yr$^{-1}$, constituting about 10 % of the total frontal ablation. We apply a density conversion factor of 900 kg m$^{-3}$ for volume-to-mass conversion, following Braun et al.[9]. A lower conversion factor of 850 kg m$^{-3}$[54] does not change the CMB model performance metric significantly and transmits linearly into the frontal ablation estimates. This is covered by the uncertainty ranges.

The CDI is one of the largest temperate ice bodies in the Southern Hemisphere. We present the first icefield-wide simulation of the climatic energy and mass balance as well as frontal ablation, which represents the combined terms of subaqueous melt and calving and is, therefore, mainly controlled by ice dynamics. Our results allow an attribution of the observed mass loss to climatic or ice-dynamic forcing. Furthermore, these simulations shed light on the atmospheric imprint of glacier evolution at a unique location in the higher mid-latitudes of the Southern Hemisphere, the only major land mass disrupting the westerly wind belt. Results reveal strong climatic gradients across the CDI that cause regional differences in the energy and mass fluxes. Overall, we show that the CDI has been climatically balanced in the recent two decades, but is entering a state of accelerated mass loss due to increasing surface melt. The melt increase is associated with an

intense warming rate that exceeds current projections for the early 21st century. The current melt trend is more pronounced at lower elevations. If recent warming trends continue, glaciers in the CDI will follow a trajectory of mass loss acceleration. Frontal ablation accounts for a significant share of the total ablation (26 %), but is only important for a minority of glaciers. These few glaciers dominate the CDI-wide frontal losses of 1.44 ± 0.94 Gt yr$^{-1}$. We conclude that climatic warming has impacted the CDI glaciers and their evolution is mainly controlled by atmospheric conditions.

## Methods

### Data

Meteorological observations in the Cordillera Darwin are sparse (Fig. 1c) due to the harsh environmental conditions and the inaccessibility of the region. Details of all automatic weather stations (AWSs) used in this study are listed in Table S1. Operators are the Chilean Water Directorate (Dirección General de Aguas, short DGA) or individual researchers installing stations within the framework of different research projects in the Cordillera Darwin. Measured variables range from near-surface air temperature, relative humidity, wind and air pressure to global radiation and precipitation (Table S1). However, the station network is located close to sea level, lacking information at higher elevation. This deficit is especially problematic for precipitation, which is strongly influenced by orographic effects. At one AWS (Río Betbeder) a gauging station is installed, providing valuable information on precipitation amounts over the entire river catchment (Fig. 1c). All stations have been quality checked including a screening for outliers or drift in the data. In March 2020, the COrdillera Darwin Ice CorE Survey (CODICES) project drilled a 3.25 m firn core in a flat, northwest-southeast oriented 150 × 150 m saddle (54.6814°S, 69.6394°W, 2324 m a.s.l), one of the highest flat areas in the Cordillera Darwin. Seasonal variability in major ions and insoluble microparticles were used to estimate annual layers. The firn core record extends from March 2020 to austral spring 2016.

Measurements of surface ablation or mass balance are limited in the Cordillera Darwin. Ablation stakes have been installed between 2013 and 2020 at Schiaparelli Glacier located within the Mount Sarmiento Massif at the western edge of the CDI. Stakes are limited to the lowest part of the ablation area, delivering information about surface melt only[20]. A 21 year long record of annual, winter and summer mass balance exists at the Martial Este Glacier located east of the main body of the CDI[55]. The glacier is located outside of the direct study region but we extended the domain for model validation.

We use atmospheric variables from the ERA5 reanalysis product (the latest global product of the European Centre for Medium-Range Weather Forecasts, ECMWF) to generate the climatic forcing for the COSIPY model over the study site (54.25–55.00°S, 71.00–68.25°W). ERA5 provides high temporal (hourly) and spatial (31 × 31 km) resolution[56]. For Southern Patagonia, ERA5 and its previous versions have proven reliable in several modeling studies (e.g., refs. 20,21,30,46,57,58). Required variables for this study are near-surface air temperature, relative humidity, air pressure, wind speed, cloud cover fraction and total precipitation over the study region. For downscaling of precipitation, upstream (53.75–55.50°S, 74.50–73.25°W) information about air temperature, relative humidity, wind vectors and geopotential height between 850 and 500 hPa is needed.

Glacier outlines are taken from the DGA glacier inventory[59,60] since the Randolph Glacier Inventory (V6) has revealed poor representation of outlines for the southern Andes, especially for smaller glaciers[61]. Glacier outlines available comprise two time stamps: 2000–2003 (in the following denoted as 2000) and 2019. Since glacier catchments in the more recent inventory had been partially upgraded, catchments in the 2000 inventory had to be homogenized with the 2019 inventory for consistency. Furthermore, it was shown that neglecting the

temporal evolution of the glacial extent by relying on constant glacier outlines can bias the comparison between surface and geodetic mass balance[62,63]. Since elevation changes cover the period 2000 to 2013, we manually produced outlines for 2013. Therefore, we used late-summer images from Landsat 7 and 8 (2013) and ASTER (Advanced Spaceborne Thermal Emission and Reflection Radiometer) (2013 and 2014) and manually adjusted the outlines to the current glacier extent. Thus, inventories available in this study cover years 2000, 2013 and 2019.

Geodetic mass changes of glaciers of the Cordillera Darwin are derived from interferometric Synthetic Aperture Radar (SAR) digital elevation models (DEMs) of the Shuttle Radar Topography Mission (SRTM) of the National Aeronautics and Space Administration (NASA) and the TerraSAR-X add-on for Digital Elevation Measurement satellite mission (TanDEM-X) of the German Aerospace Center (DLR). The SRTM C-band DEM has been acquired in February 2000 at a spatial resolution of 1 arcsec[64]. We use the void-filled LP DAAC NASA SRTM DEM[65]. Bistatic SAR acquisitions of the TanDEM-X mission are available since 2011[66]. Here, we use Co-registered Single look Slant range Complex (CoSSC) data of the Southern Hemisphere ablation periods of the years 2012–2014. To further minimize elevation offsets due to differences in ice accumulation or time-varying depths of SAR signal penetration into the glacier volume, we use TanDEM-X acquisition dates which are close to the mean SRTM acquisition date (2000-02-16) whenever possible.

Ice flow velocities and reconstructed ice thickness are taken from Millan et al.[3]. These datasets are used for calculation of frontal ablation based on a flux gate approach. To constrain the uncertainty in ice thickness, we further consider ice thickness fields from Carrivick et al.[67] and from the consensus estimate[2] to calculate the standard deviation per pixel. Furthermore, surface velocities close to the glacier fronts are used for a classification of non-calving glaciers. A reliable classification is essential for a comparison between climatic and geodetic mass balance. Glaciers exceeding a velocity threshold of 60 m yr$^{-1}$ are classified as marine- or lake-terminating (MALT) glaciers, glaciers below the threshold velocity are classified as glaciers without significant frontal ablation and treated as land-terminating in this study.

### Atmospheric forcing

Atmospheric input data required for the COSIPY simulation includes air temperature, relative humidity, incoming shortwave radiation, wind speed, air pressure, cloud cover and precipitation. To extend the climatic data beyond the respective measurement periods, we apply a downscaling scheme where we combine statistical downscaling with the application of a radiation model and a model of orographic precipitation.

Following previous studies in southern Patagonia (e.g., refs. 20,21,39), we apply quantile mapping for statistical downscaling of air temperature, relative humidity and air pressure. Quantile mapping is a method of statistical bias correction, where the cumulative distribution function of the model is adjusted to align with the cumulative distribution function of the observation[26,68]. Statistically downscaled air temperature and pressure are adjusted to sea level conditions, interpolated between the available station points via Ordinary Kriging[69], and subsequently spatially extrapolated over the topography using a linear temperature lapse rate of -0.6 K/100 m[20] and the barometric equation, respectively. Relative humidity is likewise interpolated between the recording AWSs with Ordinary Kriging. Wind speed and cloud cover fraction are taken directly from ERA5 and interpolated to the model resolution.

A radiation model[70] is applied over the study site to calculate global radiation over the glacier surface, following the methodology of Temme et al.[20] at the Mount Sarmiento Massif. The model calculates both the direct and diffuse component of the solar radiation based on cloud cover, temperature, humidity and pressure. Corrections are applied for the slope and aspect of the respective grid cell. Shaded grid

cells, either from the terrain or self-shaded, exclusively receive the diffuse solar radiation component[27,70].

Due to the small-scale and episodic character of precipitation events, statistical techniques often fail to infer reliable distributions over complex terrain from coarse global data sets. Furthermore, strong winds limit the reliability of observations in southern Patagonia[43,71]. An orographic precipitation model showed improved performance as compared to extrapolation of observational data using altitudinal lapse rates[72]. The model calculates the orographic portion of precipitation resulting from forced orographic uplift over a mountain[39]. It is grounded on the linear steady-state theory of orographic precipitation, considering airflow dynamics, cloud timescales and processes of advection and downslope evaporation[28,29]. Since the model assumes stable and saturated conditions with unblocked air flow crossing the CDI from west to east[28,39], time intervals that do not fulfill these constraints are excluded. Thresholds and parameter settings are taken from Temme et al.[20]. The total precipitation is calculated by adding the orographic precipitation calculated in the model to the large-scale precipitation. The large-scale precipitation is obtained by removing the orographic component, calculated by running the orographic precipitation model on the ERA5 topography, from the ERA5 total precipitation. Recent elevation- and bias-corrected precipitation products (W5E5, WFDE5) with lower spatial resolution and shorter temporal coverage indicate an ERA5 overestimation over Tierra del Fuego[73]. Comparison of ERA5 daily precipitation with observations supports this finding[74]. To guarantee that the simulated total precipitation at the AWS locations agrees with the observed amounts, we constrain the large-scale precipitation from ERA5 to the annual measurements.

To derive snowfall from precipitation, a logistic transfer function is applied scaling around a threshold temperature of 1.0 °C. A snow drift parametrization is included in the modeling framework to account for snow redistribution caused by the strong westerly winds over the CDI. Locations sheltered from or exposed to wind are identified by a topographic analysis and solid precipitation is redistributed accordingly[75]. Parameters and adjustments to the model are transferred from the Mount Sarmiento Massif[20].

Climatic forcing data are validated on a daily basis with meteorological observations from AWSs that have not been used in the downscaling (Table S1) based on a statistical analysis of mean model bias, root mean square error and correlation. Overall, the performance of downscaled and modeled climate variables is satisfying (Table S2). The agreement of downscaled variables with measurements is improved compared to the raw ERA5 input, confirming the success of the downscaling approach. For further information on precipitation amounts, we compare annual precipitation over the river catchment of Río Betbeder with observed stream flow at the gauging station there, and snowfall with results of a firn core in the central CDI. The former river catchment is located in the northeast of the CDI covering a total area of 146 km² (Fig. 1c). The comparison indicates an underestimation (Fig. S2). However, considering the simplified approach (e.g., neglecting water storage and glaciers in the system), results are satisfying. The firn core site is located at an exposed saddle where we assume important wind erosion. The small-scale local wind field and snowdrift are, however, not fully resolved in our modeling approach, which explains an overestimation in the modeled snowfall.

## COSIPY model

The open-source COSIPY model (COupled Snowpack and Ice surface energy and mass balance model in PYthon)[25] is a physically based model grounded on the concept of energy and mass conservation. It couples a surface energy and mass balance model with a multi-layer subsurface snow and ice model, with the calculated surface meltwater serving as input to the subsurface model[25]. The energy balance model solves all energy fluxes $F$ at the glacier surface:

$$F = SW_{in}(1 - \alpha) + LW_{in} + LW_{out} + Q_{sen} + Q_{lat} + Q_g + Q_{RRR} \quad (1)$$

where $SW_{in}$ is the incoming shortwave radiation taken from the radiation model, $\alpha$ is the surface albedo, $LW_{in}$ and $LW_{out}$ are the incoming and outgoing longwave radiation, $Q_{sen}$ and $Q_{lat}$ are the turbulent sensible and latent heat flux, $Q_g$ is the glacier heat flux and $Q_{RRR}$ the rain heat flux. Melt can occur if the surface temperature is at the melting point (0.0 °C) and $F$ is positive. Under this condition, the available energy for surface melt $Q_M$ equals $F$. Rain and meltwater can percolate the snowpack and cause refreezing in the snow layers. Subsurface melting is possible by penetration of shortwave radiation in the upper snow layers. Solving the surface plus the internal mass balance in the snowpack, COSIPY gives the climatic mass balance (CMB)[76]. The total ablation includes surface melting, sublimation and subsurface melting. Accumulation is the sum of snowfall, deposition and refreezing.

Albedo values are differentiated between snow, firn and ice surfaces. The decay of surface albedo due to snow aging is parameterized following the scheme of Oerlemans and Knap[77]. The albedo depends on the time since the last snowfall and the snow depth. A bulk approach is applied to parameterize the turbulent heat fluxes. COSIPY offers the option to correct the flux-profile relationship by a stability correction using the Richardson-Number or the Monin-Obukhov similarity theory[25]. The latter is applied in this study (Table S5).

We apply COSIPY version 1.4 in the period 04/1999-03/2023 with a 200 m spatial and a 3 hourly temporal resolution. All parameter settings follow the COSIPY set-up in the Mount Sarmiento Massif[20] and are summarized in Table S5. The model performance of COSIPY was positively evaluated in the Mount Sarmiento Massif, where four surface mass balance models of varying complexity were compared. COSIPY results agreed well with the other models as well as with observations of ablation stakes and geodetic mass balance[20].

## Geodetic mass balance processing

Elevation changes are calculated by DEM-differencing of SRTM and TanDEM-X. TanDEM-X DEMs are created based on differential interferometry following an established workflow[9]. First, interferograms are computed from concatenated overlapping acquisitions, phase-unwrapped based on a minimum cost flow algorithm and converted to elevation values using the SRTM DEM as reference surface. Thereafter, the raw TanDEM-X DEMs are iteratively co-registered to the SRTM DEM in the vertical and horizontal plane. Therefore, the 3D offset of each DEM is estimated based on all stable terrain with <25° surface slope excluding water and glacier areas. Finally, a regional elevation mosaic is created by merging all co-registered DEMs in the order of the relative deviation between the SRTM mean acquisition date (February 16th) and the tile-specific TanDEM-X date. The cell-specific TanDEM-X dates are stored with the DEM mosaic and subsequently used to calculate the respective elevation change rate during the SRTM and TanDEM-X DEM-differencing. The mean regional observation period of the elevation change rate measurement is 12.97 years (2000–2013).

To extract glacier-specific mass changes within the geodetic observation period, the elevation change map is masked to the glacier outlines of the 2000 inventory (see Data section). The mean elevation change rate is extracted for each glacier geometry and converted to volume and mass change based on the respective glacierized area and an approximate ice density of 900 ± 60 kg m⁻³. Since glacier area changes during the observation period can bias the derived mass budgets[78,79], the specific mass change rate of each glacier is calculated using the mean glacier area of the 2000 and 2013 inventories following the UNESCO definitions[76].

To estimate the glacier-specific uncertainty budget of our derived elevation change rates, we calculate the remaining vertical deviation after DEM co-registration of stable terrain[9,79], i.e. raster cells which are

not glaciers or water areas, in the vicinity of each glacier. First, all stable terrain cells within a 5 km buffer radius of each glacier outline are masked and filtered with a 2-98 % quantile filter. The selected cells are then aggregated within 5° slope bins to account for the dependence between surface slope and vertical deviations between different DEMs, and the respective standard deviation is extracted. Finally, the total vertical accuracy of the elevation change measurement of each glacier is estimated by weighting the offsets (standard deviations) of each stable terrain slope bin by the corresponding glacier area of the same slope bin. In addition, we multiply the derived vertical elevation change uncertainty of each glacier by a constant factor of 2 based on the glacier area fraction without valid DEM-differencing measurements as the accuracy of the glacier-specific elevation change estimates is also related to the spatial coverage of the TanDEM-X DEMs. To convert the glacier-specific elevation change uncertainty into mass change uncertainty, we use ±60 kg m$^{-3}$ as the error budget of the assumed mean glacier ice density[54].

### Model calibration and validation

Due to the limited in-situ observations in the Cordillera Darwin, calibration and validation of the CMB are a major challenge. With the large model domain and the high temporal and spatial resolution, resulting in a massive computational effort, intense model calibration is not feasible. Instead, the downscaling procedure and optimal parameter setting are grounded on the expertise gained at the Mount Sarmiento Massif, located at the western edge of the CDI[20]. Sensitivity runs are applied for further optimization, where the methods for the generation of the atmospheric input fields (relative humidity, wind velocity, air temperature) are varied, addressing the challenge to realistically reproduce the zonal climatic gradients. Temme et al.[20] conclude that calibrating against regional satellite observations of mass change significantly improves the performance of CMB models. Following this approach, we rank the sensitivity runs based on the highest agreement with regional geodetic mass balance, as observed with satellite remote sensing, for all glaciers with no frontal ablation (region-wide average -0.27 m w.e. yr$^{-1}$). The highest ranked run (region-wide average -0.23 m w.e. yr$^{-1}$) agrees well with the geodetic observations and is presented in this study.

For model validation, we compare the climatic with the geodetic mass balance of each land-terminating glacier on a catchment level (catchment information not used during calibration). MALT glaciers are excluded because they also lose mass at the calving front due to ice dynamics. To reduce uncertainties, we limit the comparison to glaciers exceeding an area of 3 km$^2$. This gives a validation dataset of glaciers covering ~37 % of the total glaciated area of the CDI. Model performance is quantified by the root mean square error between the glacier-specific climatic and geodetic mass balance for these glaciers, resulting in a model error of ±0.62 m w.e. yr$^{-1}$. Stake measurements at Schiaparelli Glacier[20] and Martial Este Glacier[80] serve as additional validation of melt on the western and eastern edges of the CDI (Table S6).

### Frontal ablation

In this study, we apply two different methods to determine frontal ablation for the entire CDI and the individual marine- and lake-terminating glaciers in the Cordillera Darwin. Firstly, we apply a mass budgeting, where the residual of the total glacier mass balance ($\Delta M_{tot}$) from geodetic observations and the CMB ($\dot{B}$) simulated with COSIPY provides the frontal ablation ($A_f$): $A_f = \Delta M_{tot} - \dot{B}$[16]. Uncertainties in frontal ablation consist of the uncertainties in the glacier-specific CMB (see Model calibration and validation) and the uncertainties in the glacier-specific geodetic mass balance (see Geodetic mass balance processing) following classical Gaussian error propagation. Since the former uncertainty already includes random errors in the geodetic mass balance, the resulting uncertainties of frontal ablation values are likely an overestimation.

Secondly, we apply a flux gate approach[15,23]. Here, frontal ablation is calculated based on the discharge ($D$) at a flux gate located upstream of the glacier front and the CMB downstream of the flux gate ($\dot{B}_{FG}$): $A_f = -D - \dot{B}_{FG}$. $D$ is calculated by integrating the product of ice thickness and surface velocity perpendicular to the gate. For our study site, the lack of ice thickness measurements in the CDI makes modeled ice thickness uncertain. To quantify the uncertainty in the ice thickness, we calculate the standard deviation of the available ice thickness products from Carrivick et al.[67], Millan et al.[3] and the participants in the consensus estimate covering the CDI[2]. Time discrepancies are corrected for by satellite-derived elevation changes.

## Data availability

Average annual fields of the simulation data generated in this study are available in the Zenodo database under ref. 81 (https://doi.org/10.5281/ZENODO.14003166). Temporally higher resolved model data is available from the corresponding author on request. The data used in Figs. 3, 4, S2, S3, S5, S6, S10 and S11 are provided in the Source Data file.

## Code availability

The code for the COSIPY model (version 1.4) is available at Github (https://github.com/cryotools/cosipy) and Zenodo under ref. 82 (https://doi.org/10.5281/ZENODO.4439551).

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

## Acknowledgements

FT was funded by the German Research Foundation (DFG) within the MAGIC project (FU 1032/5-1) and the RESPONSE project (TA 1719/2-1) and by the Bavarian State Ministry of Science and Arts under the Elite Network Bavaria (Grand reference: IDP M³OCCA). JJF has received funding from the European Union's Horizon 2020 research and innovation program via the European Research Council (ERC) as a Starting Grant (StG) under grant agreement No 948290. JA has received funding from the CNPq project (308831/2022-5) and the FAPERGS project (21/2551-0002034-2). RJ has received funding from the ANID FONDECYT Regular project (1231707) and INACH project (FR_04-23). EI was funded by the University of the Basque Country (UPV/EHU) under Grant PIF17/182 and by the Basque Government under the Consolidated Research Group IT1029-16 and IT1678-22. The authors are grateful for the scientific support and resources provided by the Erlangen National High Performance Computing (HPC) Center (NHR@FAU) of the Friedrich-Alexander-Universität Erlangen-Nürnberg (FAU). NHR funding is provided by federal and Bavarian state authorities. NHR@FAU hardware is partially funded by the DFG—440719683. TanDEM-X data were kindly provided free of charge by the German Aerospace Center (DLR) under AO mabra_XTI_GLAC0264. The authors want to thank the Chilean National Forest Corporation (CONAF) for authorizing the field work in the Cordillera Darwin, Parque Nacional Alberto de Agostini. We want to truly thank David Farías for his valuable support of this study in terms of data identification and support in correspondence.

## Author contributions

The concept of this study was developed by J.J.F., F.T and M.S., F.T. implemented the simulations with the support of J.J.F. In situ observational data were collected and provided by E.I., R.G., R.J., J.A.N., I.G. and D.T. Elevation change rates and geodetic mass balances were processed by C.S. F.T. led the writing process with the support of J.J.F., C.S., M.S., E.I., R.G., R.J., J.A.N., I.G. and D.T.

## Funding

Open Access Funding enabled and organized by Projekt DEAL.

## Competing interests

The authors declare no conflict of interest.
