## [Transparent Peer Review file · Nature Communications]

Climate's firm grip on glacier ablation in the Cordillera Darwin Icefield, Tierra del Fuego

Corresponding Author: Ms Franziska Temme

Version 0:

Reviewer comments:

Reviewer #1

(Remarks to the Author)

In this work, Temme and collaborators present a climatic mass balance forced by reanalysis data validated with in situ observations. Frontal ablation is also estimated. The methods are standard for this kind of research. The novelty is related to the study area, as the modelling is applied to the glaciers of the Cordillera Darwin Icefield (CDI) in southernmost South America, an area poorly studied. The main conclusion is atmospheric forcing is the main driver of the ablation in the region. The work is well-written and easy to follow and the figures are mostly clear, but I think some clarifications need to be made before publication. I hope this review helps the authors improve the manuscript.

I suggest a small change in the title, including "ablation" instead of "melt". Results of the manuscript show that melt is the main component in the total ablation processes, i.e. higher percentage (74%) compared to frontal ablation (26%). Melt by definition depends on the atmosphere conditions (could be also by oceanic or lake conditions but this is not directly quantified here). If I understand, I think the title looks to remark that the climate or atmosphere drives the "ablation" as stated in lines 260-261.

The sentence in L275 and repeated in L347 is a bit confusing. How the mass loss is controlled by atmospheric conditions if the CMB is positive (0.02 m w.e. yr⁻¹) or close to balance (L332)? Results show climatic ablation is quite larger than frontal ablation (L275-277) and this is forced by the atmosphere, I understand this, but in terms of mass balance, the overall CMB is still positive for the period meaning that CDI has been climatically balanced in the recent two decades (L380). In consequence, overall no mass loss forced by atmosphere conditions between 2000-2022. Or do you refer to the last six years of CMB modelling? or just to land-terminating glaciers that show negative specific CMB (L145-146)? Please clarify this.

No justification for the period. For instance, why 22 years and not 30 years? In the title, the word "Climate" is used, and by definition, the climate is a 30-year period (WMO). If the 22-year period is well justified, maybe change "Climate's" to "Atmosphere's".

As usual in glacier modelling-approach investigations some choices need to be better justified, for instance, the use of a constant air temperature lapse rate over a spatial variable lapse rate. Also, other sources of uncertainties are not (at least) mentioned in the manuscript. For instance, no mention of debris cover areas. Is this area important to quantify melt rates at CDI?

Minor comments

L 38-39: "The large altitudinal range...." I don't understand this sentence. How do the location and the climatic conditions control the altitudinal range of the CDI? Or do you refer to the climate conditions extreme gradients? Please clarify.

L59-60: Reference 17 includes a frontal reconstruction of Marinelli Glacier back to 1910.

L63: I suggest including the area of the marine-terminating and the lake-terminating glaciers separately.

L107: Why this period?

L110: Any insight on why the correlation with Isla Hoste AWS is low?

L114-115: Ok, but you also show individual glacier CMB results, for instance in Fig. 1. Are these values, especially the accumulation, reliable?

L123-124: I understand what you mention here, and agree, but be aware that relative humidity is an indicator of how close to saturation the atmosphere is and depends on the air temperature.

L145-146: Similar to a previous comment, how reliable is the CMB estimation for individual glaciers? I agree that the regional CMB is a good approach for CDI considering the uncertainties, lack of measurements and differences in the accumulation rate reported in L114 but, along the manuscript, CMB of individual glaciers is used for comparisons.

L152-153: I suggest adding a Figure (could be in Supplementary Material) to explore the differences in the CMB between land and marine-lacustrine glaciers.

L176: Change Juli by July.

L297-308: Why present ELA results in the Discussion section and not in the Results sections?

L317-321: Any clue for this behaviour? Southern Annular Mode or another mode of interannual or interdecadal variability?

L315-328: Missing references about glacier energy balance in Patagonian glaciers that could be useful to add to the discussion.

L329: Also missing references about surface melt trends and projections in Patagonian glaciers. I suggest comparing your findings with previous works at regional scale. Considering the uncertainties that are mentioned in the paper I suggest keeping a regional point of view for comparisons. Some comparison by glacier is fine, but the approach here is regional.

L380-381: This sentence seems more appropriate for the climate or atmosphere forcing on mass balance.

L446: Is there any reference for this velocity threshold?

Reviewer #2

(Remarks to the Author)

The study models the surface mass balance of all glaciers in the Cordillera Darwin Icefield (CDI) in Tierra del Fuego from 2000 to 2019, utilizing the open-source surface energy balance model COSIPY (Sauter et al., 2020), which is forced by downscaled ERA5 data. The model parameters are primarily calibrated with in-situ mass balance observations from one or a few glaciers in the region. Notably, this calibration process is based on the authors' previously published work (Temme et al., 2023). The difference between the geodetic mass balance and the simulated surface mass balance for lake- and ocean-terminating glaciers is used to estimate mass loss due to frontal ablation.

Overall, the paper is well-written, but there are areas where additional methodological detail would enhance clarity (elaborated further in my specific comments). The authors highlight the novelty of the study as "the first simulation of the climatic energy and mass balance of glaciers in this region." However, given that the same first author has previously applied the COSIPY model (developed by Sauter et al., 2020) to a subset of glaciers in the region (Temme et al., 2023), this work may be better framed as an extension of that earlier study, expanding the model's application to a broader set of glaciers.

The method used to estimate frontal ablation raises some concerns, particularly regarding the quantification of uncertainties (detailed in my specific comments). A more comprehensive uncertainty analysis would strengthen the credibility of the results, particularly in relation to frontal ablation estimates.

In summary, while the study provides valuable insights into the region, the novelty of the contribution could be more clearly articulated. Additionally, addressing some critical methodological elements, such as a more rigorous uncertainty analysis, would significantly enhance the robustness of the results.

Specific comments:

Abstract: The authors should include the period of the simulations in the abstract. It took me some time to determine that the simulations cover 2000–2019, and having this information up front would clarify the scope of the study.

Line 77-78: "...melt, which is strongly linked to air temperature"

One of the references cited here used a positive-degree-day (PDD) model to simulate melt, so linking melt to temperature is

an inherent assumption of the model.

Line 78-79: "Despite these local efforts to determine the mass balance of individual glaciers, a systematic approach for a CDI-wide estimate of the CMB is to this day missing."

However, Temme et al. (2023) (reference #20) addresses not just individual glacier modeling but also regional-scale modeling. The authors may want to reconsider this statement, as it somewhat underrepresents the existing work in this area.

Line 80: "This lack of knowledge can be addressed with a well-informed modelling product of CMB."

The term "well-informed modeling product of CMB" is unclear. It would improve clarity to specify what is meant by this term and what type of model product the authors are referring to.

Line 82-83: "Thus, with the climatic loss term available, the mass budget of the CDI glaciers can successfully be closed and give the dynamic loss as residual."

There seems to be a disconnect between this statement and the preceding one. The total mass loss is derived from geodetic estimates, not the climatic loss term. I suggest rephrasing this to clarify that the dynamic loss is inferred from the residual of the modeled climatic mass balance (CMB) and geodetic mass balance. Additionally, errors in both the CMB and geodetic estimates could be significant, making it challenging to assume that the residual solely represents dynamic loss.

Line 94: "In this way, an unprecedented attribution of the observed mass loss to climatic and ice-dynamic forcing becomes possible."

While this is an important step forward, the authors should also account for the uncertainty in the modeled CMB and geodetic estimates. Without addressing these uncertainties, the attribution may not be as straightforward as suggested here. For instance, Hugonnet et al. (2021) point out that the accuracy of geodetic mass balance estimates can decline when moving from multi-annual to annual timescales.

Line 96-97: "The energy and mass balance will be examined using the physically-based COupled Snowpack and Ice surface energy and mass balance model in PYthon (COSIPY)."

It would be helpful to explain why COSIPY was chosen for this study. Temme et al. (2023) compared COSIPY with simpler models, such as PDD approaches, and noted that some of the model parameters, like those in the albedo parameterization, still require calibration with mass balance observations. If the model is calibrated with geodetic mass balance, how can it provide an independent estimate of the ice-dynamic component? If the in-situ mass balance data from one or a few glaciers is used for calibration, how do the authors address the potential issue of limited parameter transferability across the CDI?

Line 422: "...since comparison with the Randolph Glacier Inventory (V6) has revealed strong weaknesses of the latter for the southern Andes."

The term "weaknesses" seems too vague and possibly unsuitable here. It would be more informative to specify what shortcomings of the Randolph Glacier Inventory were identified in the region. Does it poorly represent glacier outlines in this area?

Line 428-430: "Since elevation changes cover the period 2000 to 2013, we manually produced outlines for 2013, which have been missing so far. Thus, inventories available in this study cover years 2000, 2013 and 2019 oriented on inventory availability and geodetic data."

The phrase "oriented on inventory availability" is unclear. I suggest rewording for clarity. Also, more detail is needed on how the 2013 outlines were manually produced, as this is a crucial methodological step.

Line 431-441: What is the final output of this method? Is it glacier-wide geodetic mass changes over the entire period? Please clarify the time period and spatial scale covered by these mass change estimates.

Line 464: Why was a radiation model used instead of directly employing incoming solar radiation from ERA5? Have you evaluated ERA5's radiation data for this region, and if so, why was it deemed unsuitable for glacier-scale modeling? Additionally, if you use cloudiness data from ERA5 for the radiation model, how reliable is this parameter, given that ERA5 cloud data may be less accurate than its radiation estimates?

Line 481: "...which is obtained by removing the orographic component from the ERA5 precipitation."

Does ERA5 provide an orographic component of precipitation? Based on Hersbach et al. (2020), ERA5 supplies large-scale and convective precipitation, but not explicitly an orographic component. Please clarify this point, as convective precipitation is not synonymous with orographic precipitation.

Line 523: "Albedo values are differentiated between snow, firn and ice surfaces. The decay of surface albedo due to snow aging is parameterized following the scheme of Oerlemans and Knap."

Given the critical role of albedo in surface energy balance (SEB) modeling, have you evaluated the modeled albedo against observations from your study area? If in-situ albedo data is unavailable, MODIS-derived albedo could be a useful comparison. Without any evaluation, it's unclear how reliable the albedo model is for this region, and its validation elsewhere (e.g., in the European Alps) may not be sufficient due to spatial variability in albedo characteristics.

Line 526: "correction using the Richardson-Number or the Monin-Obukhov similarity theory."

Which approach was ultimately used? Further, how were roughness lengths determined? Have you compared your modeled turbulent heat fluxes with measurements from glacier sites in your region to evaluate performance?

Line 529: “was positively evaluated”

This phrase is unclear. Could you elaborate on what is meant by “positively evaluated” in this context?

Line 532: “COSIPY results agreed well with the other models as well as with observations of ablation stakes and geodetic mass balance.”

While this is a useful finding, it would be important to specify which parameters in COSIPY were calibrated and how this calibration was performed. As noted earlier, calibration with geodetic mass balance or in-situ data has implications for the interpretation of the results and the separation of the climatic and dynamic components of mass balance.

Line 559: “Sensitivity runs are applied for further optimization.”

Please clarify what parameters or inputs were varied in the sensitivity runs. Were different datasets or methods used for calibration, or were the model parameters adjusted? Additionally, it is crucial to provide a breakdown of the uncertainties that arise from your modeling approach.

“Model performance is quantified by the root mean square error between the glacier-specific climatic and geodetic mass balance for the individual catchments.”

How do you address the fact that geodetic mass balance includes both the climatic and ice-flow components? Are you comparing glacier-wide mass balance only? Please provide more details.

Line 577-578: “Uncertainties of the glacier-specific CMB, constrained by model validation, directly translate into the uncertainties of frontal ablation estimations.”

More details are needed on how the uncertainties in glacier-specific CMB are quantified. Furthermore, uncertainties in geodetic mass balance estimates should also be considered. Ideally, validation of frontal ablation estimates should be conducted on a study glacier where in-situ measurements of frontal ablation, ice-flow, and surface mass balance are available.

Line 582-583: “For our study site, the lack of ice thickness measurements in the CDI makes ice thickness highly speculative. This directly translates into elevated uncertainties for frontal ablation.”

Could you provide more details on how these uncertainties due to speculative ice thickness were quantified? It would strengthen the paper to explain the methodology or assumptions used to assess these elevated uncertainties.

Results section:

Since you are using a surface energy balance (SEB) model for ablation, it would be informative to estimate the individual contributions of key SEB components (incoming shortwave and longwave radiation, albedo, sensible and latent heat fluxes) to the total annual melt. While some information is provided, it would be beneficial to present a more systematic breakdown of how each term changes over the study period, both temporally and spatially. In particular, further discussion on albedo and incoming longwave radiation beyond the estimates of trends would provide valuable insights.

Line 204: “Our results show that the annual CMB (Fig. 3c) is mainly controlled by air temperature (Fig. S3a).”

The word “controlled” may not be the best choice here. Temperature influences the CMB model primarily through the bulk method for sensible heat fluxes, which you indicate contribute around 30% to melt. Perhaps what you mean to say is that the inter-annual variability in CMB is most strongly correlated with temperature. Additionally, have you considered correlating CMB with albedo? Several studies (e.g., Davaze et al., 2018; Williamson et al., 2020; Xiao et al., 2022) demonstrate that mass balance is often strongly correlated with summer albedo.

Line 230-236:

It was noted earlier that the flux-gate approach carries large uncertainties due to the lack of ice-thickness estimates, yet the reported errors for CDI frontal ablation using this approach are smaller than those from the method based on the difference between geodetic and CMB estimates. How is the total error from the flux-gate approach assessed? It would be helpful to include more detail on this in the Methods section. Additionally, how are the error estimates for frontal ablation of individual glaciers calculated?

Table S5:

While you’ve listed the roughness length for momentum, it would also be useful to include the roughness lengths for temperature and humidity, as these are also important in the bulk method used in your SEB model. Additionally, how were the roughness lengths determined? The roughness length of momentum for ice seems quite small (on the order of 10^{-4} m). Typically, roughness lengths for ice are closer to 10^{-3} m (see Hock, 2005 for reference). Could you clarify how this value was chosen and whether it might be underestimated?

Reviewer #3

(Remarks to the Author)

This study produces the first climatic mass balance (CMB) simulation over the Cordillera Darwin Icefield (CDI), one of the largest ice bodies in the Southern Hemisphere. These outputs are differenced from satellite-derived elevation change measurement to calculate the mass loss due to iceberg calving and subaqueous melting (collectively termed frontal ablation) for the first time. Results show that there is a strong climatic gradient associated with westerly winds across the CDI which definitively influences glacier mass balance. Frontal ablation accounts for 26% of the total mass loss in the CDI. Over the 22-year study period, surface melt has increased (+0.18 m w.e. yr⁻¹ per decade). While frontal ablation is important for

predicting the evolution of specific glaciers, atmospheric conditions are the main control on glacier evolution.

The study results and conclusions are valid, robust, and represent an important contribution to the field of glaciology. The manuscript is well written, and the figures and tables are formatted properly and of excellent quality. The suggested changes are minor and include drawing conclusions regarding the contrasting glacier behavior in the CDI within the discussion, applying a density conversion of $850 \pm 60 \text{ kgm}^{-3}$ to convert the geodetic change to mass change, and modifying the introduction to improve readability and clarity.

Kind regards,
Nicole Schaffer, PhD.

General comments

Validity: The results and conclusions of this study are valid and robust.

Significance: Given that the CDI is one of the largest ice bodies in the Southern Hemisphere and contributes ~5% of the total loss in South America (Braun et al., 2019), the results are quite important for the field of glaciology and climate change science. The CMB data set produced provides a second method to estimate mass loss that complements the satellite-derived elevation change measurements calculated here and previously published. Together these provide a more robust estimate of mass loss for the CDI, especially considering they are in close agreement. Confirming that climatic variables exert the greatest influence on glacier evolution, insight into how these variables are distributed over the CDI, and knowledge of how much loss is attributed to frontal ablation and how this varies through time helps tremendously for modeling the evolution of these glaciers in the future.

Data and methodology: The approach is valid, reproducible, and meets the standards presented in published work within highly respected journals. The data quality is good and the figures and tables provided both in the main text and SI support the research and illustrate the results very effectively. In general, the manuscript is well written.

Analytical approach: The analysis is robust and the statistical results presented are appropriate.

Clarity and context: The manuscript text is clear and accessible. Sufficient context has been provided with the appropriate references and the results have been placed in the context of previous literature. The only section that needs some work in terms of the readability and clarity is the introduction (see comments in the “suggested improvements” section).

References: Yes, the manuscript references previous literature appropriately.

Your expertise: With respect to the CMB I have published using more simplistic models (e.g. enhanced temperature-index model) and have a strong background in glaciology and mass balance modelling theory (MSc and PhD in glaciology), but have not published first-author articles using a full energy balance model. With respect to elevation change measurements, I have a strong background in this area with publications on the subject but have not processed interferometry data sets myself. I have published articles including calculations for frontal ablation.

Suggested improvements:

A sub-objective of this manuscript is to gain insight and draw conclusions on the contrasting glacier behavior in the last two decades. This is addressed directly and possibly indirectly in several locations within the results and discussion sections. For example, L262-274 where the four advancing glaciers are discussed in detail. L151-160 where the diversity is related to the hypsometry, aspects and MALT versus land-termination of the respective glaciers. In the results/discussion the authors discuss how more precipitation in the south-western part of the CDI result in more positive mass balance here and more losses in the north-east (e.g. L293-296). The impact of the ELA on mass balance is also discussed (e.g. L297-308). It would be good to bring together all the above points and any other highly relevant results/information within the discussion section to draw some conclusions on why we observe contrasts in mass balance over the CDI. Particularly for those glaciers with a positive mass balance. Some or all of the text on L151-161 and L262-274 could be deleted from the results and moved to the discussion. If relevant, incorporate the four glaciers dominated by frontal ablation into this discussion. It would also be good to incorporate a very brief discussion on why MALT glaciers show on average a more positive CMB than land-terminating ones (1-2 sentences).

A density of $900 \pm 60 \text{ kgm}^{-3}$ has been used to convert geodetic change to mass change in this manuscript. This is the standard approach. However, it is well known that the “density” for geodetic volume-to-mass conversion is not constant and is systematically lower than ice density in most cases (Huss et al. 2013). This is due to the accretion/removal of low-density firn layers and changes in the firn density profile with positive/negative mass balance. A value of $850 \pm 60 \text{ kgm}^{-3}$ is appropriate for most situations. Therefore, I would suggest applying a density of $850 \pm 60 \text{ kgm}^{-3}$ in addition to $900 \pm 60 \text{ kgm}^{-3}$. This could be included within the discussion section. New calculations for the dynamic component should be included and any changes to your main observations/conclusions should be discussed.

In general, the manuscript reads very well. The only section that needs some work in terms of the readability and clarity is the introduction. This section could benefit from some restructuring and minor sentence modifications. To that end, I have made some suggested changes and included this as a pdf with tracked changes. Here, the sentence starting on L67 has been moved after the mass budgeting is explained as this seems to flow better. I have modified the aims of the study to

improve clarity and have also made many suggestions to improve the readability (see tracked changes pdf). Details of the studies quoted have been removed as this breaks the flow of the introduction. These sentences could be incorporated nicely into the discussion (e.g. sentence on L76 could be incorporated in the sentence on L309, the sentence starting on L86 could be incorporated in the paragraph on L339 and the sentence starting on L89 is redundant as this information is already in the discussion). Finally, I have added a sentence on L70 to first to explicitly make the reader aware that most of the measurements available (e.g. those described on L53-55) are geodetic and include both loss terms before discussing the mass budgeting approach.

Detailed comments:

Introduction

Please see the tracked changes .pdf

Results

L110 What is "local"? Please define explicitly.

L110-112 This sentence states that "A comparison of local precipitation with a firn core in the central Cordillera Darwin shows an overestimation", but there is no data to back this statement up. Please provide a quantitative comparison (e.g. amount of overestimation in SWE). Likewise, the statement "catchment-wide precipitation about 50 km downwind indicates an underestimation" is qualitative (Figure S2). Please provide a quantitative comparison as well (e.g. mean and standard deviation for the difference in streamflow between the stream gauges and downscaled data).

L114 Consider changing "As we aim at a..." to "As our aim is a..."

L121 I assume the results presented here (e.g. 5.2°C) are from the downscaled climate data, but it would be good to explicitly state this so it is clear.

L132 Consider changing to "...wind also show an increasing ..."

L172 consider rephrasing as "...decreasing snowfall towards the east results in CMB decreases...."

L237-238 Here the author indicates that "...the majority of MALT glaciers show only low frontal ablation." Can you quantify and specify what you mean by "low frontal ablation"? Are you referring to low frontal ablation rates for the majority of MALT glaciers compared to the average frontal ablation rate? It would be helpful to provide a simple reference value (e.g. the average frontal ablation of all glaciers compared to the average for 50% of the glaciers with the lowest values)

L290 I assume you mean global warming rates. If so modify this sentence to "...exceeds global warming rates..."

L360 did you mean "temperate" instead of "temperature" here?

L353 This paragraph could be modified to improve the readability and make it more concise in some places. I have simply highlighted the text that should be reviewed in the tracked changes pdf.

L238 change "For few.." to "For a few...". Same comment for L253.

L251-253 The first two sentences are a bit redundant and could be reduced to: "With the CMB and the frontal ablation available, we can disentangle the total glacier mass changes observed from remote sensing into climatical and dynamical forcing."

Figure 1 For clarity it would be good to add the years associated with the CMB data here. For example, the first sentence could be modified to "... (CDI) over the period 2000-2022."

Figure S2 consider modifying the figure caption from "...to precipitation over the river catchment..." to "...to downscaled precipitation over the river catchment..." so it is clear that the yellow data set is the downscaled data set.

Discussion

L293 modify "...moisture, more rich in snow conditions..." to "moister, snow-rich conditions..."

L343 change "Alike" to "Like"

L378 change "of glacier..." to "and glacier..."

L378 It would be good to highlight why this location is unique and very relevant for climate science. Consider modifying to: "...at this unique location which is part of the only continental landmass crossed by the Southern Hemisphere westerly wind belt." ...or something similar that highlights why the location is unique.

L353-354 consider modifying to "...we satisfactorily constrain the CMB."

L355 consider modifying to "However, energy fluxes..."

L359 consider modifying to "Furthermore, the considerable..."

L386-387 The current statement about climatic warming is vague and not very convincing. Consider modifying to "...climatic warming has definitively impacted CDI glaciers and their evolution is mainly controlled by atmospheric conditions."

Figure S4 In the caption it says that "The color scheme gives the model error" while in the legend the colour scheme is labeled as the "difference climatic to geodetic mass balance." I am assuming the legend is correct, so the caption should be modified.

Figure S6 change "Snowfall SNOW" to "snowfall (SNOW)"

Table S1 – Consider modifying the label for column one from "Name" to "Station name". Also change the header "responsibility" to "responsible"

Table S2 – label the first column (e.g. Station name).

Data and Methods

L395 replace "...temperature via relative..." with "...temperature, relative..."

L429 consider removing "..., which have been missing so far" as this is redundant because it is clear from the previous sentences that there are no outlines for this year.

L430 consider removing "oriented on inventory availability and geodetic data" since this is redundant. It is already mentioned in the preceding sentences.

L442 replace "had been" with "are"

Table S2 From the text it sounds like this comparison is between the downscaled climate data and AWS data only (e.g. river catchment data and firn core are not used). It would be good to state this explicitly by modifying the caption to: "...compared to the AWS observations."

References

Huss M (2013) Density assumptions for converting geodetic glacier volume change to mass change. *The Cryosphere* 7(3), 877–887. doi: 10.5194/tc-7-877-2013

Version 1:

Reviewer comments:

Reviewer #1

(Remarks to the Author)

Thank you to the authors for answering and clarifying my concerns regarding their manuscript. I understand and agree with the answers and changes introduced in the manuscript.

Just some (very) minor comments:

L49-50: However, stratospheric ozone depletion also forces storm-track poleward trend. Recovery of the ozone is projected so this could counteract the poleward trend. Do the authors analyze this scenario? And how could this affect CDI mass balance?

L179 and L300-313: I suggest presenting the ELA results as m a.s.l. (I'm sorry I didn't note this in the first version.)

L188: delete space between 15 and %

L556: so, Which one is used in the bulk-approach applied here?

Answering to the authors, there are several energy balance studies for short-term in Patagonia linked to Japan's research program on Patagonia. Most of them, were published in the *Bulletin of Glacier Research* during the 80's and 90's.

Reviewer #2

(Remarks to the Author)

Dear Editor,

I have reviewed the authors' responses to my comments; however, I have not had the opportunity to thoroughly review the revised manuscript itself. Based on the responses provided, I am generally satisfied and believe the authors have addressed my main criticisms.

That said, I would like to bring to your attention a few minor points from their responses that may require further clarification:

The authors state: "Taking only the land-terminating glaciers, the dynamic component is zero, and the climatic and geodetic mass balance are directly comparable." While this is true when considering glacier-wide (or region-wide) mass balance, it does not hold if distributed specific mass balance (e.g., mass balance versus elevation) of individual glaciers is considered. Geodetic mass balance for any glacier (land-terminating or not) always reflects a combination of surface mass balance (climatic) and ice flow (dynamic). This distinction was not fully addressed in the authors' response, and I suggest that this clarification should be explicitly incorporated into the manuscript to ensure methodological accuracy.

The authors also mention: "We are comparing the glacier-wide integrated mass balance (called specific mass balance) for each land-terminating glacier." However, glacier-wide mass balance is not the same as specific mass balance, and these terms should not be conflated. This mislabeling caused confusion in their responses and could lead to further misunderstandings in the manuscript. I recommend that the authors refer to the *Glossary of Mass Balance* (https://wgms.ch/downloads/Cogley_etal_2011.pdf) to ensure that terminology is used correctly and consistently throughout the paper.

Finally, the authors' use of the term "ice dynamics" appears to describe glacier calving or frontal ablation. However, ice dynamics is often synonymous with ice flow, which occurs in all glaciers regardless of their terminus type (land or marine).

To avoid confusion, I suggest the authors revise this terminology and use more precise language when describing these processes (again, see the Glossary for guidance).

I hope these observations are helpful in guiding the final revisions. Please let me know if any further input is required from my side.

Reviewer #3

(Remarks to the Author)

Dear authors,

The manuscript has been improved significantly through the review process and all my previous comments and additional concerns have been addressed directly or indirectly through your response to the reviewers. The remaining comments I have are minor.

Kind regards,
Nicole Schaffer, PhD.

Detailed comments:

L38-39 Suggestion to delete "...with glaciers descending down to sea level.." as this is redundant since it can be inferred from the previous sentence. I realize this was included to respond to a comment from reviewer 1 so if you decide to keep it for that reason that is ok. The modification to include "...inducing high mas input.." in response to reviewer 1 is a helpful addition but I would suggest replacing "at the top" with something more explicit such as "at the highest elevations".

L46-47 suggestion to modify this sentence to "...not only the formation of clouds...but also the global ocean..."

L78-79 The term "this way" is somewhat vague and informal. Suggestion to replace this with "With this dataset, ...". Also consider replacing "given" with a more appropriate word such as "produced".

L272-278 The sentence starting on L272 and the rest of this paragraph is mostly duplicated in the new discussion paragraph on advancing glaciers. Given that this part of the paragraph is more suited for the discussion since it is an interpretation of the results, I would suggest deleting this text from the results and merging it with the new paragraph in the discussion (see my comments further down for that paragraph).

L313 change "extends generally" to "generally extends"

L315 Replace the term "exposition" with a more suitable term. For example, "aspect".

L311-320 In the second sentence within this paragraph, the author highlights that MALT glaciers show more positive CMB. This prompts one to question whether all the advancing glaciers are MALT glaciers? I would suggest adding a sentence starting on L315 (if it is true that they are all MALT, otherwise modify!) to answer this question and aid in linking the sentence on MALT to the discussion on advancing glaciers. Additionally, moving the last sentence to L317 would help with the readability and cohesive flow of ideas. Finally, I would suggest merging this paragraph with the text on L272-278 to reduce unnecessary repetition in the manuscript as follows:

"...century. While these advancing glaciers are all MALT, they have medium to low frontal ablation which, in addition to a thickening further upstream, suggest a primarily climatic control. We ascribe these climatological favorable conditions to the glaciers' hypsometry and aspect. All four glaciers have their origin at the high-elevated central plateau of the CDI (above 2000 m) and steep topography resulting in high snowfall amounts and above-average accumulation-area-ratios (between 0.72 and 0.88 compared to the CDI average of 0.61). Glaciers with high accumulation-area-ratio are often less sensitive to changes in ELA and, thus, warming due to the small and steep ablation area¹⁵. Southward exposures of Garibaldi, Guilcher Oeste and Guilcher Este glaciers further reduce surface ablation (Fig. S8a). Altogether, these characteristics ultimately lead to high snowfall amounts together with reduced surface ablation (Fig. 2c, Fig. S9d), favoring mass gain."

L311-320 (b) Does the strong precipitation gradient across the CDI have an impact on the contrasting glacier behaviour? If you consider the answer relevant and worth mentioning, please add a sentence to this paragraph.

L365 suggestion to replace "...ablation at the CDI...is in a similar..." with "...ablation for the CDI...is a similar..."

L377 replace "...study are.." with "...study is..."

L379 replace "verifiably" with "verifiable"

L380 suggestion to replace "...should be handled with certain care.." to "...should be interpreted with care..."

L443 Consider changing “Required variables comprise...” to “Required variables for this study are ...”

L445-447 consider changing sentence to “For downscaling of precipitation (53.75-55.50°S, 74.50-73.25°W) information about air temperature, relative humidity, wind vectors and geopotential height between 850 and 500 hPa is needed.” The term “we need” is too informal for a publication in my opinion.

L487-488 Reading the phrase “...where the cumulative distribution function of the model is transferred to the cumulative distribution function of the observation” could be interpreted to mean that the cumulative distribution of the observation data is corrected by aligning it with the cumulative distribution of the model output. From my limited understanding of quantile mapping bias correction, I understand that the correction is the other way around. The quantile distribution of the model outputs is shifted to align with the quantile distribution of the observed data. Consider modifying the sentence to “...where the cumulative distribution function of the model is shifted to align with the cumulative distribution function of the observations”

Table S2 The RMSE for the station Rio Azorpado is quite low (0.33). Do you know why this might be? If relevant, include the explanation in the manuscript.

Dear editor and referees,

We would like to thank you very much for the detailed and constructive comments to our manuscript. We are convinced that the revisions have improved the overall quality of our manuscript. Before we reply in detail to the individual comments, we would like to mention the most important changes implemented in the manuscript:

1) The introduction was revised following the suggestions of the reviewers (mainly reviewer 3).

2) We revised and streamlined the methods section and provide now more details especially on the model calibration and validation and the uncertainty analysis. In this way, we hope to clarify the distinction between the actual calibration targets on regional scale and the glacier-specific observations used for uncertainty metrics.

3) The uncertainty analysis of the frontal ablation estimation has been extended and comprises now the measurement error of the (specific) geodetic mass balance. We follow a statistical uncertainty propagation. Uncertainties have been updated throughout the manuscript.

4) We extended the modelling period by one year, now covering 2000-2023. Changes in numbers and trends are minor, corroborating the robustness of our CMB results.

Kind regards,

Franziska Temme, in the name of all co-authors

Point-by-point response to reviewer comments

Reviewer #1 (Remarks to the Author):

In this work, Temme and collaborators present a climatic mass balance forced by reanalysis data validated with in situ observations. Frontal ablation is also estimated. The methods are standard for this kind of research. The novelty is related to the study area, as the modelling is applied to the glaciers of the Cordillera Darwin Icefield (CDI) in southernmost South America, an area poorly studied. The main conclusion is atmospheric forcing is the main driver of the ablation in the region. The work is well-written and easy to follow and the figures are mostly clear, but I think some clarifications need to be made before publication. I hope this review helps the authors improve the manuscript.

We would like to thank you very much for the detailed and constructive review of our manuscript. In the following, you find our point-by-point list of answers to the raised comments. We are convinced that our actions have significantly improved the quality of the manuscript. We sincerely hope you find our response satisfactory, and we have been able to clarify the raised points. Referee comments are reproduced in black font color. Our response and the undertaken actions are formulated in green font color, text that was adjusted/added in the manuscript is highlighted in italic font.

I suggest a small change in the title, including “ablation” instead of “melt”. Results of the manuscript show that melt is the main component in the total ablation processes, i.e. higher percentage (74%) compared to frontal ablation (26%). Melt by definition depends on the atmosphere conditions (could be also by oceanic or lake conditions but this is not directly quantified here). If I understand, I think the title looks to remark that the climate or atmosphere drives the “ablation” as stated in lines 260-261.

Thank you for this comment. We agree and changed the word ‘melt’ to ‘ablation’.

The sentence in L275 and repeated in L347 is a bit confusing. How the mass loss is controlled by atmospheric conditions if the CMB is positive (0.02 m w.e. yr⁻¹) or close to balance (L332)? Results show climatic ablation is quite larger than frontal ablation (L275-277) and this is forced by the atmosphere, I understand this, but in terms of mass balance, the overall CMB is still positive for the period meaning that CDI has been climatically balanced in the recent two decades (L380). In consequence, overall no mass loss forced by atmosphere conditions between 2000-2022. Or do you refer to the last six years of CMB modelling? or just to land-terminating glaciers that show negative specific CMB (L145-146)? Please clarify this.

Thank you for pointing this out. We are referring to the ablation being mainly controlled by atmospheric conditions (instead of dynamic). We exchanged 'mass loss' with 'ablation' in both mentioned sentences.

No justification for the period. For instance, why 22 years and not 30 years? In the title, the word "Climate" is used, and by definition, the climate is a 30-year period (WMO). If the 22-year period is well justified, maybe change "Climate 's" to "Atmosphere 's".

See response to your comment on L107 regarding the justification of the period. We would like to stick to this terminology in the title. Although the WMO defines 30 years as the standard climate period, it is not uncommon to also use shorter periods (from 20 years) in this context. The IPCC Glossary explicitly names periods of 20 years as applicable (IPCC 2012):

"Climate in a narrow sense is usually defined as the average weather, or more rigorously, as the statistical description in terms of the mean and variability of relevant quantities over a period of time ranging from months to thousands or millions of years. The classical period for averaging these variables is 30 years, as defined by the World Meteorological Organization. (...) In various chapters in this report different averaging periods, such as a period of 20 years, are also used."

As usual in glacier modelling-approach investigations some choices need to be better justified, for instance, the use of a constant air temperature lapse rate over a spatial variable lapse rate. Also, other sources of uncertainties are not (at least) mentioned in the manuscript. For instance, no mention of debris cover areas. Is this area important to quantify melt rates at CDI?

Thank you for this comment. In the sensitivity runs, we tested the application of a zonal gradient in the temperature lapse rate (added in the 'Model calibration and validation' section). Such gradient differences were reported by Bravo et al. (2019), who found steeper lapse rates east of the Andes as compared to west. Comparison with the geodetic mass balances (for land-terminating glaciers) showed, however, no superior behavior in the calibration.

Debris-covered glacier areas are not common in the region. The Chilean Glacier Inventory (Barcaza et al., 2017) defines only around 1 % of the glacier area as debris covered. Therefore, we are convinced that debris-cover is negligible in this study.

Minor comments

L 38-39: "The large altitudinal range..." I don't understand this sentence. How do the location and the climatic conditions control the altitudinal range of the CDI? Or do you refer to the climate conditions extreme gradients? Please clarify.

Thank you for this question. We are referring to the altitudinal range of the glaciers, descending down to sea level. This is possible due to the extreme climatic conditions producing high accumulation amounts at the top. We reformulated the sentence to make the statement clearer.

“This large altitudinal range, with glaciers descending down to sea level, is possible due to the extreme climatic conditions inducing high mass input at the top.”

L59-60: Reference 17 includes a frontal reconstruction of Marinelli Glacier back to 1910.

We added the reference, thank you.

L63: I suggest including the area of the marine-terminating and the lake-terminating glaciers separately.

We included the information as suggested:

“Around half of the CDI area consists of marine- or lake-terminating (MALT) glaciers (35% and 13%, respectively).”

L107: Why this period?

We rely on 2000 (-1 year as spin-up) as start year because of data availability. Several datasets that we use are dated at ~2000. Those are the glacier inventory (Barcaza et al. 2017), the surface topography from SRTM and the geodetic mass balance (with elevation changes calculated 2000-2013). The end date is chosen as close to present day as possible (when we started working on the study). To model full mass balance years, we oriented the simulation times on the hydrological year. We have, however, extended the climate data and the COSIPY simulation to 2023 now.

L110: Any insight on why the correlation with Isla Hoste AWS is low?

We assume that you are referring to the precipitation at AWS Hoste, since the correlation for the other variables is high. Looking at the measurements again, we identify issues with the precipitation sensor starting after ~1 year of measurements. We shortened the timeseries accordingly and updated the statistics in table S2. The updated correlation is 0.72.

L114-115: Ok, but you also show individual glacier CMB results, for instance in Fig. 1. Are these values, especially the accumulation, reliable?

See response to your comment on L145-146.

L123-124: I understand what you mention here, and agree, but be aware that relative humidity is an indicator of how close to saturation the atmosphere is and depends on the air temperature.

Thank you for that comment. We reformulated the sentence to make the statement clearer:

“While air temperature at sea level exhibits no clear west-east gradient, annual average relative humidity shows a drying towards the east (Fig. S3b).”

L145-146: Similar to a previous comment, how reliable is the CMB estimation for individual glaciers? I agree that the regional CMB is a good approach for CDI considering the uncertainties, lack of measurements and differences in the accumulation rate reported in L114 but, along the manuscript, CMB of individual glaciers is used for comparisons.

Thank you for this question. Admittedly, we do report CMB values of individual glacier catchments but we do so including the specific CMB uncertainty, which is quantified in an objective and systematic way. With this uncertainty assessment, we think it is warranted to forward values for individual glacier catchments. Let me explain the uncertainty assessment:

We are comparing the glacier-wide integrated mass balance (specific mass balance) for each land-terminating glacier (> 3 km²). As land-terminating glaciers have no dynamically controlled losses, the two datasets are directly comparable. The glacier-specific geodetic information has not been used for model calibration (only the regional average has been used), and is, thus, an independent validation dataset. We compare the (specific) geodetic and climatic mass balance per glacier catchment and calculate the root mean square error (RMSE ± 0.62 m w.e. yr⁻¹). This error presents the accuracy/uncertainty of the climatic mass balance on a glacier-specific basis. We reformulated the ‘Model calibration and validation’ section to increase clarity (former L553-571), especially the following sentence:

“Model performance is quantified by the root mean square error between the glacier-specific climatic and geodetic mass balance for these glaciers, resulting in a model error of ± 0.62 m w.e. yr⁻¹.”

L152-153: I suggest adding a Figure (could be in Supplementary Material) to explore the differences in the CMB between land and marine-lacustrine glaciers.

Thank you for this comment. The difference can be seen in Fig. 1, where both the CMB and the termination type of the glaciers are included. We, however, agree that the information is hard to extract from this figure. Therefore, we added a figure with the average annual CMB for each glacier, color-coding the termination type, to the supplement (Fig. S11).

L176: Change Juli by July.

Changed.

L297-308: Why present ELA results in the Discussion section and not in the Results sections?

Thank you for this comment. At this point, we compare the simulated ELA results with results from previous studies, which we want to keep in the Discussion section. However, we moved the first two sentences (L296-298), where we only present results, to the results section.

L317-321: Any clue for this behaviour? Southern Annular Mode or another mode of interannual or interdecadal variability?

Thank you for this comment. The Southern Annular Mode is claimed to be the main mode of climate variability over southern Patagonia. We tested correlations between the SAM index and the climatic mass balance, snowfall and melt energy. None of these variables indicates a relation between SAM and the mass/energy balance of the CDI. We clearly see that the more positive phase around 2009 - 2015 is accompanied by higher snowfall amounts and less surface melt compared to the years before and after (Fig. S10a). Climate data shows slightly lower air temperatures and years with increased precipitation amounts. However, we do not have an explanation for this variability at the moment. It would certainly be an interesting future study!

We added the following sentences:

“Those more positive or negative periods are strongly linked to snowfall amounts and surface melt (Fig. S10a). A distinct relation to the Southern Annular Mode, which is proposed as the main mode of climate variability in Tierra del Fuego³³, cannot reliably be confirmed from our results. This finding is in agreement with a recent study focused on the Patagonian Icefields⁴⁴.”

L315-328: Missing references about glacier energy balance in Patagonian glaciers that could be useful to add to the discussion.

Thank you for this comment. We already give several references about glacier energy balance of Patagonian glaciers in this section (Temme et al. 2023, Weidemann et al. 2020, Weidemann et al. 2018, Schaefer et al. 2020, Minowa et al. 2023, Schneider et al. 2007). We now added the publication of Bravo et al. (2022). These are all publications addressing the energy balance of Patagonian glaciers that we are aware of. If you know other essential studies that we overlooked for this region, we would ask you to specify those so that we can include them. Thank you!

L329: Also missing references about surface melt trends and projections in Patagonian glaciers. I suggest comparing your findings with previous works at regional scale.

Considering the uncertainties that are mentioned in the paper I suggest keeping a regional point of view for comparisons. Some comparison by glacier is fine, but the approach here is regional.

Thank you for this comment. We extended the discussion on surface mass balance and melt trends and projections for Patagonia:

“A trend analysis over the study period clearly demonstrates an extremely likely increase in surface melt, which is in agreement with simulations of surface melt for the Northern and Southern Patagonian Icefield estimating a positive trend of +0.30 m w.e. yr⁻¹ per decade (1975-2005)³⁷. This rise in surface melt is the main driver of a likely trend towards more negative mass balance in the CDI. Congruently, Bravo et al.³⁷ present negative trends in the surface mass balance of the Northern and Southern Patagonian Icefield (1975-2005). However, contrary results are presented in ref. ^{45,43,46}. Despite the likely positive trend in precipitation amounts in the CDI, we do not find an increase in snowfall amounts. In general, glaciers in the CDI that are covering a lower elevation range (< 600 m) already experience more pronounced mass losses with an average CMB trend of -0.43 m w.e. yr⁻¹ per decade (extremely likely). At Perito Moreno Glacier some 400 km north of the CDI, Minowa et al.⁴¹ found a virtually certain decreasing trend in the surface mass balance over the ablation zone of -0.90 ±0.3 m w.e. yr⁻¹ per decade (1996-2020). This decreasing trend is in general agreement with our findings for areas at similar altitudes in the CDI (-0.50 m w.e. yr⁻¹ per decade). If the recent trends continue, the CDI will get further out of balance and glaciers will trace a path of accelerating mass loss as already indicated by the current trend in the CMB (Fig. 3c). Projections for the Northern and Southern Patagonian Icefield predict just such a pathway of enhanced surface melt causing decreasing surface mass balance until 2050³⁷.”

L380-381: This sentence seems more appropriate for the climate or atmosphere forcing on mass balance.

Thank you.

L446: Is there any reference for this velocity threshold?

Thank you for this question. In order to enable a direct comparison between the specific geodetic and climatic mass balance, we need to identify all glaciers with no (or negligible) frontal ablation. The Chilean Glacier Inventory (Barcaza et al. 2017), which is used in this study, includes a classification of the termination type, but has a large share of lake- and marine-terminating glaciers. Especially for the lake-terminating glaciers, we (visually) identified many cases where frontal ablation is negligible due to a small calving front and low ice velocity.

Therefore, we had to revise the classification for our purposes. The goal was to find an objective classification to distinguish those glaciers with significant frontal losses from those with no or negligible frontal losses. Ice velocity is suitable because glaciers with

negligible ice flow at the front also have negligible calving. The classification is a compromise between on the one hand excluding glaciers from the calibration/validation dataset with significant calving losses to avoid biased results, and on the other hand guaranteeing a calibration/validation dataset that is large enough. With our approach, we achieve a calibration/validation dataset that covers ~37% of the entire CDI.

To guarantee that glaciers that had been defined as land terminating in the Chilean Inventory will keep this classification, the threshold velocity of 60 m yr^{-1} (there was a typo in the text) was suitable.

A rough assessment of the calving fluxes of newly defined land-terminating glaciers (=glaciers terminating in lakes or fjords but defined as land-terminating in our study) shows that the specific calving losses are mostly $< 0.1 \text{ m w.e. yr}^{-1}$.

Reviewer #2 (Remarks to the Author):

The study models the surface mass balance of all glaciers in the Cordillera Darwin Icefield (CDI) in Tierra del Fuego from 2000 to 2019, utilizing the open-source surface energy balance model COSIPY (Sauter et al., 2020), which is forced by downscaled ERA5 data. The model parameters are primarily calibrated with in-situ mass balance observations from one or a few glaciers in the region. Notably, this calibration process is based on the authors' previously published work (Temme et al., 2023). The difference between the geodetic mass balance and the simulated surface mass balance for lake- and ocean-terminating glaciers is used to estimate mass loss due to frontal ablation.

Overall, the paper is well-written, but there are areas where additional methodological detail would enhance clarity (elaborated further in my specific comments). The authors highlight the novelty of the study as "the first simulation of the climatic energy and mass balance of glaciers in this region." However, given that the same first author has previously applied the COSIPY model (developed by Sauter et al., 2020) to a subset of glaciers in the region (Temme et al., 2023), this work may be better framed as an extension of that earlier study, expanding the model's application to a broader set of glaciers.

The method used to estimate frontal ablation raises some concerns, particularly regarding the quantification of uncertainties (detailed in my specific comments). A more comprehensive uncertainty analysis would strengthen the credibility of the results, particularly in relation to frontal ablation estimates.

In summary, while the study provides valuable insights into the region, the novelty of the contribution could be more clearly articulated. Additionally, addressing some critical methodological elements, such as a more rigorous uncertainty analysis, would significantly enhance the robustness of the results.

We would like to thank you very much for the detailed and constructive review of our manuscript. In the following, you find our point-by-point list of answers to the raised comments. We are convinced that our actions have significantly improved the quality of the manuscript. We sincerely hope you find our response satisfactory, and we have been able to overcome your methodological concerns. Referee comments are reproduced in black font color. Our response and the undertaken actions are formulated in green font color, text that was adjusted/added in the manuscript is highlighted in italic font.

Specific comments:

Abstract: The authors should include the period of the simulations in the abstract. It

took me some time to determine that the simulations cover 2000–2019, and having this information up front would clarify the scope of the study.

Thank you for this comment. We added the simulation period (2000-2023).

Line 77-78: "...melt, which is strongly linked to air temperature"

One of the references cited here used a positive-degree-day (PDD) model to simulate melt, so linking melt to temperature is an inherent assumption of the model.

Thank you for this comment. Based on another reviewer's comment, we removed this sentence. It is giving too much detail for the Introduction and those points are better being discussed in the Discussion section. For the exact reformulation, see next comment.

Line 78-79: "Despite these local efforts to determine the mass balance of individual glaciers, a systematic approach for a CDI-wide estimate of the CMB is to this day missing."

However, Temme et al. (2023) (reference #20) addresses not just individual glacier modeling but also regional-scale modeling. The authors may want to reconsider this statement, as it somewhat underrepresents the existing work in this area.

Thank you for this comment. We agree that the formulation was misleading. We revised the entire introduction based on another reviewer's comments. The sentence was reformulated to:

"The climatic mass balance (CMB) has been studied in the Cordon Martial (east of the study region, located in Argentina)¹⁹ and in the Mount Sarmiento Massif^{20,21} (western CDI), but these local efforts are insufficient and a systematic CDI-wide estimate of the CMB is needed."

Line 80: "This lack of knowledge can be addressed with a well-informed modelling product of CMB."

The term "well-informed modeling product of CMB" is unclear. It would improve clarity to specify what is meant by this term and what type of model product the authors are referring to.

Thank you for this comment. The statement was removed in the course of the revision of the introduction section based on another reviewer's comments.

Line 82-83: "Thus, with the climatic loss term available, the mass budget of the CDI glaciers can successfully be closed and give the dynamic loss as residual."

There seems to be a disconnect between this statement and the preceding one. The total mass loss is derived from geodetic estimates, not the climatic loss term. I suggest rephrasing this to clarify that the dynamic loss is inferred from the residual of the modeled climatic mass balance (CMB) and geodetic mass balance. Additionally, errors

in both the CMB and geodetic estimates could be significant, making it challenging to assume that the residual solely represents dynamic loss.

We added a more detailed explanation in the text, saying:

“Thus, with the climatic mass balance term being quantified, we assume that the residual with respect to the geodetic mass budget is primarily explained by frontal ablation – an ice-dynamically controlled loss term.”

Line 94: “In this way, an unprecedented attribution of the observed mass loss to climatic and ice-dynamic forcing becomes possible.”

While this is an important step forward, the authors should also account for the uncertainty in the modeled CMB and geodetic estimates. Without addressing these uncertainties, the attribution may not be as straightforward as suggested here. For instance, Hugonnet et al. (2021) point out that the accuracy of geodetic mass balance estimates can decline when moving from multi-annual to annual timescales.

We refined and added more details on the uncertainty analysis of the frontal ablation in the Data & Methods section on ‘Frontal ablation’. See also response to your comment on L577-578.

Line 96-97: “The energy and mass balance will be examined using the physically-based COupled Snowpack and Ice surface energy and mass balance model in PYthon (COSIPY).”

It would be helpful to explain why COSIPY was chosen for this study. Temme et al. (2023) compared COSIPY with simpler models, such as PDD approaches, and noted that some of the model parameters, like those in the albedo parameterization, still require calibration with mass balance observations. If the model is calibrated with geodetic mass balance, how can it provide an independent estimate of the ice-dynamic component? If the in-situ mass balance data from one or a few glaciers is used for calibration, how do the authors address the potential issue of limited parameter transferability across the CDI?

Thank you for these questions.

For model calibration we use the region-wide average geodetic mass balance of all land-terminating glaciers ($> 3\text{km}^2$) ($= -0.27\text{ m w.e./yr}$). By taking the land-terminating glaciers only, the dynamic component is zero and the climatic and geodetic mass balance are directly comparable. Temme et al. (2023) conclude that calibrating against regional geodetic observations is the best strategy to achieve realistic regional simulations of climatic/surface mass balance, and to address the limited parameter transferability. We follow this strategy, and we achieve a good agreement between the geodetic (-0.27 m w.e./yr) and climatic mass balance (-0.23 m w.e./yr), confirming the applicability on the CDI.

For validation of the climatic mass balance, we again stick to the land-terminating glaciers, preventing a contribution of dynamic losses. However, now we compare the (specific) geodetic and climatic mass balance per glacier catchment (instead of the regional average taken for calibration), and calculate the root mean square error (RMSE ± 0.62 m w.e. yr⁻¹). This error presents the accuracy/uncertainty of the climatic mass balance on a glacier-specific basis. In-situ ablation observations from Schiaparelli and Martial Este Glacier are used for additional validation information.

We chose the COSIPY model since its application has been proven to be successful in the area before (Temme et al. 2023) and we focus not only on the climatic mass balance but also on the energy balance. Less complex models (positive-degree-day or simplified-energy-balance model) would not have enabled a detailed analysis of the CDI glaciers' energy balance.

Line 422: "...since comparison with the Randolph Glacier Inventory (V6) has revealed strong weaknesses of the latter for the southern Andes."

The term "weaknesses" seems too vague and possibly unsuitable here. It would be more informative to specify what shortcomings of the Randolph Glacier Inventory were identified in the region. Does it poorly represent glacier outlines in this area?

Thank you. We reformulated the sentence for more clarity:

"Glacier outlines are taken from the DGA glacier inventory^{55,56} since the Randolph Glacier Inventory (V6) has revealed poor representation of outlines for the southern Andes, especially for smaller glaciers⁵⁸."

Line 428-430: "Since elevation changes cover the period 2000 to 2013, we manually produced outlines for 2013, which have been missing so far. Thus, inventories available in this study cover years 2000, 2013 and 2019 oriented on inventory availability and geodetic data."

The phrase "oriented on inventory availability" is unclear. I suggest rewording for clarity. Also, more detail is needed on how the 2013 outlines were manually produced, as this is a crucial methodological step.

Thank you for this comment. The outlines for 2013 were produced using satellite images from late summer 2013 and 2014, and manually adjusting the outlines to the 2013 glacier extent. Satellite images used are Landsat 7 and 8 (March 2013) as well as ASTER (Advanced Spaceborne Thermal Emission and Reflection Radiometer) (January to March 2013 and 2014).

We reformulated the section to: *"Since elevation changes cover the period 2000 to 2013, we manually produced outlines for 2013. Therefore, we used late-summer images from Landsat 7 and 8 (2013) and ASTER (Advanced Spaceborne Thermal Emission and Reflection Radiometer) (2013 and 2014) and manually adjusted the outlines to the*

current glacier extent. Thus, inventories available in this study cover years 2000, 2013 and 2019.”

Line 431-441: What is the final output of this method? Is it glacier-wide geodetic mass changes over the entire period? Please clarify the time period and spatial scale covered by these mass change estimates.

The details on the processing and output of the method are given in (former) L533-552, while in this section only the data itself is described. To highlight the time period more clearly, we added it in (former) L 552: *“The mean regional observation period of the elevation change rate measurement is 12.97 years (2000-2013).”*

Line 464: Why was a radiation model used instead of directly employing incoming solar radiation from ERA5? Have you evaluated ERA5’s radiation data for this region, and if so, why was it deemed unsuitable for glacier-scale modeling? Additionally, if you use cloudiness data from ERA5 for the radiation model, how reliable is this parameter, given that ERA5 cloud data may be less accurate than its radiation estimates?

Thank you for this question. The coarse resolution of ERA5 data does not resolve the local topography, and is thus not directly applicable for our high-resolution (200x200m) study. A topographic correction would have been required. The applied radiation model runs on the target resolution and, thus, the topography is well resolved, incorporating corrections for slope and aspect of the topography as well as topographic shading.

In our study, we follow in general the approach and setup of Temme et al. (2023) in the Mount Sarmiento Massif, where COSIPY was carefully calibrated. In this study, the same radiation model was employed. To ensure transferability, the methodology must remain consistent.

The performance of the modelled radiation is validated with available AWS measurements (Table S2).

Line 481: *“...which is obtained by removing the orographic component from the ERA5 precipitation.”*

Does ERA5 provide an orographic component of precipitation? Based on Hersbach et al. (2020), ERA5 supplies large-scale and convective precipitation, but not explicitly an orographic component. Please clarify this point, as convective precipitation is not synonymous with orographic precipitation.

Thank you for this question. The orographic component of ERA5 is calculated by running the orographic precipitation model on the ERA5 elevation model. This ERA5 orographic precipitation is subtracted from the ERA5 total precipitation, giving the large-scale precipitation without an orographic component. We reformulated the section

accordingly:

“The total precipitation is calculated by adding the orographic precipitation calculated in the model to the large-scale precipitation. The large-scale precipitation is obtained by removing the orographic component, calculated by running the orographic precipitation model on the ERA5 topography, from the ERA5 total precipitation.”

Line 523: “Albedo values are differentiated between snow, firn and ice surfaces. The decay of surface albedo due to snow aging is parameterized following the scheme of Oerlemans and Knap.”

Given the critical role of albedo in surface energy balance (SEB) modeling, have you evaluated the modeled albedo against observations from your study area? If in-situ albedo data is unavailable, MODIS-derived albedo could be a useful comparison.

Without any evaluation, it’s unclear how reliable the albedo model is for this region, and its validation elsewhere (e.g., in the European Alps) may not be sufficient due to spatial variability in albedo characteristics.

Thank you for this comment. Unfortunately, there are no in-situ observations of albedo available in the Cordillera Darwin to our knowledge. The albedo scheme of Oerlemans and Knap was applied and validated, however, at the Southern Patagonian Icefield based on satellite observations (Bravo et al. 2022).

Following your suggestion, we extracted daily values of surface albedo of the MODIS/Terra Snow Cover Daily L3 Global 500m SIN Grid, Version 61 dataset (Hall and Riggs 2021) for the period 2000-2022. We use the "Snow_Albedo_Daily_Tile" band (<https://nsidc.org/data/mod10a1/versions/61#anchor-documentation>) provided by the National Snow and Ice Data Center (NSIDC). As anticipated, the spatial and temporal coverage over the CDI is very low due to persistent cloud coverage of the majority of glacierized areas. Using MODIS data directly as an input to the model would thus not be feasible. Instead, we decided to derive seasonal-average-tiles for winter (JJA) and summer (DJF) (see figures below). Also in those fields, we observe incomplete coverage and many pixels with few observations. Furthermore, we do not see a clear elevation dependent albedo as we would have expected it. At several points, neighboring pixels show extreme differences.

Even in winter (JJA), when snowfall is extremely frequent, many pixels show albedo values around 40-50% in the central plateau (>1500 m a.s.l.), as can be seen for the example year 2020 below (the year with the best coverage). A possible explanation for these inconsistencies could be a misclassification of pixels, indicating clear sky conditions while clouds are still present over the ice surface. We strongly doubt these values.

In summer (DJF), the largest part of the CDI has no albedo observations which we relate to cloud coverage and/or misclassifications of the albedo product over the snow-covered areas (see example from year 2011/12 below).

Due to these complications, we unfortunately don't see a way to make use of the MODIS-derived albedo for our study site.

The albedo scheme that we apply over the CDI was part of the model calibration in the Mount Sarmiento Massif in the western part of the CDI (Temme et al. 2023). Thus, the feasibility of the ice and firn albedo is given.

Line 526: “correction using the Richardson-Number or the Monin-Obukhov similarity theory.”

Which approach was ultimately used? Further, how were roughness lengths determined? Have you compared your modeled turbulent heat fluxes with measurements from glacier sites in your region to evaluate performance?

Thank you for these questions. The model set-up is summarized in Table S5 (see former L 529). Roughness lengths were taken from Temme et al. 2023, where the ice roughness length was calibrated in the Mount Sarmiento Massif (western CDI).

Unfortunately, there are no measurements of turbulent heat fluxes available in the

Cordillera Darwin. A direct comparison is, thus, not possible.

Line 529: “was positively evaluated”

This phrase is unclear. Could you elaborate on what is meant by “positively evaluated” in this context?

Thank you for the question. The model evaluation and comparison with three less complex surface mass balance models showed an overall good performance of COSIPY. Despite being more challenging to calibrate, results of COSIPY agreed well with the other models and the observations of glacier-specific geodetic mass balance and ablation stakes used for validation (see former L531-532). Accordingly, Temme et al. (2023) show that the COSIPY model can be applied in the Monte Sarmiento Massif with similar performance to less complex models.

Line 532: “COSIPY results agreed well with the other models as well as with observations of ablation stakes and geodetic mass balance.”

While this is a useful finding, it would be important to specify which parameters in COSIPY were calibrated and how this calibration was performed. As noted earlier, calibration with geodetic mass balance or in-situ data has implications for the interpretation of the results and the separation of the climatic and dynamic components of mass balance.

Thank you for the question. The parameters calibrated in COSIPY in Temme et al. 2023 are listed in Table 1 of that paper: albedo of ice, albedo of firn, and roughness length of ice. In that paper, also the calibration strategy is explained in more detail.

In short, climatic mass balance was calibrated against the region-wide average geodetic mass balance of the land-terminating glaciers (same approach as for this study). Taking only the land-terminating glaciers, the dynamic component is zero and the climatic and geodetic mass balance are directly comparable.

In this study, a full calibration is not feasible due to the size of the domain. Instead, sensitivity runs are used for optimization of the climatic input data (see next comment).

Line 559: “Sensitivity runs are applied for further optimization.”

Please clarify what parameters or inputs were varied in the sensitivity runs. Were different datasets or methods used for calibration, or were the model parameters adjusted? Additionally, it is crucial to provide a breakdown of the uncertainties that arise from your modeling approach.

Thank you for this question. COSIPY model parameters are taken from the Mount Sarmiento Massif. In the 5 sensitivity runs, we varied the atmospheric input fields to address the challenge for extrapolating from the local (AWS) scale to the regional scale. The gradients across the cordillera are strong, requiring the testing of different interpolation methods. Tests include variations in the interpolation of the relative

humidity and wind velocity fields as well as the application of a gradient in the temperature lapse rate (based on Bravo et al. 2019, suggesting steeper gradients east of the Andes compared to west). The target is the region-wide average geodetic mass balance for the land terminating glaciers (= -0.27 m w.e./yr) that was met well by the chosen, presented run (-0.23 m w.e./yr). The sensitivity runs covered a spread between +0.51 and -1.58 m w.e./yr. We added this information into the text:

“Sensitivity runs are applied for further optimization, where the methods for the generation of the atmospheric input fields (relative humidity, wind velocity, air temperature) are varied, addressing the challenge to realistically reproduce the zonal climatic gradients.”

The uncertainty of the modelled climatic mass balance is calculated in the model validation based on glacier specific geodetic mass balances. See next comment for more detail.

“Model performance is quantified by the root mean square error between the glacier-specific climatic and geodetic mass balance for the individual catchments.”

How do you address the fact that geodetic mass balance includes both the climatic and ice-flow components? Are you comparing glacier-wide mass balance only? Please provide more details.

Thank you for this question. We are comparing the glacier-wide integrated mass balance (called specific mass balance) for each land-terminating glacier. As land-terminating glaciers have no ice-dynamically controlled losses, the two datasets are directly comparable. The glacier-specific geodetic information has not been used for model calibration (targeting the regional average), and is, thus, an independent validation dataset.

The information that we are only using land-terminating glaciers without frontal losses is stated several times in this section (former L562, 564, 565-566). We reformulated the following sentences to increase clarity (former L560-562 & former L568-570):

“Following this approach, we rank the sensitivity runs based on the highest agreement with regional geodetic mass balance, as observed with satellite remote sensing, for all glaciers with no frontal ablation (region-wide average -0.27 m w.e. yr⁻¹). The highest ranked run (region-wide average -0.23 m w.e. yr⁻¹) agrees well with the geodetic observations and is presented in this study.”

“Model performance is quantified by the root mean square error between the glacier-specific climatic and geodetic mass balance for these glaciers, resulting in a model error of ± 0.62 m w.e. yr⁻¹.”

Line 577-578: “Uncertainties of the glacier-specific CMB, constrained by model validation, directly translate into the uncertainties of frontal ablation estimations.”

More details are needed on how the uncertainties in glacier-specific CMB are quantified. Furthermore, uncertainties in geodetic mass balance estimates should also be considered. Ideally, validation of frontal ablation estimates should be conducted on a study glacier where in-situ measurements of frontal ablation, ice-flow, and surface mass balance are available.

Thank you for this comment. The estimation of uncertainties in the glacier-specific CMB has been explained in the comment above. To consider the uncertainties in the specific geodetic mass balance more explicitly, we now add the geodetic mass balance error to the uncertainty analysis. The total uncertainty adds the uncertainty in the specific climatic mass balance and the uncertainty in the specific geodetic mass balance. This way, we overestimate the inherent uncertainty because random errors in the geodetic mass balance may be counted twice: 1) In the geodetic mass balance error itself and 2) in the climatic mass balance error as it is inferred from the RMSE with the glacier-specific geodetic values. We adjusted the section accordingly:

“Uncertainties in frontal ablation consist of the uncertainties in the glacier-specific CMB (see Model calibration and validation) and the uncertainties in the specific geodetic mass balance (see Geodetic mass balance processing) following classical Gaussian error propagation. Since the former uncertainty already includes random errors in the geodetic mass balance, the resulting uncertainties of frontal ablation values are likely an overestimation.”

The only glacier with in-situ observations of ice thickness, ablation and surface velocities is Schiaparelli Glacier. We can calculate the discharge through a flux gate (D_{FG}) at the location of the ice thickness measurement (Gacitúa et al. 2021). Together with the climatic mass balance above this flux gate (CMB_{FG}), this value should match the geodetic mass balance (GMB_{FG}) above this flux gate:

$$GMB_{FG} = CMB_{FG} + D_{FG}$$

The results agree within their uncertainty ranges:

$$GMB_{FG} = -0.014 \pm 0.004 \text{ Gt/yr}$$

$$CMB_{FG} + D_{FG} = -0.024 \pm 0.020 \text{ Gt/yr}$$

Line 582-583: “For our study site, the lack of ice thickness measurements in the CDI makes ice thickness highly speculative. This directly translates into elevated uncertainties for frontal ablation.”

Could you provide more details on how these uncertainties due to speculative ice thickness were quantified? It would strengthen the paper to explain the methodology or assumptions used to assess these elevated uncertainties.

Thank you for this question. With this statement we want to highlight the elevated uncertainty in ice thickness reconstructions in the CDI compared to other regions due to

the lack of observations. There is no possibility to validate those fields with in-situ observations, and no in-situ observations of ice thickness are included in the modelling.

To quantify these uncertainties more thoroughly, we compare the available ice thickness products from modelling studies and calculate the standard deviation per pixel. While locally the deviations can be extremely high (up to 250 m), the standard deviation along the flux gates is smaller. The uncertainty in frontal ablation following this new approach increases for some glaciers while it decreases for others (see new Figure 4). We adjusted the section on the flux gate approach accordingly:

“For our study site, the lack of ice thickness measurements in the CDI makes modelled ice thickness uncertain. To quantify the uncertainty in the ice thickness, we calculate the standard deviation of the available ice thickness products from Carrivick et al.⁶⁴, Millan et al.³ and the participants in the consensus estimate covering the CDI². Time discrepancies are corrected for by satellite-derived elevations changes.”

Results section:

Since you are using a surface energy balance (SEB) model for ablation, it would be informative to estimate the individual contributions of key SEB components (incoming shortwave and longwave radiation, albedo, sensible and latent heat fluxes) to the total annual melt. While some information is provided, it would be beneficial to present a more systematic breakdown of how each term changes over the study period, both temporally and spatially. In particular, further discussion on albedo and incoming longwave radiation beyond the estimates of trends would provide valuable insights.

Thank you for the comment. Most of the relevant energy and mass fluxes and their contributions are provided in Fig. 3 and Fig. S6-S9. We added a figure that gives the annual values of those variables over the study period in the supplement (Fig. S10). Incoming and outgoing short- and longwave radiation only show minor annual variability, which is why we excluded those variables here for visibility reasons.

Line 204: “Our results show that the annual CMB (Fig. 3c) is mainly controlled by air temperature (Fig. S3a).”

The word “controlled” may not be the best choice here. Temperature influences the CMB model primarily through the bulk method for sensible heat fluxes, which you indicate contribute around 30% to melt. Perhaps what you mean to say is that the inter-annual variability in CMB is most strongly correlated with temperature. Additionally, have you considered correlating CMB with albedo? Several studies (e.g., Davaze et al., 2018; Williamson et al., 2020; Xiao et al., 2022) demonstrate that mass balance is often strongly correlated with summer albedo.

Thank you for this comment. We revised the sentence according to your suggestions:
“Our results show that the inter-annual variability of the CMB (Fig. 3c) is mainly influenced by air temperature ...”

Furthermore, we tested the correlation between albedo and CMB. Indeed, there is a significant correlation in summer (DJF), but also during the entire year. We included this information into the manuscript.

“Melt energy and CMB are also linked to the annual average incoming solar radiation ($r = 0.54$ and $r = -0.46$, respectively) as well as the albedo ($r = -0.68$ and $r = 0.69$, respectively).”

Line 230-236:

It was noted earlier that the flux-gate approach carries large uncertainties due to the lack of ice-thickness estimates, yet the reported errors for CDI frontal ablation using this approach are smaller than those from the method based on the difference between geodetic and CMB estimates. How is the total error from the flux-gate approach assessed? It would be helpful to include more detail on this in the Methods section. Additionally, how are the error estimates for frontal ablation of individual glaciers calculated?

See reply to the comments on the Data & Methods section above. Numbers in this section, the discussion and the abstract were adjusted following the revised uncertainty estimations.

Table S5:

While you've listed the roughness length for momentum, it would also be useful to include the roughness lengths for temperature and humidity, as these are also important in the bulk method used in your SEB model. Additionally, how were the roughness lengths determined? The roughness length of momentum for ice seems quite small (on the order of 10^{-4} m). Typically, roughness lengths for ice are closer to 10^{-3} m (see Hock, 2005 for reference). Could you clarify how this value was chosen and whether it might be underestimated?

Thank you for these questions. The roughness length of ice was taken from Temme et al. (2023) where it was one of the calibration parameters, as given in Table S5. An underestimation is not impossible, however, larger values of ice roughness length produced larger deviation from the calibration targets (regional geodetic mass balance) and validation datasets (specific geodetic mass balance and ablation stake measurements) in that study (see Temme et al. 2023, Supplement).

The formulation of roughness lengths for temperature and humidity in COSIPY follows the renewal theory for turbulent flow. Thus, the roughness lengths for humidity and temperature are assumed to be one and two orders of magnitude smaller than the roughness length for momentum, respectively (Sauter et al. 2020).

Reviewer #3 (Remarks to the Author):

This study produces the first climatic mass balance (CMB) simulation over the Cordillera Darwin Icefield (CDI), one of the largest ice bodies in the Southern Hemisphere. These outputs are differenced from satellite-derived elevation change measurement to calculate the mass loss due to iceberg calving and subaqueous melting (collectively termed frontal ablation) for the first time. Results show that there is a strong climatic gradient associated with westerly winds across the CDI which definitively influences glacier mass balance. Frontal ablation accounts for 26% of the total mass loss in the CDI. Over the 22-year study period, surface melt has increased (+0.18 m w.e. yr⁻¹ per decade). While frontal ablation is important for predicting the evolution of specific glaciers, atmospheric conditions are the main control on glacier evolution.

The study results and conclusions are valid, robust, and represent an important contribution to the field of glaciology. The manuscript is well written, and the figures and tables are formatted properly and of excellent quality. The suggested changes are minor and include drawing conclusions regarding the contrasting glacier behavior in the CDI within the discussion, applying a density conversion of 850 +/- 60 kgm⁻³ to convert the geodetic change to mass change, and modifying the introduction to improve readability and clarity.

Kind regards,
Nicole Schaffer, PhD.

We would like to thank you very much for the detailed and constructive review of our manuscript. In the following, you find our point-by-point list of answers to the raised comments. We are convinced that our actions have significantly improved the quality of the manuscript. We sincerely hope you find our response satisfactory. Referee comments are reproduced in black font color. Our response and the undertaken actions are formulated in green font color, text that was adjusted/added in the manuscript is highlighted in italic font.

General comments

Validity: The results and conclusions of this study are valid and robust.

Significance: Given that the CDI is one of the largest ice bodies in the Southern Hemisphere and contributes ~5% of the total loss in South America (Braun et al., 2019), the results are quite important for the field of glaciology and climate change science. The CMB data set produced provides a second method to estimate mass loss that complements the satellite-derived elevation change measurements calculated here and previously published. Together these provide a more robust estimate of mass loss

for the CDI, especially considering they are in close agreement. Confirming that climatic variables exert the greatest influence on glacier evolution, insight into how these variables are distributed over the CDI, and knowledge of how much loss is attributed to frontal ablation and how this varies through time helps tremendously for modeling the evolution of these glaciers in the future.

Data and methodology: The approach is valid, reproducible, and meets the standards presented in published work within highly respected journals. The data quality is good and the figures and tables provided both in the main text and SI support the research and illustrate the results very effectively. In general, the manuscript is well written.

Analytical approach: The analysis is robust and the statistical results presented are appropriate.

Clarity and context: The manuscript text is clear and accessible. Sufficient context has been provided with the appropriate references and the results have been placed in the context of previous literature. The only section that needs some work in terms of the readability and clarity is the introduction (see comments in the “suggested improvements” section).

References: Yes, the manuscript references previous literature appropriately.

Your expertise: With respect to the CMB I have published using more simplistic models (e.g. enhanced temperature-index model) and have a strong background in glaciology and mass balance modelling theory (MSc and PhD in glaciology), but have not published first-author articles using a full energy balance model. With respect to elevation change measurements, I have a strong background in this area with publications on the subject but have not processed interferometry data sets myself. I have published articles including calculations for frontal ablation.

Suggested improvements:

A sub-objective of this manuscript is to gain insight and draw conclusions on the contrasting glacier behavior in the last two decades. This is addressed directly and possibly indirectly in several locations within the results and discussion sections. For example, L262-274 where the four advancing glaciers are discussed in detail. L151-160 where the diversity is related to the hypsometry, aspects and MALT versus land-termination of the respective glaciers. In the results/discussion the authors discuss how more precipitation in the south-western part of the CDI result in more positive mass balance here and more losses in the north-east (e.g. L293-296). The impact of the ELA on mass balance is also discussed (e.g. L297-308). It would be good to bring together all the above points and any other highly relevant results/information within the discussion

section to draw some conclusions on why we observe contrasts in mass balance over the CDI. Particularly for those glaciers with a positive mass balance. Some or all of the text on L151-161 and L262-274 could be deleted from the results and moved to the discussion. If relevant, incorporate the four glaciers dominated by frontal ablation into this discussion. It would also be good to incorporate a very brief discussion on why MALT glaciers show on average a more positive CMB than land-terminating ones (1-2 sentences).

Thank you for this comment. We added a short paragraph about the variability in glacier-specific CMB and the characteristics in the Discussion section. The information about the difference in CMB between MALT and land-terminating glaciers was moved here from the results section. The reason for the difference lies in the glacier hypsometry. Land-terminating glaciers cover on average lower elevations and have little glacier area above 1200m, while MALT glaciers have a more even area distribution between 500 and 1500m.

“Glacier-specific CMB across the CDI is highly variable (Fig. 1c) and depends strongly on glacier hypsometry. MALT glaciers show on average a more positive CMB than land-terminating glaciers as their catchment area extends generally to higher altitudes (above 1200 m). In contrast to the overall mass loss and retreat of glaciers in the CDI, few glaciers have advanced during the first two decades of the 21st century. An explanation for this contrasting behavior may lie in the exposition and hypsometry of the glaciers. High-elevation accumulation areas and steep topography result in high snowfall amounts and above-average accumulation-area-ratios. Southward exposures of Garibaldi, Guilcher Oeste and Guilcher Este glaciers further reduce surface ablation. Glaciers with high accumulation-area-ratio are often less sensitive to changes in ELA and, thus, warming due to the small and steep ablation area¹⁵.”

A density of 900 +/- 60 kgm⁻³ has been used to convert geodetic change to mass change in this manuscript. This is the standard approach. However, it is well known that the “density” for geodetic volume-to-mass conversion is not constant and is systematically lower than ice density in most cases (Huss et al. 2013). This is due to the accretion/removal of low-density firn layers and changes in the firn density profile with positive/negative mass balance. A value of 850 +/- 60 kgm⁻³ is appropriate for most situations. Therefore, I would suggest applying a density of 850 +/- 60 kgm⁻³ in addition to 900 +/- 60 kgm⁻³. This could be included within the discussion section. New calculations for the dynamic component should be included and any changes to your main observations/conclusions should be discussed.

Thank you for this comment. We decided to follow Braun et al. (2019) who suggest a density conversion of 900 ±60 kg m⁻³ for Patagonian glaciers where melt of ice plays the major role. We tested the application of a density of 850 ±60 kg m⁻³, and results did not change significantly. The presented simulation stays the best-ranked run in the

sensitivity tests and the calculated model error (RMSE) stays unchanged at $\pm 0.62 \text{ m w.e. yr}^{-1}$.

We could provide the frontal ablation results for the second density-scenario, however, it is redundant in our opinion. Reducing the density from 900 kg m^{-3} to 850 kg m^{-3} results in a decrease in geodetic mass loss by $\sim 5\%$. This reduction directly translates into the frontal ablation calculated with the mass budgeting approach. The frontal ablation will, thus, also reduce by $\sim 5\%$. In our opinion, the gain of information by including the second density scenario is limited. We added the following paragraph in the Discussion:

“We apply a density conversion factor of 900 kg m^{-3} for volume-to-mass conversion, following Braun et al.⁹. A lower conversion factor of 850 kg m^{-3} ⁵² does not change the CMB model performance metric significantly and transmits linearly into the frontal ablation estimates. This is covered by the uncertainty ranges.”

In general, the manuscript reads very well. The only section that needs some work in terms of the readability and clarity is the introduction. This section could benefit from some restructuring and minor sentence modifications. To that end, I have made some suggested changes and included this as a pdf with tracked changes. Here, the sentence starting on L67 has been moved after the mass budgeting is explained as this seems to flow better. I have modified the aims of the study to improve clarity and have also made many suggestions to improve the readability (see tracked changes pdf). Details of the studies quoted have been removed as this breaks the flow of the introduction. These sentences could be incorporated nicely into the discussion (e.g. sentence on L76 could be incorporated in the sentence on L309, the sentence starting on L86 could be incorporated in the paragraph on L339 and the sentence starting on L89 is redundant as this information is already in the discussion). Finally, I have added a sentence on L70 to first to explicitly make the reader aware that most of the measurements available (e.g. those described on L53-55) are geodetic and include both loss terms before discussing the mass budgeting approach.

Thank you for your detailed and valuable revision on the introduction. We revised the introduction based on the suggested changes.

Detailed comments:

Introduction

Please see the tracked changes .pdf

Thank you for your detailed and valuable revision on the introduction. We implemented the suggested changes.

Results

L110 What is “local”? Please define explicitly.

Thank you for this question. We reformulated the sentence stating more clearly what we compared:

“A comparison between a firn core in the central Cordillera Darwin and modelled precipitation at the closest grid point shows...”

L110-112 This sentence states that “A comparison of local precipitation with a firn core in the central Cordillera Darwin shows an overestimation”, but there is no data to back this statement up. Please provide a quantitative comparison (e.g. amount of overestimation in SWE). Likewise, the statement “catchment-wide precipitation about 50 km downwind indicates an underestimation” is qualitative (Figure S2). Please provide a quantitative comparison as well (e.g. mean and standard deviation for the difference in streamflow between the stream gauges and downscaled data).

Thank you for this comment. We add the main bias between modelled precipitation and both measurements (firn core and stream gauge) to the manuscript:

“A comparison between a firn core in the central Cordillera Darwin and modelled precipitation at the closest grid point shows an overestimation (mean bias of +0.63 m w.e. yr⁻¹) while catchment-wide precipitation about 50 km downwind measured with a stream gauge indicates an underestimation (mean bias of -0.43 m w.e. yr⁻¹) (Fig. 1, Fig. S2).”

The correlation and bias have been added to Fig. S2. The firn core data has not been published yet (currently under review), which is why we cannot show the exact values here.

L114 Consider changing “As we aim at a...” to “As our aim is a...”

Changed, thank you.

L121 I assume the results presented here (e.g. 5.2°C) are from the downscaled climate data, but it would be good to explicitly state this so it is clear.

Changed.

L132 Consider changing to “...wind also show an increasing ...”

Changed to *“Precipitation and wind both show an increasing trend ...”*

L172 consider rephrasing as “...decreasing snowfall towards the east results in CMB decreases....”

Changed.

L237-238 Here the author indicates that “..the majority of MALT glaciers show only low frontal ablation.” Can you quantify and specify what you mean by “low frontal ablation”? Are you referring to low frontal ablation rates for the majority of MALT glaciers compared to the average frontal ablation rate? It would be helpful to provide a simple reference value (e.g. the average frontal ablation of all glaciers compared to the average for 50% of the glaciers with the lowest values)

Thank you for this comment. We added the information, that the majority of MALT glaciers has a frontal ablation $< 0.03 \text{ Gt yr}^{-1}$ and an average contribution to frontal ablation of less than 1%.

“For both approaches, the majority of MALT glaciers shows only low frontal ablation ($< 0.03 \text{ Gt yr}^{-1}$) (Fig. 4) accounting on average for less than 1 % of the total amount.”

L290 I assume you mean global warming rates. If so modify this sentence to “...exceeds global warming rates...”

Thank you for this question. The cited rates are referring to southern South America. We, now, adjusted it to the results for Punta Arenas (located close to the CDI) and added this information to the sentence:

“The warming trend for the beginning of the 21st century found in this study ($+0.42 \text{ }^\circ\text{C per decade}$) exceeds warming rates reported for the 20th century in Punta Arenas ($+0.21 \text{ }^\circ\text{C per decade}$)³⁵ as well as the projected warming in the Magellan region until mid of this century under a high-emission scenario (RCP8.5) (total warming of $+0.5 \text{ }^\circ\text{C until 2050}$)³⁶.”

L360 did you mean “temperate” instead of “temperature” here?

Yes, thank you.

L353 This paragraph could be modified to improve the readability and make it more concise in some places. I have simply highlighted the text that should be reviewed in the tracked changes pdf.

Thank you for this comment. We revised the paragraph and reformulated especially the highlighted parts.

L238 change “For few..” to “For a few...”. Same comment for L253.

Changed.

L251-253 The first two sentences are a bit redundant and could be reduced to: “With the CMB and the frontal ablation available, we can disentangle the total glacier mass changes observed from remote sensing into climatical and dynamical forcing.”

Changed as suggested, thank you.

Figure 1 For clarity it would be good to add the years associated with the CMB data here. For example, the first sentence could be modified to “...(CDI) over the period 2000-2022.”

Thank you for this comment. We adjusted it as suggested.

Figure S2 consider modifying the figure caption from “..to precipitation over the river catchment...” to “..to downscaled precipitation over the river catchment...” so it is clear that the yellow data set is the downscaled data set.

Changed as suggested, thank you.

Discussion

L293 modify “...moisture, more rich in snow conditions...” to “moister, snow-rich conditions...”

Changed as suggested.

L343 change “Alike” to “Like”

Changed.

L378 change “of glacier...” to “and glacier...”

We are analyzing the climatic influence on glacier evolution. Changing the sentence as suggested would change the meaning. Instead, we changed it to:

“Furthermore, these simulations shed light on the climatic imprint of glacier evolution at a unique location in the higher mid-latitudes of the Southern Hemisphere.”

L378 It would be good to highlight why this location is unique and very relevant for climate science. Consider modifying to: "...at this unique location which is part of the only continental landmass crossed by the Southern Hemisphere westerly wind belt." ...or something similar that highlights why the location is unique.

Thank you. We reformulated the sentence to:

"Furthermore, these simulations shed light on the climatic imprint of glacier evolution at a unique location in the higher mid-latitudes of the Southern Hemisphere, the only major land mass disrupting the westerly wind belt."

L353-354 consider modifying to "...we satisfactorily constrain the CMB."

Changed.

L355 consider modifying to "However, energy fluxes..."

Changed the sentence to:

"However, due to a lack of observations most energy fluxes are not directly verifiably."

L359 consider modifying to "Furthermore, the considerable..."

Changed.

L386-387 The current statement about climatic warming is vague and not very convincing. Consider modifying to "...climatic warming has definitively impacted CDI glaciers and their evolution is mainly controlled by atmospheric conditions."

Changed as suggested (without the word 'definitively').

Figure S4 In the caption it says that "The color scheme gives the model error" while in the legend the colour scheme is labeled as the "difference climatic to geodetic mass balance." I am assuming the legend is correct, so the caption should be modified.

Changed, thank you.

Figure S6 change "Snowfall SNOW" to "snowfall (SNOW)"

Changed.

Table S1 – Consider modifying the label for column one from “Name” to “Station name”. Also change the header “responsibility” to “responsible”

Changed.

Table S2 – label the first column (e.g. Station name).

Changed, thank you.

Data and Methods

L395 replace “...temperature via relative...” with “...temperature, relative...”

Replaced as suggested.

L429 consider removing “, which have been missing so far” as this is redundant because it is clear from the previous sentences that there are no outlines for this year.

Removed as suggested.

L430 consider removing “oriented on inventory availability and geodetic data” since this is redundant. It is already mentioned in the preceding sentences.

We agree that the information is already mentioned in the preceding sentences. We still want to keep the statement to summarize the used data.

L442 replace “had been” with “are”

Changed.

Table S2 From the text it sounds like this comparison is between the downscaled climate data and AWS data only (e.g. river catchment data and firn core are not used). It would be good to state this explicitly by modifying the caption to: “...compared to the AWS observations.”

Changed as suggested.

References

- Barcaza, G., Nussbaumer, S.U., Tapia, G., Valdés, J., García, J., Videla, Y., Albornoz, A., and Arias, V. (2017): Glacier inventory and recent glacier variations in the Andes of Chile, South America. *Ann. Glaciol.* **58**, 166–180, doi:10.1017/aog.2017.28
- Bravo, C., Quincey, D. J., Ross, A. N., Rivera, A., Brock, B., Miles, E., & Silva, A. (2019): Air Temperature Characteristics, Distribution, and Impact on Modeled Ablation for the South Patagonia Icefield. *Journal of Geophysical Research* **124** (2): 907-925, <https://doi.org/10.1029/2018JD028857>
- Bravo, C., Ross, A.N., Quincey, D.J., Cisternas, S., Rivera, A. (2022): Surface ablation and its drivers along a west–east transect of the Southern Patagonia Icefield. *Journal of Glaciology* **68** (268), 305–318, <https://doi.org/10.1017/jog.2021.92>.
- Gacitúa, G., Schneider, C., Arigony, J., González, I., Jaña, R., and Casassa, G. (2021): First ice thickness measurements in Tierra del Fuego at Schiaparelli Glacier, Chile, *Earth Syst. Sci. Data*, **13**, 231–236, <https://doi.org/10.5194/essd-13-231-2021>.
- Hall, D. K. & Riggs, G. A. (2021): MODIS/Terra Snow Cover Daily L3 Global 500m SIN Grid. (MOD10A1, Version 61), [Data Set], *NASA National Snow and Ice Data Center Distributed Active Archive Center*, <https://doi.org/10.5067/MODIS/MOD10A1.061>.
- IPCC (2012): Glossary of terms. In: *Managing the Risks of Extreme Events and Disasters to Advance Climate Change Adaptation* [Field, C.B., V. Barros, T.F. Stocker, D. Qin, D.J. Dokken, K.L. Ebi, M.D. Mastrandrea, K.J. Mach, G.-K. Plattner, S.K. Allen, M. Tignor, and P.M. Midgley (eds.)]. A Special Report of Working Groups I and II of the Intergovernmental Panel on Climate Change (IPCC). Cambridge University Press, Cambridge, UK, and New York, NY, USA, pp. 555-564.
- Sauter, T., Arndt, A. & Schneider, C. (2020): COSIPY v1.3 – an open-source coupled snowpack and ice surface energy and mass balance model. *Geoscientific Model Development* **13**, 5645–5662, <https://doi.org/10.5194/gmd-13-5645-2020>.
- Temme, F., Farías-Barahona, D., Seehaus, T., Jaña, R., Arigony-Neto, J., Gonzalez, I., Arndt, A., Sauter, T., Schneider, C., and Fürst, J. J. (2023): Strategies for regional modeling of surface mass balance at the Monte Sarmiento Massif, Tierra del Fuego, *The Cryosphere*, **17**, 2343–2365, <https://doi.org/10.5194/tc-17-2343-2023>.

Dear editor and referees,

We would like to thank you for the constructive review of our manuscript. We are convinced that your suggestions have improved the readability and overall quality of the manuscript.

For reviewers 1 and 3, the requested changes were mainly of editorial nature. They are fully implemented in the revised version of the manuscript. Reviewer 2 raised concerns not on the manuscript but on terminology used in the rebuttal to the first review round. To avoid ambiguities or misunderstanding, we decided to adapt the terminology to be clear and specific.

Please find our detailed reply to the individual comments below.

Kind regards,

Franziska Temme,
in the name of all co-authors

Point-by-point response to reviewer comments

Reviewer #1 (Remarks to the Author):

Thank you to the authors for answering and clarifying my concerns regarding their manuscript. I understand and agree with the answers and changes introduced in the manuscript.

Just some (very) minor comments:

We would like to thank you for the constructive review of our manuscript. In the following you find a point-by-point response to the raised comments. We are convinced that our actions have improved the quality and readability of the manuscript. Referee comments are reproduced in black font color. Our response and the undertaken actions are formulated in green font color, text that was adjusted in the manuscript is highlighted in italic font.

L49-50: However, stratospheric ozone depletion also forces storm-track poleward trend. Recovery of the ozone is projected so this could counteract the poleward trend. Do the authors analyze this scenario? And how could this affect CDI mass balance? Thank you for this question. The counteracting effect of ozone recovery is considered in the cited publication (Goyal et al. 2021). They mention that ozone depletion is seen as the main forcing mechanism for the southward shift in the past few decades and discuss the counteracting effect of ozone recovery. However, they find that, for high emission scenarios, the effect of greenhouse gases will dominate over the effect of ozone recovery and cause a future southward shifting. In general, a southward shift of the westerlies would cause an intensification of those over the CDI, potentially enhancing precipitation. For our study period we observe an increasing trend in wind velocities and precipitation (see L137-138) that can be explained by the shift in westerlies. However, the trend for precipitation is small and we do not find a significant increase in snowfall amounts (see L229-230).

L179 and L300-313: I suggest presenting the ELA results as m a.s.l. (I'm sorry I didn't note this in the first version.)

Adjusted as suggested, thank you.

L188: delete space between 15 and %

According to the International System of Units, a non-breaking space should be added between the number and the % symbol. We found a few occasions in the text, where the space was missing so far and adjusted those in the revised version of the manuscript to ensure consistency.

L556: so, Which one is used in the bulk-approach applied here?

Thank you for this question. In this study, we apply Monin-Obukhov similarity theory, as noted in Table S5. We added this information into the text:

“The latter is applied in this study (Table S5).”

Answering to the authors, there are several energy balance studies for short-term in

Patagonia linked to Japan's research program on Patagonia. Most of them, were published in the Bulletin of Glacier Research during the 80's and 90's.

Thank you for this hint. Despite an extensive search, the only publication we were able to access is Takeuchi et al. 1999, *Global and Planetary Change*. We incorporated this publication for completeness.

Reviewer #2 (Remarks to the Author):

Dear Editor,

I have reviewed the authors' responses to my comments; however, I have not had the opportunity to thoroughly review the revised manuscript itself. Based on the responses provided, I am generally satisfied and believe the authors have addressed my main criticisms.

We would like to thank you for the constructive comments on our manuscript. In the following you find a response to the raised comments. We are convinced that our actions have improved the quality and readability of the manuscript. Referee comments are reproduced in black font color. Our response and the undertaken actions are formulated in green font color, text that was adjusted in the manuscript is highlighted in italic font.

That said, I would like to bring to your attention a few minor points from their responses that may require further clarification:

The authors state: "Taking only the land-terminating glaciers, the dynamic component is zero, and the climatic and geodetic mass balance are directly comparable." While this is true when considering glacier-wide (or region-wide) mass balance, it does not hold if distributed specific mass balance (e.g., mass balance versus elevation) of individual glaciers is considered. Geodetic mass balance for any glacier (land-terminating or not) always reflects a combination of surface mass balance (climatic) and ice flow (dynamic). This distinction was not fully addressed in the authors' response, and I suggest that this clarification should be explicitly incorporated into the manuscript to ensure methodological accuracy.

The authors also mention: "We are comparing the glacier-wide integrated mass balance (called specific mass balance) for each land-terminating glacier." However, glacier-wide mass balance is not the same as specific mass balance, and these terms should not be conflated. This mislabeling caused confusion in their responses and could lead to further misunderstandings in the manuscript. I recommend that the authors refer to the Glossary of Mass Balance (https://wgms.ch/downloads/Cogley_etal_2011.pdf) to ensure that terminology is used correctly and consistently throughout the paper.

Finally, the authors' use of the term "ice dynamics" appears to describe glacier calving or frontal ablation. However, ice dynamics is often synonymous with ice flow, which occurs in all glaciers regardless of their terminus type (land or marine). To avoid confusion, I suggest the authors revise this terminology and use more precise language when describing these processes (again, see the Glossary for guidance).

I hope these observations are helpful in guiding the final revisions. Please let me know if any further input is required from my side.

Thank you for your comments. Regarding the terminology of “specific mass balance” we understand your concerns and excuse for the confusion. With the term “specific mass balance” we intend to refer to the glacier-wide specific mass balance, where the statement that ice-dynamically controlled losses are zero for land-terminating glaciers holds true. We agree that this is not true in a case where not the entire glacier area but only a subsection is considered. Throughout the manuscript we mainly use the term “glacier-specific mass balance”, which is a widely used terminology when referring to the specific mass balance of an entire glacier and should resolve this ambiguity. We added “glacier-“ where it was missing so far (L627 & caption of figure 1) to ensure consistency. Furthermore, we added the following sentence in L149: *“Glacier-wide specific mass balance is used for the comparison, in the following denoted as glacier-specific mass balance.”*

Regarding the terminology of “frontal ablation” and “ice-dynamically controlled mass loss”, we understand your concerns. In the Introduction (L65-67) (and in the last, summarizing paragraph of the Discussion), we describe that frontal ablation is a loss term that is controlled by ice-dynamics through calving. To avoid confusion, we now use the term “frontal ablation” throughout the results and discussion sections of the manuscript.

Reviewer #3 (Remarks to the Author):

Dear authors,

The manuscript has been improved significantly through the review process and all my previous comments and additional concerns have been addressed directly or indirectly through your response to the reviewers. The remaining comments I have are minor.

Kind regards,
Nicole Schaffer, PhD.

We would like to thank you for the constructive review of our manuscript. In the following you find a point-by-point response to the raised comments. We are convinced that our actions have improved the quality and readability of the manuscript. Referee comments are reproduced in black font color. Our response and the undertaken actions are formulated in green font color, text that was adjusted in the manuscript is highlighted in italic font.

Detailed comments:

L38-39 Suggestion to delete "...with glaciers descending down to sea level.." as this is redundant since it can be inferred from the previous sentence. I realize this was included to respond to a comment from reviewer 1 so if you decide to keep it for that reason that is ok. The modification to include "..inducing high mas input.." in response to reviewer 1 is a helpful addition but I would suggest replacing "at the top" with something more explicit such as "at the highest elevations".

Thank you for this comment. We adjusted the sentence as suggested.

L46-47 suggestion to modify this sentence to "..not only the formation of clouds...but also the global ocean..."

Changed as suggested, thank you.

L78-79 The term "this way" is somewhat vague and informal. Suggestion to replace this with "With this dataset, ...". Also consider replacing "given" with a more appropriate word such as "produced".

Adjusted as suggested, thank you.

L272-278 The sentence starting on L272 and the rest of this paragraph is mostly duplicated in the new discussion paragraph on advancing glaciers. Given that this part of the paragraph is more suited for the discussion since it is an interpretation of the results, I would suggest deleting this text from the results and merging it with the new paragraph in the discussion (see my comments further down for that paragraph).

Thank you for this comment. We moved the paragraph to the discussion.

L313 change "extends generally" to "generally extends"

Changed as suggested, thank you.

L315 Replace the term “exposition” with a more suitable term. For example, “aspect”.
Adjusted as suggested, thank you.

L311-320 In the second sentence within this paragraph, the author highlights that MALT glaciers show more positive CMB. This prompts one to question whether all the advancing glaciers are MALT glaciers? I would suggest adding a sentence starting on L315 (if it is true that they are all MALT, otherwise modify!) to answer this question and aid in linking the sentence on MALT to the discussion on advancing glaciers. Additionally, moving the last sentence to L317 would help with the readability and cohesive flow of ideas. Finally, I would suggest merging this paragraph with the text on L272-278 to reduce unnecessary repetition in the manuscript as follows:

“...century. While these advancing glaciers are all MALT, they have medium to low frontal ablation which, in addition to a thickening further upstream, suggest a primarily climatic control. We ascribe these climatological favorable conditions to the glaciers’ hypsometry and aspect. All four glaciers have their origin at the high-elevated central plateau of the CDI (above 2000 m) and steep topography resulting in high snowfall amounts and above-average accumulation-area-ratios (between 0.72 and 0.88 compared to the CDI average of 0.61). Glaciers with high accumulation-area-ratio are often less sensitive to changes in ELA and, thus, warming due to the small and steep ablation area¹⁵. Southward exposures of Garibaldi, Guilcher Oeste and Guilcher Este glaciers further reduce surface ablation (Fig. S8a). Altogether, these characteristics ultimately lead to high snowfall amounts together with reduced surface ablation (Fig. 2c, Fig. S9d), favoring mass gain.”

Thank you for this comment and the detailed revision of the paragraph. We adjusted the paragraph following your suggestions. The advancing glaciers are all marine terminating. We added this information.

L311-320 (b) Does the strong precipitation gradient across the CDI have an impact on the contrasting glacier behaviour? If you consider the answer relevant and worth mentioning, please add a sentence to this paragraph.

Thank you for this question. We do not see the precipitation gradient as a relevant forcing for the contrasting glacier behavior. The reason is that we see retreating glaciers on the west and east part of the CDI, thus there is no zonal gradient in the retreating/advancing behavior. The advancing glaciers are all located in the central area and are also surrounded by several retreating glaciers.

L365 suggestion to replace “...ablation at the CDI...is in a similar...” with “...ablation for the CDI...is a similar...”

Adjusted as suggested, thank you.

L377 replace “...study are..” with “...study is...”

Adjusted to “*A major limitation of this study is the scarcity of in-situ observations, ...*”

L379 replace “verifiably” with “verifiable”

Changed as suggested, thank you.

L380 suggestion to replace “...should be handled with certain care..” to “...should be interpreted with care...”

Adjusted as suggested, thank you.

L443 Consider changing “Required variables comprise...” to “Required variables for this study are ...”

Adjusted as suggested, thank you.

L445-447 consider changing sentence to “For downscaling of precipitation (53.75-55.50°S, 74.50-73.25°W) information about air temperature, relative humidity, wind vectors and geopotential height between 850 and 500 hPa is needed.” The term “we need” is too informal for a publication in my opinion.

Changed as suggested, thank you.

L487-488 Reading the phrase “...where the cumulative distribution function of the model is transferred to the cumulative distribution function of the observation” could be interpreted to mean that the cumulative distribution of the observation data is corrected by aligning it with the cumulative distribution of the model output. From my limited understanding of quantile mapping bias correction, I understand that the correction is the other way around. The quantile distribution of the model outputs is shifted to align with the quantile distribution of the observed data. Consider modifying the sentence to “...where the cumulative distribution function of the model is shifted to align with the cumulative distribution function of the observations”

Thank you for this comment. We adjusted the section to “... where the cumulative distribution function of the model is adjusted to align with the cumulative distribution function of the observation”

Table S2 The RMSE for the station Rio Azorpado is quite low (0.33). Do you know why this might be? If relevant, include the explanation in the manuscript.

We do not have a clear explanation for the low correlation for precipitation at Río Azorpado. We screened the timeseries for outliers or periods of consecutive low or high measurements and found many periods of corrupt values that we deleted, but it is possible that this did not resolve all issues. Therefore, issues with the sensor would be the most obvious explanation.

Climate's firm grip on glacier melting in the Cordillera Darwin Icefield, Tierra Del Fuego

Franziska Temme^{1*}, Christian Sommer¹, Marius Schaefer², Ricardo Jaña³, Jorge Arigony-Neto⁴, Inti Gonzalez^{5,6}, Eñaut Izagirre^{7,8}, Ricardo Giesecke^{9,10}, Dieter Tetzner¹¹, Johannes J. Fürst¹

¹Institut für Geographie, Friedrich-Alexander-Universität Erlangen-Nürnberg, Erlangen, Germany

²Instituto de Ciencias Físicas y Matemáticas, Universidad Austral de Chile, Valdivia, Chile

³Departamento Científico, Instituto Antártico Chileno, Punta Arenas, Chile

⁴Instituto de Oceanografía, Universidade Federal do Rio Grande, Rio Grande, Brazil

⁵Centro de Estudios del Cuaternario de Fuego-Patagonia y Antártica, Punta Arenas, Chile

⁶Programa Doctorado Ciencias Antárticas y Subantárticas, Universidad de Magallanes, Punta Arenas, Chile

⁷Hydro-Environmental Processes Research Group, University of the Basque Country UPV/EHU, Leioa, Spain

⁸Basque Centre for Climate Change BC3, Leioa, Spain

⁹Instituto de Ciencias Marinas y Limnológicas, Universidad Austral de Chile, Valdivia, Chile

¹⁰Centro FONDAP de Investigación en Dinámica de Ecosistemas Marinos de Altas Latitudes (IDEAL), Valdivia, Chile

¹¹Ice Dynamics and Paleoclimate, British Antarctic Survey, Cambridge, UK

*corresponding author: franziska.temme@fau.de

Abstract

The Cordillera Darwin Icefield (CDI) in Tierra del Fuego is one of the largest ice bodies in the Southern Hemisphere. We present the first simulation of the climatic energy and mass balance of its glaciers, which are sensitive indicators of climatic changes in the higher mid-latitudes of the Southern Hemisphere. Conditions are characterized by year-round westerly winds causing strong climatic gradients across the cordillera, which are reflected in the climatic energy and mass fluxes. Our results reveal a significant increase in surface melt ($+0.18 \text{ m w.e. yr}^{-1}$ per decade) over the past two decades. We also present the first estimate of dynamically controlled mass loss into adjacent fjords and lakes by frontal ablation. It amounts to $1.44 \pm 0.72 \text{ Gt yr}^{-1}$, accounting for 26 % of the total mass loss in the CDI. Frontal losses are mainly channelized through few marine-terminating glaciers. We conclude that while ice dynamics are important for predicting the fate of individual glaciers, for the CDI as a whole, atmospheric conditions exert the main control on the current glacier evolution.

1. Introduction

The Cordillera Darwin Icefield (CDI) is one of the largest icefields in the Southern Hemisphere ¹, holding a substantial mass of ice that is at least twice as large as the mass of all glaciers in the European Alps ^{2,3}. The main continuous icefield covers the Cordillera Darwin mountain range and is extended by few smaller adjacent ice bodies separated by fjords (Fig. 1), such as the Mount Sarmiento Massif in the west, summing up to a total glaciated area of 2356 km² in 2022 ⁴. Glaciers in the CDI descend from up to 2500 m a.s.l. down to sea level. The large altitudinal range is possible due to the extraordinary location and the extreme climatic conditions. Tierra del Fuego, located at the southernmost end of South America (Fig. 1a), is the closest land mass to Antarctica. Being situated between the subtropical anticyclone and the subpolar low-pressure trough, the area faces strong, year-round westerly winds. Within this so-called storm track, frontal systems continuously transport moist air masses towards the continent ⁵. Orographic uplift causes abundant precipitation along the western slopes of the Cordillera Darwin while lee-side effects result in more arid conditions in the east ⁶. The strength and position of the Southern Hemisphere westerlies impacts not only on the formation of clouds and precipitation but also on the global ocean circulation ⁷. In the past few decades, the storm track has shifted poleward due to an intensification of the subtropical high in the southeast Pacific which is partly ascribed to human induced climate change ⁸. The southward shifting together with and intensification of the westerlies is projected to continue until the end of the 21st century based on high emission scenarios ⁷. Southern Patagonia is the only continental land mass disrupting the Southern Hemisphere westerly wind belt. Since glaciers are susceptible indicators of climate change, the glacier evolution of the CDI provides valuable insights into climatic changes in this region.

In the last decades, the CDI experienced strong ice loss ^{1,9,10}, contributing about 5 % of the total loss in South America between 2000 and 2011/2014 ⁹. Estimates of mass loss rates for Tierra del Fuego range between 1.02 ± 0.11 Gt yr⁻¹ (2000-2011/14) ⁹ and 1.9 ± 1.1 Gt yr⁻¹ (2000-2018) ¹⁰. Despite the general retreat pattern, individual glaciers are stable or even advancing, mostly in the central region of the CDI ¹¹. Such advance continues to this day as confirmed in glacier inventories covering the last two decades (Fig. S1). The largest advance of around 2 km was observed for Garibaldi Glacier (Fig. S1). The advancing behavior is all the more remarkable when compared to the extreme retreat of Marinelli Glacier ¹², the largest glacier of the CDI, which is located in close vicinity at the northern slope of Mount Shipton (Fig. 1c). From 1945 to 2005, it experienced an extreme recession of 12.2 km ¹³, explained by warming and fast retreat along over-deepened fjord bathymetry ¹².

Around 60 % of CDI glaciers are marine- or lake-terminating (MALT) ¹⁴. Thus, they do not only lose mass on their surface in contact with the atmosphere, but also at the ocean/lake interfaces via iceberg calving and subaqueous melting, collectively known as frontal ablation ¹⁵, which is controlled by ice dynamics and fracturing. Ice is also lost at the ocean/lake interfaces via iceberg calving and subaqueous melting, collectively known as frontal ablation ¹⁵. The ice losses and the contrasting behavior observed

for individual CDI glaciers can possibly be explained by both climatic and ice-dynamic changes. ~~The attribution of ice loss to climatic or ice-dynamic forcing is still missing to this day.~~ ~~Most mass balance estimates are geodetic^{1,9,10} which includes both of these loss terms giving the total mass change.~~ Attribution is possible for individual glaciers if one of the two ice-loss terms is computed. In a mass budgeting approach, the total mass change is quantified, along with one of the two primary loss terms controlled either by climate or ice-dynamics, leaving the other as a residual¹⁸. However, due to the harsh climatic conditions and the inaccessibility of this region, the CDI remains poorly studied^{13,14,16,17} and ~~neither of~~ these two loss terms cannot be quantified reliably. The climatic mass balance (CMB) in the Cordillera Darwin was conducted in the Cordon Martial (east of the study region, located in Argentina)¹⁹ and the Mount Sarmiento Massif^{20,21}, but these local efforts are insufficient and a CDI-wide estimate of the CMB is needed. ~~The attribution of ice loss to climatic or ice dynamic forcing is still missing to this day.~~ ~~It can~~ This would make attribution of the CDI possible and additionally provide ~~give~~ vital information on current trends and shifts in atmospheric conditions of the Southern Hemisphere's higher mid-latitudes.

~~Attribution is possible for individual glaciers if one of the two ice-loss terms is computed. In a mass budgeting approach, the total mass change is quantified, along with one of the two primary loss terms controlled either by climate or ice-dynamics, leaving the other as a residual¹⁸. First attempts to study the climatic mass balance (CMB) in the Cordillera Darwin were conducted in the Cordon Martial (east of the study region, located in Argentina)¹⁹ and the Mount Sarmiento Massif^{20,21}. In both areas, the mass balance is negative and dominated by snowfall and surface melt, which is strongly linked to air temperature^{19,20}. Despite these local efforts to determine the mass balance of individual glaciers, a systematic approach for a CDI wide estimate of the CMB is to this day missing. This lack of knowledge can be addressed with a well informed modelling product of CMB. Geodetic techniques in satellite remote sensing allow for operational inference of total glacier mass changes worldwide^{9,10,22}. Thus, with the climatic loss term available, the mass budget of the CDI glaciers can successfully be closed and give the dynamic loss as residual. Another common method~~ An alternative approach is the direct quantification of frontal losses with a flux gate approach^{15,23}. Here, the frontal ablation is estimated based on the ice flux through a gate upstream of the glacier front. Due to the lack of ice thickness observations in the CDI producing high uncertainties on reconstruction products (e.g., ref.^{2,3}), results ~~will, however, remain~~ are inaccurate. First estimates of frontal ablation in the CDI are limited to two glaciers: At Schiaparelli Glacier frontal ablation calculated with a mass budgeting approach²⁴ agrees well with inferred values from time-lapse camera observations²⁵. At Marinelli Glacier, Koppes et al.¹⁶ find a calving flux of around 0.4 Gt yr⁻¹ (2000-2005) applying an ice budget model.

The main aim of this study is to 1) quantify the unknown CMB of the CDI and 2) attribute ice loss to climatic or ice-dynamic forcing. We simulate the CMB over a 22-year period which provides first insights in climatic and glaciological trends and increase our process understanding of glacier response

Commented [NS1]: Move the preceding sentence to another section (discussion)?

~~to climate. The energy and mass balance will be examined using the physically-based CoCoupled Snowpack and Ice surface energy and mass balance model in PYthon (COSIPY)²⁶, which combines a surface energy and mass balance model with a subsurface multi-layer snow and ice model (Methods section). The fully distributed model allows an analysis of the spatial and temporal variability in the surface energy and mass fluxes across the CDI. Together with geodetic observations, we can use the CMB to close the mass budget of all glaciers in the CDI (objective 2) and provide a first estimate of their frontal losses. In this way, an unprecedented attribution of the observed mass loss to climatic and ice-dynamic forcing becomes possible. With a continuous 22-year CMB simulation, we will furthermore provide first insights in climatic and glaciological trends, and increase our process understanding of glacier response to climate. The energy and mass balance will be examined using the physically-based CoCoupled Snowpack and Ice surface energy and mass balance model in PYthon (COSIPY)²⁶, which combines a surface energy and mass balance model with a subsurface multi-layer snow and ice model (Methods section). The fully distributed model allows an analysis of the spatial and temporal variability in the surface energy and mass fluxes across the CDI. The unprecedented attribution of the observed mass loss to climatic and ice-dynamic forcing plus the fully distributed CMB model allow allowing conclusions to be drawn on the contrasting glacier behavior in the last two decades.~~

Results

Climatological characteristics of the CDI

High-resolution (200 m spatial and 3-hourly temporal) atmospheric forcing was created by observation-informed downscaling of ERA5 reanalysis data (Methods section). Our multi-method downscaling relies on quantile mapping²⁷ as well as modelling of solar radiation²⁸ and orographic precipitation²⁹⁻³¹ for the study period 04/1999-03/2022. Variables comprise air temperature, relative humidity, air pressure, wind velocities, cloud cover, incoming solar radiation and precipitation (Methods section). Evaluation of atmospheric variables shows overall good agreement with observations from automatic weather stations (Table S1, Table S2). A comparison of local precipitation with a firn core in the central Cordillera Darwin shows an overestimation while catchment-wide precipitation about 50 km downwind indicates an underestimation (Fig. 1, Fig. S2). The location of the former firn core is in an exposed saddle position where the local wind field and snowdrift are not resolved at the process level, while the latter catchment comparison suffers from neglecting water storage. As we aim at a first climatic mass balance estimate on regional scale, consistent with regional geodetic measurements, local deviations are acceptable. The CDI is divided into four subregions (Fig. 1b) to analyze spatial variability of climatic characteristics and energy and mass fluxes across the study region. Significance of trends over the study period are formulated following the IPCC guidance for communication of confidence (Table S3)³². These different significant levels are marked in italic font.

The study region is characterized by temperate maritime climate. Annual mean air temperatures close to sea level (2 m above ground) lie around 5.2 °C with moderate interannual variability (± 3 to 4 °C). Air temperatures show a positive trend (*virtually certain*) over the study period of +0.42 °C per decade with an intensification from west to east. Conditions are overall humid with annual average relative humidity showing a drying towards the east (Fig. S3b). Relative humidity is on average higher in winter than summer, with the amplitude increasing from around 7 % in the west to around 12 % in the east. Within the westerly wind belt, frontal systems move from the southern Pacific Ocean towards southern Patagonia causing high precipitation amounts due to orographic uplift. Highest amounts are reached in the westernmost edge of the Cordillera Darwin, going up to 4000 mm yr⁻¹ at Mount Sarmiento (Fig. 2a). As the air masses move eastwards over the cordillera, fallout of precipitation causes a drying effect towards the east of the CDI (Fig. 2a, Fig. S3e). Precipitation amounts peak in the summer months, related to the increased wind velocities. The seasonality is forced by a southward shift of the westerly wind belt during summer³³. Precipitation and wind also agree on an increasing trend (*likely*) over the study period (+70.8 mm yr⁻¹ per decade and +0.1 m s⁻¹ per decade, respectively). Wind velocities are high throughout the year especially in the westernmost edge of the CDI where annual averages go up to 5.5 m s⁻¹, and decrease towards the central and eastern part of the CDI (Fig. 2b, Fig. S3d). High amounts of annual precipitation are accompanied by an average cloud cover of over 84 %. Such extensive cloud cover strongly limits direct solar radiation (Fig. S3c, Fig. S8a). Regional radiation differences between north and south are mainly explained by the orientation to the sun and the aspect of the slopes.

Climatic energy and mass balance of the CDI

The climatic energy and mass balance is simulated with COSIPY²⁶, a fully-distributed surface energy and mass balance model coupled to a multi-layer subsurface snow and ice model (Methods section). For model evaluation, simulation results are compared to glacier-specific geodetic mass balances from elevation change observations (2000-2013) based on the results by Braun et al.⁹ for glaciers with no frontal losses (Methods section) (Fig. S4). Based on this performance, we infer a glacier-specific CMB uncertainty of ± 0.62 m w.e. yr⁻¹. The simulated icefield-wide average CMB of land-terminating glaciers (-0.23 m w.e. yr⁻¹) agrees well with the geodetic reference dataset of Braun et al.⁹ (-0.27 m w.e. yr⁻¹) (Fig. S5).

The CMB of the entire CDI (including marine-, lake- and land-terminating glaciers) is nearly balanced for the study period (2000-2022) with +0.02 m w.e. yr⁻¹. Temporal variability is high with annual values between -0.81 and +0.93 m w.e. yr⁻¹ (Fig. 3c). Across the CDI, glacier-specific values show a high spatial variability (Fig. 1c), ranging from -5.44 m w.e. yr⁻¹ to +2.42 m w.e. yr⁻¹. We relate this diversity to hypsometry and aspect of the respective glacier catchments. MALT glaciers show on average a more positive CMB than land-terminating glaciers. Positive CMBs dominate especially in the central high-elevated part of the CDI (e.g., Ruginor, Garibaldi, Guilcher oeste and este), whereas lower-elevated glaciers or glaciers with large outlet tongues show more pronounced mass loss (e.g., Schiaparelli,

Romanche, Alemania and many of the small, unnamed glaciers at the icefield margin) (Fig. 1c). The altitudinal gradient of the CMB is steep with strong mass losses on the glacier tongues at low elevation and high mass gain towards the mountain peaks (Fig. 2c). The largest range is present in the western part of the CDI, where the annual CMB reaches nearly $-10 \text{ m w.e. yr}^{-1}$ at the lowest point of Schiaparelli Glacier and nearly $+10 \text{ m w.e. yr}^{-1}$ at the highest peak (Fig. 2c), resulting in a gradient of $-0.92 \text{ m w.e. yr}^{-1}$ per 100 m.

Mass is mainly gained from snowfall (62 %) at the surface and from refreezing within the snowpack (37 %) (Fig. 3a). Deposition of water vapor at the glacier surface contributes around 1 % to the accumulation. Mass loss is dominated by melt at the surface (92 %) and subsurface (7 %), with sublimation contributing only around 1 % to the total ablation. The average annual CMB is positive in the southern and western part of the CDI ($+0.35$ and $+0.17 \text{ m w.e. yr}^{-1}$, respectively), while it is close to zero in the north and east ($+0.03$ and $-0.06 \text{ m w.e. yr}^{-1}$, respectively) (Fig. 3a). These differences mainly stem from the contributions of snowfall and surface melt. Snowfall amounts strongly reduce from $+1.51 \text{ m w.e. yr}^{-1}$ in the west over south and north to $+0.92 \text{ m w.e. yr}^{-1}$ in the east of the CDI (Fig. 3a). In the western part, the high accumulation is partly balanced by enhanced surface melt ($-1.95 \text{ m w.e. yr}^{-1}$). Over the remaining CDI we find very similar surface melt rates of $-1.50 \text{ m w.e. yr}^{-1}$. With the decreasing snowfall towards the east, also the resulting CMB decreases (Fig. 3a). The CMB shows a pronounced seasonality (Fig. S6). In austral summer (December, January, February), it is strongly negative with the highest losses in the western ($-0.72 \text{ m w.e. yr}^{-1}$) and the smallest losses in the southern ($-0.48 \text{ m w.e. yr}^{-1}$) part of the CDI (Fig. S6a). This spatial difference is primarily explained by spatial variability in the surface melt. In austral winter (June, July, August), the CMB is positive all over the CDI with reduced melting (Fig. S6c). Regional differences are mainly linked to the snowfall gradient.

To understand the variability in surface melt, the energy fluxes at the surface and subsurface are analyzed. On average, the largest energy input comes from the net shortwave radiation (51 %) and the sensible heat flux (32 %) (Fig. 3b). The glacier heat flux (energy generated from penetrating shortwave radiation and refreezing that is transported to the surface via heat conduction) brings on average 15 % of energy, and is mostly constrained to the higher reaches of the glaciers (Fig. S9b). The heat flux from rain is limited to the glacier tongues (Fig. S9c), contributing around 2 % in total. Energy loss at the surface is dominated by the net longwave radiation (88 %), followed by the latent heat flux (12 %), which is an energy source on the glacier tongues but a sink on the spatial average (Fig. S9a). The largest amount of energy available for melting is found in the western part of the CDI (Fig. 3b, Fig. S9d). The energy surplus mainly stems from the sensible heat flux that is more than twice as high (14.74 W m^{-2}) than compared to the northern and eastern part (around 6.30 W m^{-2}) of the CDI (Fig. 2d). Subsequently, the sensible heat flux is the primary energy source (47 %) in the west (Fig. 3b). The enhanced values there are related to the higher wind velocities prevailing over the south-western part of the CDI (Fig. 2b). The slightly reduced net shortwave radiation in the south-west is caused on the one hand by reduced

incoming radiation due to orientation and topographic shading, on the other hand by higher average surface albedo (around 0.83 in the south-west, around 0.81 in the north-east) related to the enhanced snowfall (Fig. S8). The latent heat flux is an energy sink for the south, north and east part of the CDI (Fig. 3b). Its importance is reduced in the west. Due to the moister conditions in the west (Fig. S3b), water vapor transport towards the glacier surface is more common generating deposition and condensation instead of sublimation and evaporation, which are favored in the other sub-regions (Fig. S7, Fig. S9a). In summer, the latent heat flux temporarily turns into an energy source in the west, which is not seen over the rest of the icefield (Fig. S6b). The energy balance shows a strong seasonality with highest melt energy in summer (Fig. S6b) and strongly reduced energy in winter (Fig. S6d). In winter, energy availability is limited due to the minimized solar energy together with reduced wind velocities and lower air temperatures.

Our results show that the annual CMB (Fig. 3c) is mainly controlled by air temperature (Fig. S3a) (Pearson's correlation $r = -0.71$) and by precipitation (Fig. S3e) dictating accumulation ($r = 0.72$). The air temperature strongly influences the energy available for melting ($r = 0.85$). Melt energy and CMB are also linked to the annual average incoming solar radiation ($r = 0.54$ and $r = -0.45$, respectively). The importance of wind speed over the CDI is highlighted by a significant correlation with the sensible heat flux ($r = 0.62$) as well as accumulation ($r = 0.52$).

Over the 22-year study period, we see an increasing trend (*extremely likely*) in the surface melt ($+0.18$ m w.e. yr^{-1} per decade) that translates into a *likely* decreasing trend for mass balance (-0.21 m w.e. yr^{-1} per decade) which is caused by the more negative annual CMB towards the end of the study period (Fig. 3c). The increasing surface melt is mainly caused by an *extremely likely* increasing trend in the sensible heat flux ($+0.71$ W m^{-2} per decade) and net longwave radiation ($+0.63$ W m^{-2} per decade), which we primarily relate to increasing air temperature (both variables) and wind velocity (sensible heat flux). Additionally, we observe an *extremely likely* increasing net shortwave radiation ($+0.81$ W m^{-2} per decade), which we explain with an albedo feedback (albedo decrease of -0.01 per decade, *extremely likely*). Altogether, an *extremely likely* increasing trend in available melt energy ($+1.93$ W m^{-2} per decade) is obtained. The trend in the CMB is more pronounced for lower elevation bins. Below 600 m elevation, the CMB shows an *extremely likely* decreasing trend (up to -0.47 m w.e. yr^{-1} per decade). Snowfall or accumulation trends are not significant and currently unable to compensate for the increased surface melt.

Estimation of frontal ablation in the CDI

Mass budgeting of the geodetic and climatic mass balance gives information on frontal ablation of the 39 marine- and lake-terminating (MALT) glaciers of the CDI (Methods section). This way, we are able to assess the direct ice flux into the fjords. For comparison, we also calculate frontal ablation following a flux gate approach relying on surface velocities and reconstructed ice thickness³ (Methods section).

However, these results inhibit large errors due to the uncertain modelled ice thickness data in the CDI in the absence of in-situ observations.

With the mass budgeting approach, the total frontal ablation of the CDI is estimated to $1.44 \pm 0.72 \text{ Gt yr}^{-1}$ in 2000 to 2013. About half of the total flux is channelized through Marinelli Glacier with $0.40 \pm 0.08 \text{ Gt yr}^{-1}$ and Grande Glacier with $0.30 \pm 0.08 \text{ Gt yr}^{-1}$ (Fig. 4). Another 20 % of the frontal ablation is explained by two other prominent marine-terminating glaciers: Darwin with $0.14 \pm 0.03 \text{ Gt yr}^{-1}$, and Rugidor with $0.11 \pm 0.03 \text{ Gt yr}^{-1}$. The flux-gate approach gives almost the same total frontal ablation with $1.49 \pm 0.54 \text{ Gt yr}^{-1}$ for the year 2013 and shows the overall same pattern (Fig. 4). Largest frontal losses are found for the same four glaciers, however, in this case being responsible for only around half of the total flux. For both approaches, the majority of MALT glaciers show only low frontal ablation (Fig. 4). For few glaciers, the flux-gate approach shows clearly elevated values - though with large uncertainties. In general, calving into fjords explains the main share of the frontal ablation (around 90 %) although these glaciers constitute only 75 % of the surface area of all MALT glaciers. Calving into lakes plays a secondary role.

The availability of geodetic datasets covering multiple observation periods (e.g., ref. ²²) allows an analysis of changes in frontal ablation rates over the study period (from 2000-2010 to 2010-2020). Bias-corrected estimates based on mass budgeting using these datasets and the CMB, reveal a total frontal ablation of $1.55 \pm 0.71 \text{ Gt yr}^{-1}$ (equaling $1.34 \pm 0.61 \text{ m w.e. yr}^{-1}$) for the first period, and a slightly reduced value of $1.42 \pm 0.75 \text{ Gt yr}^{-1}$ (equaling $1.25 \pm 0.66 \text{ m w.e. yr}^{-1}$) for the second period. Associated uncertainties do not allow any reliable statement to be made about changes over the study period. The derived values suggest, however, that the frontal ablation has changed less over the last 20 years ($-0.05 \text{ m w.e. yr}^{-1}$ per decade) than the CMB.

Disentangling climatical and dynamical control on observed glacier changes

With the CMB and the frontal ablation available, we are able to quantify the main contributors to the mass change in the CDI. This way, we can disentangle the glacier mass changes observed from remote sensing into climatical and dynamical forcing. For few glaciers, the mass budgeting suggests frontal ablation as the major contributor to mass loss. The frontal ablation exceeds twice the climatic ablation for glaciers Rugidor, Marinelli, Darwin, Italia and an unnamed glacier at the south-western margin of Parry fjord (CL112833070) (Fig. 1c, Table S4). For these glaciers, the mass loss is primarily dictated by ice dynamics instead of climate. For about 15% of the glaciers, the contribution of climatic and frontal ablation is rather even (e.g., Grande, Cuevas, Lovisato, Frances) (Table S4). However, for more than half of the MALT glaciers, the climatic losses at the surface and within the snowpack largely exceed the frontal ablation, meaning that these glaciers are dominated by atmospheric processes at the glacier surface rather than ice dynamics.

Glaciers that have been advancing over the study period (Garibaldi, Finlandia, Guilcher oeste and este) are in general characterized by a strongly positive CMB and by medium to low frontal ablation (Fig. 1c). These glaciers do not only advance, but also show thickening further upstream (thus, gaining mass), suggesting a primary climatic control. We ascribe these climatological favorable conditions to the glaciers' exposition, geometry and hypsometry. All four glaciers have their origin at the high-elevated central plateau of the CDI (above 2000 m) causing a steep topography, expressed in an over-average accumulation-area-ratio (between 0.72 and 0.88 compared to the CDI average of 0.61). Glaciers with high accumulation-area-ratio are often less sensitive to changes in equilibrium line altitude (ELA) - the elevation at which surface mass gain equals surface mass loss over one year - and, thus, warming due to the small and steep ablation area¹⁵. Garibaldi, Guilcher oeste and Guilcher este face southward and are, thus, topographically stronger shaded from solar radiation (Fig. S8). Altogether, these characteristics ultimately lead to high snowfall amounts together with reduced surface ablation (Fig. 2c, Fig. S9d), favoring mass gain.

Overall, mass loss in the CDI is primarily controlled by atmospheric conditions. The climatic ablation of the entire CDI on average amounts to $4.20 \pm 1.48 \text{ Gt yr}^{-1}$ (74 % of total ablation) while the frontal ablation on average amounts to $1.44 \pm 0.72 \text{ Gt yr}^{-1}$ (26 % of total ablation). Considering the MALT glaciers only, for almost half of the glaciers the mass loss is mainly controlled by ice dynamics. For six individual glaciers covering around 15 % of the CDI area, the ice-dynamical overtakes the climatical contribution to mass loss by a factor of two. While trends in CMB impose an increasing mass loss associated with climatic changes, frontal ablation remains without significant change over the study period.

Discussion

The climatic conditions during our study period (1999-2022) show strong zonal gradients across the Cordillera Darwin, specifically for precipitation, wind velocities and relative humidity (Fig. S3). These gradients were already reported for the 20th century climate¹¹. The highest precipitation amounts are located in the north-northwestern part of the CDI (Fig. 2a), as previously reported³⁴. With regard to atmospheric trends we confirm that wind velocities and precipitation are increasing over Tierra del Fuego (e.g., ref. ⁶). The warming trend for the beginning of the 21st century found in this study ($+0.42 \text{ }^\circ\text{C}$ per decade) exceeds warming rates reported for the 20th century ($+0.13$ to $+0.20 \text{ }^\circ\text{C}$ per decade)³⁵ as well as the projected warming until mid of this century under a high-emission scenario (RCP8.5) (total warming of $+0.5 \text{ }^\circ\text{C}$ until 2050)³⁶.

The moister, more rich in snow conditions in the south-western part of the CDI have been proposed to cause less thinning and retreat, while the opposite is true for the drier north-eastern part¹. Our results support the fact that climatic conditions force more negative mass balances in the north-eastern part of the CDI compared to the south-west (Fig. 3a). Regional differences are also reflected in equilibrium line

altitudes (ELAs) in the CDI. Simulated ELA values increase from the west (758 m) over the center (810 m) to the east (894 m). At Schiaparelli Glacier in the west, an average ELA of 730 ± 50 m (2000-2017) has been reported²¹ which is in very good agreement with our results (average ELA of 734 m). ELAs reported for Grande Glacier with around 640 ± 200 m¹ to 650 m (for 2011)¹⁷ based on single year end-of-summer snowline altitudes are distinctly lower than our long-term average of 790 m. For Marinelli Glacier, we calculate an average ELA around 802 m for the 22-year study period, which falls in between previous estimates ranging from around 600 m in the year 2000¹² to 1100 m in 2011¹⁷. Melkonian et al.¹ derive an ELA of 650 ± 200 m from single year end-of-summer snowlines at Garibaldi Glacier which is lower than the average ELA of 760 m we found in this study. Discrepancies between end-of-summer snowline and mean long-term ELA are explained by the possibility of snow fall events during the summer months³⁷, which do impose large uncertainties on ELA detected from satellite imagery.

Snowfall and surface melt are determined as the main contributors to the CMB at Schiaparelli Glacier (western CDI) and Martial Este (east of CDI), reflected in a strong correlation with air temperature and precipitation^{21,38}, which is in agreement with our results (Fig. 3a). Similar to our results for the western CDI, the energy input at Schiaparelli Glacier is dominated by incoming radiation and sensible heat flux, while the largest energy losses are attributed to the outgoing radiation and the energy consumed by melting^{21,20}. The importance of the sensible heat flux as an energy source has been highlighted in previous energy balance studies at the Southern Patagonian Icefield³⁹⁻⁴¹ and the Gran Campo Nevado⁴². The pronounced altitudinal mass balance gradient observed in the CDI (Fig. 2c) is a typical characteristic of Patagonian glaciers (e.g., ref. ^{39,41,43}), as is the high inter-annual variability³⁹. Periods of more negative/positive CMB (Fig. 3c) are in agreement with findings of geodetic mass balance from Dussailant et al.¹⁰ reporting stronger losses in the period 2000-2006 compared to the period 2012-2016 for the Fuegian Andes, although these values suffer from poor data coverage in that region (see supplementary material in ref. ¹⁰). Seasonality in the energy fluxes over the CDI (Fig. S6) is in general agreement with findings at Perito Moreno Glacier, located at the Southern Patagonian Icefield⁴¹: The energy input from the sensible heat flux exceeds the input from net shortwave radiation during austral winter, latent heat flux has a minimum value during austral spring, and the glacier (conductive) heat flux peaks during austral winter, when the glacier surface gets cooled by the atmosphere. Schaefer et al.⁴⁰ found latent heat flux as an energy source for two Patagonian glaciers during the ablation season, which confirms our findings for the western region of the CDI, where latent heat flux turns from energy sink to energy source during summer.

A trend analysis over the study period clearly demonstrates an *extremely likely* increase in surface melt. This development causes a *likely* trend towards more negative mass balance. Despite the *likely* positive trend in precipitation amounts, we do not find an increase in snowfall amounts. If these recent trends continue, the CDI will get further out of balance and glaciers will tread a path of accelerating mass loss as already indicated by the current trend in the CMB (Fig. 3c). In general, glaciers that are covering a

lower elevation range (< 600 m) already experience more pronounced mass losses with an average CMB trend of -0.41 m w.e. yr^{-1} per decade (*extremely likely*). At Perito Moreno Glacier, Minowa et al. ⁴¹ found a *virtually certain* decreasing trend in the surface mass balance over the ablation zone of -0.90 ± 0.3 m w.e. yr^{-1} per decade (1996-2020), which is in general agreement with our findings for areas of similar elevation in the CDI (-0.47 m w.e. yr^{-1} per decade).

Frontal ablation is to this day unexplored in the CDI, except for Marinelli Glacier. Koppes et al. ¹⁶ found 0.40 Gt yr^{-1} for the beginning of the 21st century, which is in agreement with our estimates of 0.40 ± 0.08 Gt yr^{-1} by mass budgeting. Frontal ablation at the CDI is in a similar order of magnitude as for the Northern Patagonian Icefield (2.5 ± 0.5 Gt yr^{-1} in 2000-2019), which covers almost twice the area ¹⁵. Alike the Northern Patagonian Icefield, ice-dynamic losses are also channelized through only a few prominent outlet glaciers (Fig. 4). Yet, the fraction of frontal ablation to total ablation (26 %) for the entire CDI is substantial. Considering the MALT glaciers only, this fraction (48 %) is similar to the MALT glaciers of the Southern Patagonian Icefield (48 %) ¹⁵. Apart from these few glaciers, however, mass loss is primarily controlled by atmospheric conditions. Frontal ablation calculated by two different methods in this study overall agrees well (Fig. 4). Discrepancies between the two approaches are within the error ranges for most glaciers. For a few glaciers (e.g., Finlandia, Garibaldi) the flux gate method produces higher frontal ablation, which can be explained by a possible overestimation of ice thickness and/or velocity, or underestimation of modelled CMB, as well as frontal advances that are neglected in the flux gate approach.

Major limitations of this study are the scarce observations, making model calibration and validation a challenge. Combining in-situ and remotely sensed observations, we accomplished to satisfiably constrain the CMB. However, we want to note that most energy fluxes are not directly verifiably due to a lack of observations. Thus, absolute values should be handled with a certain care, while relative comparison and trends are considered more reliable. During model calibration and analysis, we found a strong sensitivity of modelled latent heat flux to relative humidity, a model input variable which is prone to uncertainties. Furthermore, we want to point out that the considerable refreezing rates predicted by the model are a surprising result for a supposedly temperature icefield in a maritime climate setting. Using a 3-hourly timestep, we assume that the sub-daily melt-refreeze cycle is resolved properly. Thus, refreezing rates are expected to be higher than in studies using a daily resolution⁴⁴. Veldhuijsen et al. ⁴⁵ found that reducing the temporal model resolution from hourly to daily causes a strong underestimation of refreezing (84 %). Due to the high amounts of rainfall and melt water, refreezing is expected to be important over parts of the CDI. Our results suggest that about 23 % of percolated water (rainfall plus melt water) refreezes within the snowpack with largest proportion in spring and autumn. These values lie in-between numbers for mid-latitude glaciers (~ 10 %) ^{46,47} and the Antarctic (Peninsula) (~ 70 -95 %) ^{48,49}. To better quantify the refreezing in this region, we recommend the acquisition and investigation of firn cores in future studies. A minor limitation is related to the mass budgeting approach: Geodetic elevation change products cannot measure subaqueous ice losses when glaciers are retreating.

Field Code Changed

However, an assessment of these losses demonstrates that they are distinctly smaller compared to iceberg calving and lie within the reported uncertainty. For Marinelli Glacier, who experienced the strongest retreat over the study period, we estimate a maximum subaqueous ice loss of roughly 0.05 Gt yr^{-1} , constituting about 10 % of the total frontal ablation.

The CDI is one of the largest ice bodies in the Southern Hemisphere. We present the first simulation of the climatic energy and mass balance as well as frontal ablation. Our results allow an attribution of the observed mass loss to climatic or ice-dynamic forcing. Furthermore, these simulations shed light on the climatic imprint of glacier response at a unique location in the higher mid-latitudes of the Southern Hemisphere. Results reveal strong climatic gradients across the CDI that cause regional differences in the energy and mass fluxes. Overall, we show that the CDI has been climatically balanced in the recent two decades, but is entering a state of accelerated mass loss due to increasing surface melt. The melt increase is associated with an intense warming rate that exceeds projections for the early 21st century. The current melt trend is more pronounced at lower elevations. If recent warming trends continue, glaciers in the CDI will follow a trajectory of mass loss acceleration. Frontal ablation accounts for a significant share of the total ablation (26 %), but is only important for a minority of glaciers. These few glaciers dominate the CDI-wide frontal losses of $1.44 \pm 0.72 \text{ Gt yr}^{-1}$. We conclude that climatic warming has reached the CDI glaciers and that atmospheric conditions exert main control on the current glacier evolution in Tierra del Fuego.

Data & Methods

Data

Meteorological observations in the Cordillera Darwin are sparse (Figure 1c) due to the harsh conditions and the inaccessibility of the region. Details of all automatic weather stations (AWSs) used in this study are listed in Table S1. Operators are the Chilean Water Directorate (Dirección General de Aguas, short DGA) or individual researchers installing stations within the framework of different research projects in the Cordillera Darwin. Measured variables range from air temperature via relative humidity, global radiation, wind, air pressure to precipitation (Table S1). However, the station network is located close to sea level, lacking information at higher elevation. This deficit is especially problematic for precipitation, which is strongly influenced by orographic effects. At one AWS (Río Betbeder) a gauging station is installed, providing valuable information on precipitation amounts over the entire river catchment (Fig. 1c). All stations have been quality checked including a screening for outliers or drift in the data. In March 2020, the COrdillera Darwin Ice CorE Survey (CODICES) project drilled a 3.25 m firn core in a flat, northwest-southeast oriented 150 x 150 m saddle (54.6814°S, 69.6394°W, 2324 m a.s.l), one of the highest flat areas in the Cordillera Darwin. Seasonal variability in major ions and insoluble microparticles were used to estimate annual layers. The firn core record extends from March 2020 to austral spring 2016.

Measurements of surface ablation or mass balance are limited in the Cordillera Darwin. Ablation stakes have been installed between 2013 and 2020 at Schiaparelli Glacier located within the Mount Sarmiento Massif at the western edge of the CDI. Stakes are limited to the lowest part of the ablation area, delivering information about surface melt only²⁰. A 21-year long record of annual, winter and summer mass balance exists at the Martial Este Glacier located east of the main body of the CDI⁵⁰. The glacier is located outside of the direct study region but we extended the domain for model validation.

We use atmospheric variables from the ERA5 reanalysis product (the latest global product of the European Centre for Medium-Range Weather Forecasts, ECMWF) to generate the climatic forcing for the COSIPY model over the study site (54.25-55.00°S, 71.00-68.25°W). ERA5 provides high temporal (hourly) and spatial (31x31 km) resolution⁵¹. For Southern Patagonia, ERA5 and its previous versions have proven reliable in several modeling studies (e.g., ref. 20,21,31,52-54). Required variables comprise air temperature, relative humidity, air pressure, wind speed, cloud cover fraction and total precipitation over the study site. For downscaling of precipitation, we need upstream (53.75-55.50°S, 74.50-73.25°W) information about air temperature, relative humidity, wind vectors and geopotential height between 850 and 500 hPa.

Glacier outlines are taken from the DGA glacier inventory^{55,56} since comparison with the Randolph Glacier Inventory (V6) has revealed strong weaknesses of the latter for the southern Andes, especially for smaller glaciers⁵⁷. Glacier outlines available comprise two time stamps: 2000-2003 (in the following

denoted as 2000) and 2019. Since glacier catchments in the more recent inventory had been partially upgraded, catchments in the 2000 inventory had to be homogenized with the 2019 inventory for consistency. Furthermore, it was shown that neglecting the temporal evolution of the glacial extent by relying on constant glacier outlines can bias the comparison between surface and geodetic mass balance^{58,59}. Since elevation changes cover the period 2000 to 2013, we manually produced outlines for 2013, which have been missing so far. Thus, inventories available in this study cover years 2000, 2013 and 2019 oriented on inventory availability and geodetic data.

Geodetic mass changes of glaciers of the Cordillera Darwin are derived from interferometric Synthetic Aperture Radar (SAR) digital elevation models (DEMs) of the Shuttle Radar Topography Mission (SRTM) of the National Aeronautics and Space Administration (NASA) and the TerraSAR-X add-on for Digital Elevation Measurement satellite mission (TanDEM-X) of the German Aerospace Center (DLR). The SRTM C-band DEM has been acquired in February 2000 at a spatial resolution of 1 arcsec⁶⁰. We use the void-filled LP DAAC NASA SRTM DEM⁶¹. Bistatic SAR acquisitions of the TanDEM-X mission are available since 2011⁶². Here, we use Co-registered Single look Slant range Complex (CoSSC) data of the Southern Hemisphere ablation periods of the years 2012-2014. To further minimize elevation offsets due to differences in ice accumulation or time-varying depths of SAR signal penetration into the glacier volume, we use TanDEM-X acquisition dates which are close to the mean SRTM acquisition date (2000-02-16) whenever possible.

Ice flow velocities and reconstructed ice thickness had been taken from Millan et al.³. These datasets are used for calculation of frontal ablation based on a flux gate approach. Furthermore, surface velocities close to the glacier fronts are used for a classification of non-calving glaciers. A reliable classification is essential for a comparison between climatic and geodetic mass balance. Glaciers exceeding a velocity threshold of 65 m yr⁻¹ are classified as marine- or lake-terminating (MALT) glaciers, glaciers below the threshold velocity are classified as glaciers without significant frontal ablation and treated as land-terminating in this study.

Atmospheric forcing

Atmospheric input data required for the COSIPY simulation includes air temperature, relative humidity, incoming shortwave radiation, wind speed, air pressure, cloud cover and precipitation. To extend the climatic data beyond the respective measurement periods, we apply a downscaling scheme where we combine statistical downscaling with the application of a radiation model and a model of orographic precipitation.

Following previous studies in southern Patagonia (e.g., ref. ^{20,21,39}), we apply quantile mapping for statistical downscaling of air temperature, relative humidity and air pressure. Quantile mapping is a method of statistical bias correction, where the cumulative distribution function of the model is transferred to the cumulative distribution function of the observation^{27,63}. Statistically downscaled air

temperature and pressure are adjusted to sea level conditions, interpolated between the available station points via Ordinary Kriging ⁶⁴, and subsequently spatially extrapolated over the topography using a linear temperature lapse rate of $-0.6 \text{ K}/100 \text{ m}$ ²⁰ and the barometric equation, respectively. Relative humidity is likewise interpolated between the recording AWSs with Ordinary Kriging. Wind speed and cloud cover fraction are taken directly from ERA5 and interpolated to the model resolution.

A radiation model ⁶⁵ is applied over the study site to calculate global radiation over the glacier surface, following the methodology of Temme et al. ²⁰ at the Mount Sarmiento Massif. The model calculates both the direct and diffuse component of the solar radiation based on cloud cover, temperature, humidity and pressure. Corrections are applied for the slope and aspect of the respective grid cell. Shaded grid cells, either from the terrain or self-shaded, exclusively receive the diffuse solar radiation component ^{28,65}.

Due to the small-scale and episodic character of precipitation events, statistical techniques often fail to infer reliable distributions over complex terrain from coarse global data sets. Furthermore, strong winds limit the reliability of observations in southern Patagonia ^{42,66}. An orographic precipitation model showed improved performance as compared to extrapolation of observational data using altitudinal lapse rates ⁶⁷. The model calculates the orographic portion of precipitation resulting from forced orographic uplift over a mountain ³⁹. It is grounded on the linear steady-state theory of orographic precipitation, considering airflow dynamics, cloud timescales and processes of advection and downslope evaporation ^{29,30}. Since the model assumes stable and saturated conditions with unblocked air flow crossing the CDI from west to east ^{29,39}, time intervals that do not fulfill these constraints are excluded. Thresholds and parameter settings are taken from Temme et al. ²⁰. The total precipitation is calculated by adding the orographic precipitation calculated in the model to the large-scale precipitation, which is obtained by removing the orographic component from the ERA5 precipitation. Recent elevation- and bias-corrected precipitation products (W5E5, WFDE5) with lower spatial resolution and shorter temporal coverage indicate, an ERA5 overestimation over Tierra del Fuego ⁶⁸. Comparison of ERA5 daily precipitation with observations supports this finding ⁶⁹. To guarantee that the simulated total precipitation at the AWS locations agrees with the observed amounts, we constrain the large-scale precipitation from ERA5 to the annual measurements.

To derive snowfall from precipitation, a logistic transfer function is applied scaling around a threshold temperature of $1.0 \text{ }^\circ\text{C}$. A snow drift parametrization is included in the modelling framework to account for snow redistribution caused by the strong westerly winds over the CDI. Locations sheltered from or exposed to wind are identified by a topographic analysis and solid precipitation is redistributed accordingly ⁷⁰. Parameters and adjustments to the model are transferred from the Mount Sarmiento Massif ²⁰.

Climatic forcing data are validated on a daily basis with meteorological observations from AWSs that have not been used in the downscaling (Table S1) based on a statistical analysis of mean model bias,

Formatted: Spanish (Chile)

Field Code Changed

root mean square error and correlation. Overall, the performance of downscaled and modelled climate variables is satisfying (Table S2). The agreement of downscaled variables with measurements is improved compared to the raw ERA5 input, confirming the success of the downscaling approach. For further information on precipitation amounts, we compare annual precipitation over the river catchment of Río Betbeder with observed stream flow at the gauging station there, and snowfall with results of a firn core in the central CDI. The former river catchment is located in the northeast of the CDI covering a total area of 146 km² (Fig. 1c). The comparison indicates an underestimation (Fig. S2). However, considering the simplified approach (e.g., neglecting water storage and glaciers in the system), results are satisfying. The firn core site is located at an exposed saddle where we assume important wind erosion. The small-scale local wind field and snowdrift are, however, not fully resolved in our modeling approach, which explains an overestimation in the modelled snowfall.

COSIPY model

The open-source ‘COupled Snowpack and Ice surface energy and mass balance model in PYthon’ (COSIPY) ²⁶ is a physically based model grounding on the concept of energy and mass conservation. It couples a surface energy and mass balance model with a multi-layer subsurface snow and ice model, with the calculated surface meltwater serving as input to the subsurface model ²⁶. The energy balance model solves all energy fluxes F at the glacier surface:

$$F = SW_{in}(1 - \alpha) + LW_{in} + LW_{out} + Q_{sen} + Q_{lat} + Q_g + Q_{RRR}$$

where SW_{in} is the incoming shortwave radiation taken from the radiation model, α is the surface albedo, LW_{in} and LW_{out} are the incoming and outgoing longwave radiation, Q_{sen} and Q_{lat} are the turbulent sensible and latent heat flux, Q_g is the glacier heat flux and Q_{RRR} the rain heat flux. Melt can occur if the surface temperature is at the melting point (0.0 °C) and F is positive. Under this condition, the available energy for surface melt Q_M equals F . Rain and meltwater can percolate the snowpack and cause refreezing in the snow layers. Subsurface melting is possible by penetration of shortwave radiation in the upper snow layers. Solving the surface plus the internal mass balance in the snowpack, COSIPY gives the climatic mass balance (CMB) ⁷¹. The total ablation includes surface melting, sublimation and subsurface melting. Accumulation is the sum of snowfall, deposition and refreezing.

Albedo values are differentiated between snow, firn and ice surfaces. The decay of surface albedo due to snow aging is parameterized following the scheme of Oerlemans and Knap ⁷². The albedo depends on the time since the last snowfall and the snow depth. A bulk approach is applied to parameterize the turbulent heat fluxes. COSIPY offers the option to correct the flux-profile relationship by a stability correction using the Richardson-Number or the Monin-Obukhov similarity theory ²⁶.

We apply COSIPY version 1.4 in the period 04/1999-03/2022 with a 200 m spatial and a 3-hourly temporal resolution. All parameter settings follow the COSIPY set-up in the Mount Sarmiento Massif ²⁰ and are summarized in Table S5. The model performance of COSIPY was positively evaluated in the

Mount Sarmiento Massif, where four surface mass balance models of varying complexity were compared. COSIPY results agreed well with the other models as well as with observations of ablation stakes and geodetic mass balance ²⁰.

Geodetic mass balance processing

Elevation changes are calculated by DEM-differencing of SRTM and TanDEM-X. TanDEM-X DEMs are created based on differential interferometry following an established workflow ⁹. First, interferograms are computed from concatenated overlapping acquisitions, phase-unwrapped based on a minimum cost flow algorithm and converted to elevation values using the SRTM DEM as reference surface. Thereafter, the 'raw' TanDEM-X DEMs are iteratively co-registered to the SRTM DEM in the vertical and horizontal plane. Therefore, the 3D offset of each DEM is estimated based on all stable terrain with less than 25° surface slope excluding water and glacier areas. Finally, a regional elevation mosaic is created by merging all co-registered DEMs in the order of the relative deviation between the SRTM mean acquisition date (February 16th) and the tile-specific TanDEM-X date. The cell-specific TanDEM-X dates are stored with the DEM mosaic and subsequently used to calculate the respective elevation change rate during the SRTM and TanDEM-X DEM-differencing. The mean regional observation period of the elevation change rate measurement is 12.97 years.

To extract glacier-specific mass changes within the geodetic observation period, the elevation change map is masked to the glacier outlines of the 2000 inventory (see Data section). The mean elevation change rate is extracted for each glacier geometry and converted to volume and mass change based on the respective glacierized area and an approximate ice density of 900 ± 60 kg m⁻³. Since glacier area changes during the observation period can bias the derived mass budgets ^{73,74}, the specific mass change rate of each glacier is calculated using the mean glacier area of the 2000 and 2013 inventories following the UNESCO definitions ⁷¹.

Model calibration and validation

Due to the limited in-situ observations in the Cordillera Darwin, calibration and validation of the CMB are a major challenge. With the large model domain and the high temporal and spatial resolution, resulting in a massive computational effort, intense model calibration is not feasible. Instead, the downscaling procedure and optimal parameter setting are grounded on the expertise gained at the Mount Sarmiento Massif, located at the western edge of the CDI ²⁰. Sensitivity runs are applied for further optimization. Temme et al. ²⁴ conclude that calibrating against regional satellite observations of mass change significantly improves the performance of CMB models. Following this approach, we rank the sensitivity runs based on the highest agreement with regional specific mass balance, as observed with satellite remote sensing, for all glaciers with no frontal ablation. The highest ranked run is presented in this study.

For model validation, we compare the climatic with the geodetic mass balance of each land-terminating glacier on a catchment level (catchment information not used during calibration). MALT glaciers are excluded because they also lose mass at the calving front due to ice dynamics. To reduce uncertainties, we limit the comparison to glaciers exceeding an area of 3 km². This gives a validation dataset of glaciers covering ~37% of the glaciated area of the CDI. Model performance is quantified by the root mean square error between the glacier-specific climatic and geodetic mass balance for the individual catchments. Stake measurements at Schiaparelli Glacier²⁰ and Martial Este Glacier⁷⁵ serve as additional validation of melt on the western and eastern edges of the CDI (Table S6).

Frontal ablation

In this study, we apply two different methods to determine frontal ablation for the entire CDI and the individual marine- and lake-terminating glaciers in the Cordillera Darwin. Firstly, we apply a mass budgeting, where the residual of the total glacier mass balance (ΔM_{tot}) from geodetic observations and the CMB (\dot{B}) simulated with COSIPY provides the frontal ablation (A_f): $A_f = \Delta M_{tot} - \dot{B}$ ¹⁸. Uncertainties of the glacier-specific CMB, constrained by model validation, directly translate into the uncertainties of frontal ablation estimations.

Secondly, we apply a flux gate approach^{15,23}. Here, frontal ablation is calculated based on the discharge (D) at a flux gate located upstream of the glacier front and the CMB downstream of the flux gate (\dot{B}_{FG}): $A_f = -D - \dot{B}_{FG}$. D is calculated by integrating the product of ice thickness and surface velocity perpendicular to the gate. For our study site, the lack of ice thickness measurements in the CDI makes ice thickness highly speculative. This directly translates into elevated uncertainties for frontal ablation.

Data availability

Average annual fields of the simulation results are available via this/is/a/dummy/link. Temporally higher resolved model data is available from the corresponding author on request.

References

1. Melkonian, A. K. *et al.* Satellite-derived volume loss rates and glacier speeds for the Cordillera Darwin Icefield, Chile. *The Cryosphere* **7**, 823–839 (2013).
2. Farinotti, D. *et al.* A consensus estimate for the ice thickness distribution of all glaciers on Earth. *Nature Geoscience* **12**, 168–173 (2019).
3. Millan, R., Mougintot, J., Rabatel, A. & Morlighem, M. Ice velocity and thickness of the world's glaciers. *Nature Geoscience* **15**, 124–129 (2022).
4. Izagirre, E. *et al.* The glacial geomorphology of the Cordillera Darwin Icefield, Tierra del Fuego, southernmost South America. *Journal of Maps* **20**, 2378000 (2024).
5. Garreaud, R. D., Vuille, M., Compagnucci, R. & Marengo, J. Present-day South American climate. *Palaeogeography, Palaeoclimatology, Palaeoecology* **281**, 180–195 (2009).
6. Garreaud, R., Lopez, P., Minvielle, M. & Rojas, M. Large-scale control on the Patagonian climate. *Journal of Climate* **26**, 215–230 (2013).
7. Goyal, R., Sen Gupta, A., Jucker, M. & England, M. H. Historical and Projected Changes in the Southern Hemisphere Surface Westerlies. *Geophysical Research Letters* **48**, e2020GL090849 (2021).
8. Garreaud, R. D., Clem, K. & Veloso, J. V. The South Pacific Pressure Trend Dipole and the Southern Blob. *Journal of Climate* **34**, 7661–7676 (2021).
9. Braun, M. H. *et al.* Constraining glacier elevation and mass changes in South America. *Nature Climate Change* **9**, 130–136 (2019).
10. Dussailant, I. *et al.* Two decades of glacier mass loss along the Andes. *Nature Geoscience* **12**, 802–808 (2019).
11. Holmlund, P. & Fuenzalida, H. Anomalous glacier responses to 20th century climatic changes in Darwin Cordillera, southern Chile. *J. Glaciol.* **41**, 465–473 (1995).
12. Porter, C. & Santana, A. Rapid 20th century retreat of Ventisquero Marinelli in the Cordillera Darwin Icefield. *Anales del Instituto de la Patagonia* **31**, 17–26 (2003).
13. Lopez, P. *et al.* A regional view of fluctuations in glacier length in southern South America. *Global and Planetary Change* **71**, 85–108 (2010).
14. Meier, W. *et al.* Late Holocene Glacial Fluctuations of Schiaparelli Glacier at Monte Sarmiento Massif, Tierra del Fuego (54°24'S). *Geosciences* **9**, 340 (2019).
15. Minowa, M., Schaefer, M., Sugiyama, S., Sakakibara, D. & Skvarca, P. Frontal ablation and mass loss of the Patagonian icefields. *Earth and Planetary Science Letters* **561**, 116811 (2021).
16. Koppes, M., Hallet, B. & Anderson, J. Synchronous acceleration of ice loss and glacial erosion, Glaciar Marinelli, Chilean Tierra del Fuego. *Journal of Glaciology* **55**, 207–220 (2009).
17. Bown, F., Rivera, A., Zenteno, P., Bravo, C. & Cawkwell, F. First Glacier Inventory and Recent Glacier Variation on Isla Grande de Tierra Del Fuego and Adjacent Islands in Southern Chile. in *Global Land Ice Measurements from Space* (eds. Kargel, J. S., Leonard, G. J., Bishop, M. P., Käab, A. & Raup, B. H.) 661–674 (Springer Berlin Heidelberg, Berlin, Heidelberg, 2014). doi:10.1007/978-3-540-79818-7_28.
18. Schaefer, M., Machguth, H., Falvey, M. & Casassa, G. Modeling past and future surface mass balance of the Northern Patagonia Icefield. *Journal of Geophysical Research: Earth Surface* **118**, 571–588 (2013).
19. Buttstädt, M., Möller, M., Iturraspe, R. & Schneider, C. Mass balance evolution of Martial Este Glacier, Tierra del Fuego (Argentina) for the period 1960–2009. *Advances in Geosciences* **22**, 117–124 (2009).
20. Temme, F. *et al.* Strategies for regional modeling of surface mass balance at the Monte Sarmiento Massif, Tierra del Fuego. *The Cryosphere* **17**, 2343–2365 (2023).
21. Weidemann, S. S. *et al.* Recent Climatic Mass Balance of the Schiaparelli Glacier at the Monte Sarmiento Massif and Reconstruction of Little Ice Age Climate by Simulating Steady-State Glacier Conditions. *Geosciences* **10**, 272 (2020).
22. Hugonnet, R. *et al.* Accelerated global glacier mass loss in the early twenty-first century. *Nature* **592**, 726–731 (2021).
23. Fürst, J. J. *et al.* The foundations of the Patagonian icefields. *Commun Earth Environ* **5**, 142 (2024).
24. Temme, F. *et al.* Strategies for Regional Modelling of Surface Mass Balance at the Monte

Formatted: Spanish (Chile)

Sarmiento Massif, Tierra Del Fuego.
<https://egusphere.copernicus.org/preprints/2022/egusphere-2022-1036/> (2022)
doi:10.5194/egusphere-2022-1036.

25. Langhamer, L. *et al.* Response of lacustrine glacier dynamics to atmospheric forcing in the Cordillera Darwin. *J. Glaciol.* 1–19 (2024) doi:10.1017/jog.2024.14.
26. Sauter, T., Arndt, A. & Schneider, C. COSIPY v1.3 – an open-source coupled snowpack and ice surface energy and mass balance model. *Geoscientific Model Development* **13**, 5645–5662 (2020).
27. Gudmundsson, L., Bremnes, J. B., Haugen, J. E. & Engen-Skaugen, T. Technical Note: Downscaling RCM precipitation to the station scale using statistical transformations – A comparison of methods. *Hydrology and Earth System Sciences* **16**, 3383–3390 (2012).
28. Mól, T., Cullen, N. J., Hardy, D. R., Winkler, M. & Kaser, G. Quantifying climate change in the tropical midtroposphere over East Africa from glacier shrinkage on Kilimanjaro. *Journal of Climate* **22**, 4162–4181 (2009).
29. Smith, R. B. & Barstad, I. A linear theory of orographic precipitation. *Journal of the Atmospheric Sciences* **61**, 1377–1391 (2004).
30. Barstad, I. & Smith, R. B. Evaluation of an orographic precipitation model. *Journal of Hydrometeorology* **6**, 85–99 (2005).
31. Sauter, T. Revisiting extreme precipitation amounts over southern South America and implications for the Patagonian Icefields. *Hydrology and Earth System Sciences* **24**, 2003–2016 (2020).
32. IPCC. *The Ocean and Cryosphere in a Changing Climate: Special Report of the Intergovernmental Panel on Climate Change.* (Cambridge University Press, 2022). doi:10.1017/9781009157964.
33. Garreaud, R. D. The Andes climate and weather. *Advances in Geosciences* **22**, 3–11 (2009).
34. Fernandez, R. A., Anderson, J. B., Wellner, J. S. & Hallet, B. Timescale dependence of glacial erosion rates: A case study of Marinelli Glacier, Cordillera Darwin, southern Patagonia. *J. Geophys. Res.* **116**, F01020 (2011).
35. Rosenblüth, B., Fuenzalida, H. A. & Aceituno, P. RECENT TEMPERATURE VARIATIONS IN SOUTHERN SOUTH AMERICA. *Int. J. Climatol.* **17**, 67–85 (1997).
36. Giesecke, R. *et al.* General Hydrography of the Beagle Channel, a Subantarctic Inter-oceanic Passage at the Southern Tip of South America. *Front. Mar. Sci.* **8**, 621822 (2021).
37. Bravo, C., Bozkurt, D., Ross, A. N. & Quincey, D. J. Projected increases in surface melt and ice loss for the Northern and Southern Patagonian Icefields. *Sci Rep* **11**, 16847 (2021).
38. Mutz, S. G. & Aschauer, J. Empirical glacier mass-balance models for South America. *J. Glaciol.* 1–15 (2022) doi:10.1017/jog.2022.6.
39. Weidemann, S. S. *et al.* Glacier Mass Changes of Lake-Terminating Grey and Tyndall Glaciers at the Southern Patagonia Icefield Derived From Geodetic Observations and Energy and Mass Balance Modeling. *Frontiers in Earth Science* **6**, 1–16 (2018).
40. Schaefer, M., Fonseca-Gallardo, D., Fariás-Barahona, D. & Casassa, G. Surface energy fluxes on Chilean glaciers: Measurements and models. *The Cryosphere* **14**, 2545–2565 (2020).
41. Minowa, M., Skvarca, P. & Fujita, K. Climate and Surface Mass Balance at Glaciar Perito Moreno, Southern Patagonia. *Journal of Climate* **36**, 625–641 (2023).
42. Schneider, C., Kilian, R. & Glaser, M. Energy balance in the ablation zone during the summer season at the Gran Campo Nevado Ice Cap in the Southern Andes. *Global and Planetary Change* **59**, 175–188 (2007).
43. Schaefer, M., MacHuguth, H., Falvey, M., Casassa, G. & Rignot, E. Quantifying mass balance processes on the Southern Patagonia Icefield. *The Cryosphere* **9**, 25–35 (2015).
44. Arndt, A. & Schneider, C. Spatial pattern of glacier mass balance sensitivity to atmospheric forcing in High Mountain Asia. *J. Glaciol.* 1–18 (2023) doi:10.1017/jog.2023.46.
45. Veldhuijsen, S. B. M. *et al.* Spatial and temporal patterns of snowmelt refreezing in a Himalayan catchment. *J. Glaciol.* **68**, 369–389 (2022).
46. Krampe, D., Arndt, A. & Schneider, C. Energy and glacier mass balance of Fürkeleferner, Italy: past, present, and future. *Front. Earth Sci.* **10**, 814027 (2022).
47. Abraham, B. N., Cullen, N. J., Conway, J. P. & Sirguey, P. Applying a distributed mass-balance model to identify uncertainties in glaciological mass balance on Brewster Glacier, New Zealand.

- J. Glaciol.* **69**, 1030–1046 (2023).
48. Van Wessem, J. M. *et al.* Modelling the climate and surface mass balance of polar ice sheets using RACMO2 – Part 2: Antarctica (1979–2016). *The Cryosphere* **12**, 1479–1498 (2018).
 49. Hansen, N. *et al.* Downscaled surface mass balance in Antarctica: impacts of subsurface processes and large-scale atmospheric circulation. *The Cryosphere* **15**, 4315–4333 (2021).
 50. WGMS. FLUCTUATIONS OF GLACIERS DATABASE. 35 MB World Glacier Monitoring Service (WGMS) <https://doi.org/10.5904/WGMS-FOG-2023-09> (2023).
 51. Hersbach, H. *et al.* The ERA5 global reanalysis. *Quarterly Journal of the Royal Meteorological Society* **146**, 1999–2049 (2020).
 52. Lenaerts, J. T. M. *et al.* Extreme precipitation and climate gradients in Patagonia revealed by high-resolution regional atmospheric climate modeling. *Journal of Climate* **27**, 4607–4621 (2014).
 53. Bravo, C. *et al.* Assessing snow accumulation patterns and changes on the Patagonian Icefields. *Frontiers in Environmental Science* **7**, 1–18 (2019).
 54. Temme, F., Turton, J. V., Mölg, T. & Sauter, T. Flow regimes and Föhn types characterize the local climate of Southern Patagonia. *Atmosphere* **11**, (2020).
 55. Barcaza, G. *et al.* Glacier inventory and recent glacier variations in the Andes of Chile, South America. *Ann. Glaciol.* **58**, 166–180 (2017).
 56. DGA. Metodología de inventario público de glaciares, SDT No. 447, Ministerio de Obras Públicas, Dirección General de Aguas Unidad de Glaciología y Nieves, realizado por: Casassa, G., Espinoza, A., Segovia, A., and Huenante, J. (2022).
 57. Zalazar, L. *et al.* Spatial distribution and characteristics of Andean ice masses in Argentina: results from the first National Glacier Inventory. *J. Glaciol.* **66**, 938–949 (2020).
 58. Elsberg, D. H., Harrison, W. D., Echelmeyer, K. A. & Krimmel, R. M. Quantifying the effects of climate and surface change on glacier mass balance. *J. Glaciol.* **47**, 649–658 (2001).
 59. Mukherjee, K. *et al.* Evaluation of surface mass-balance records using geodetic data and physically-based modelling, Place and Peyto glaciers, western Canada. *J. Glaciol.* **69**, 665–682 (2023).
 60. Farr, T. G. *et al.* The Shuttle Radar Topography Mission. *Reviews of Geophysics* **45**, 2005RG000183 (2007).
 61. Earth Resources Observation And Science (EROS) Center. Shuttle Radar Topography Mission (SRTM) 1 Arc-Second Global. U.S. Geological Survey <https://doi.org/10.5066/F7PR7TFT> (2017).
 62. Zink, M. *et al.* TanDEM-X mission status: The complete new topography of the Earth. in *2016 IEEE International Geoscience and Remote Sensing Symposium (IGARSS)* 317–320 (IEEE, Beijing, 2016). doi:10.1109/IGARSS.2016.7729075.
 63. Cannon, A. J., Sobie, S. R. & Murdock, T. Q. Bias correction of GCM precipitation by quantile mapping: How well do methods preserve changes in quantiles and extremes? *Journal of Climate* **28**, 6938–6959 (2015).
 64. Murphy, B., Yurchak, R. & Müller, S. GeoStat-Framework/PyKrige: v1.7.0. Zenodo <https://doi.org/10.5281/ZENODO.7008206> (2022).
 65. Mölg, T., Cullen, N. J. & Kaser, G. Solar radiation, cloudiness and longwave radiation over low-latitude glaciers: Implications for mass-balance modelling. *Journal of Glaciology* **55**, 292–302 (2009).
 66. Schneider, C. *et al.* Weather Observations Across the Southern Andes at 53°S. *Physical Geography* **24**, 97–119 (2003).
 67. Jarosch, A. H., Anslow, F. S. & Clarke, G. K. C. High-resolution precipitation and temperature downscaling for glacier models. *Climate Dynamics* **38**, 391–409 (2012).
 68. Cucchi, M. *et al.* WFDE5: bias-adjusted ERA5 reanalysis data for impact studies. *Earth Syst. Sci. Data* **12**, 2097–2120 (2020).
 69. Lavers, D. A., Simmons, A., Vamborg, F. & Rodwell, M. J. An evaluation of ERA5 precipitation for climate monitoring. *Quart J Royal Meteor Soc* **148**, 3152–3165 (2022).
 70. Warscher, M. *et al.* Performance of complex snow cover descriptions in a distributed hydrological model system: A case study for the high Alpine terrain of the Berchtesgaden Alps. *Water Resources Research* **49**, 2619–2637 (2013).
 71. Cogley, J. C. *et al.* Glossary of Glacier Mass Balance and Related Terms. *IACS Contribution*

Formatted: Spanish (Chile)

No. 2 (2011).

72. Oerlemans, J. & Knap, W. H. A 1 year record of global radiation and albedo in the ablation zone of Morteratschgletscher, Switzerland. *Journal of Glaciology* **44**, 231–238 (1998).
73. Florentine, C., Sass, L., McNeil, C., Baker, E. & O'Neel, S. How to handle glacier area change in geodetic mass balance. *J. Glaciol.* 1–7 (2023) doi:10.1017/jog.2023.86.
74. Sommer, C. *et al.* Rapid glacier retreat and downwasting throughout the European Alps in the early 21st century. *Nat Commun* **11**, 3209 (2020).
75. Strelin, J. & Iturraspe, R. Recent evolution and mass balance of Cordón Martial glaciers, Cordillera Fuegoina Oriental. *Global and Planetary Change* **59**, 17–26 (2007).

Acknowledgements

FT was funded by the German Research Foundation (DFG) within the MAGIC project (FU 1032/5-1) and the RESPONSE project (TA 1719/2-1). JJF has received funding from the European Union's Horizon 2020 research and innovation programme via the European Research Council (ERC) as a Starting Grant (StG) under grant agreement No 948290. JA has received funding by the CNPq project (308831/2022-5) and the FAPERGS project (21/2551-0002034-2). EI was funded by the University of the Basque Country (UPV/EHU) under Grant PIF17/182 and by the Basque Government under the Consolidated Research Group IT1029-16 and IT1678-22. The authors are grateful for the scientific support and resources provided by the Erlangen National High Performance Computing (HPC) Center (NHR@FAU) of the Friedrich-Alexander-Universität ErlangenNürnberg (FAU). NHR funding is provided by federal and Bavarian state authorities. NHR@FAU hardware is partially funded by the DFG – 440719683. TanDEM-X data were kindly provided free of charge by the German Aerospace Center (DLR) under AO mabra_XTI_GLAC0264. The authors want to thank the Chilean National Forest Corporation (CONAF) for enabling and supporting the field work in the Cordillera Darwin, Parque Nacional Alberto de Agostini.

Author contribution

The concept of this study was developed by JJF, FT and MS. FT implemented the simulations with the support of JJF. In situ observational data were collected and provided by EI, RG, RJ, JAN, IG and DT. Elevation change rates and geodetic mass balances were processed by CS. FT led the writing process with the support of all the authors.

Competing interests

The contact author has declared that none of the authors has any competing interests.

Figure 1: Climatic mass balance of the Cordillera Darwin Icefield (CDI). *a* Overview panel of the study site. *b* Subregions of the CDI defined within this study. *c* Specific climatic mass balance (color scheme), termination type of the glaciers (outline style, outlines mark 2000 extent), observations from automatic weather stations and ablation stakes in the region, and specific mass fluxes from accumulation, ablation and calving (2000-2013) for selected glaciers. Triangles mark Mount Shipton (2568 m) and Mount Sarmiento (2207 m). The river catchment of Río Betbeder (eastern edge) is shown in yellow.

Figure 2: Climatological and mass and energy balance characteristics. Panels show mean annual **a** precipitation, **b** wind speed, **c** climatic mass balance and **d** sensible heat flux over the CDI (2000-2022). Black outlines display the glacier extent in 2000 (Barcaza et al. 2017).

Figure 3: Climatic energy and mass fluxes together with the resulting mass balance. CDI-wide (land-, lake- and marine-terminating glaciers) average annual climatic **a** mass and **b** energy balance components for the four subdomains: Snowfall (SNOW), deposition (DEPO), refreezing (REFR), surface melt (surfM), subsurface melt (subM), sublimation (SUBL), net shortwave radiation (SWnet), sensible (H), latent (LE) and glacier heat flux (B), heat flux from rain (QRR) and net longwave radiation (LWnet). The black diamonds give the resulting **a** climatic mass balance and **b** energy available for melting, respectively. The lower panel **c** displays the CDI-wide annual average climatic mass balance with shading indicating positive (blue) and negative (red) years. Dashed lines give 5-year averages.

Figure 4: Frontal ablation. Mean annual frontal ablation for marine- and lake-terminating glaciers, calculated with a mass budgeting approach (2000-2013) (dark green) and a flux gate approach (2013) (light green). The respective uncertainty is given in grey caps.

Climate's firm grip on glacier melting in the Cordillera Darwin Icefield, Tierra Del Fuego

Franziska Temme^{1*}, Christian Sommer¹, Marius Schaefer², Ricardo Jaña³, Jorge Arigony-Neto⁴, Inti Gonzalez^{5,6}, Eñaut Izagirre^{7,8}, Ricardo Giesecke^{9,10}, Dieter Tetzner¹¹, Johannes J. Fürst¹

¹Institut für Geographie, Friedrich-Alexander-Universität Erlangen-Nürnberg, Erlangen, Germany

²Instituto de Ciencias Físicas y Matemáticas, Universidad Austral de Chile, Valdivia, Chile

³Departamento Científico, Instituto Antártico Chileno, Punta Arenas, Chile

⁴Instituto de Oceanografía, Universidade Federal do Rio Grande, Rio Grande, Brazil

⁵Centro de Estudios del Cuaternario de Fuego-Patagonia y Antártica, Punta Arenas, Chile

⁶Programa Doctorado Ciencias Antárticas y Subantárticas, Universidad de Magallanes, Punta Arenas, Chile

⁷Hydro-Environmental Processes Research Group, University of the Basque Country UPV/EHU, Leioa, Spain

⁸Basque Centre for Climate Change BC3, Leioa, Spain

⁹Instituto de Ciencias Marinas y Limnológicas, Universidad Austral de Chile, Valdivia, Chile

¹⁰Centro FONDAP de Investigación en Dinámica de Ecosistemas Marinos de Altas Latitudes (IDEAL), Valdivia, Chile

¹¹Ice Dynamics and Paleoclimate, British Antarctic Survey, Cambridge, UK

*corresponding author: franziska.temme@fau.de

Abstract

The Cordillera Darwin Icefield (CDI) in Tierra del Fuego is one of the largest ice bodies in the Southern Hemisphere. We present the first simulation of the climatic energy and mass balance of its glaciers, which are sensitive indicators of climatic changes in the higher mid-latitudes of the Southern Hemisphere. Conditions are characterized by year-round westerly winds causing strong climatic gradients across the cordillera, which are reflected in the climatic energy and mass fluxes. Our results reveal a significant increase in surface melt ($+0.18 \text{ m w.e. yr}^{-1}$ per decade) over the past two decades. We also present the first estimate of dynamically controlled mass loss into adjacent fjords and lakes by frontal ablation. It amounts to $1.44 \pm 0.72 \text{ Gt yr}^{-1}$, accounting for 26 % of the total mass loss in the CDI. Frontal losses are mainly channelized through few marine-terminating glaciers. We conclude that while ice dynamics are important for predicting the fate of individual glaciers, for the CDI as a whole, atmospheric conditions exert the main control on the current glacier evolution.

1. Introduction

The Cordillera Darwin Icefield (CDI) is one of the largest icefields in the Southern Hemisphere ¹, holding a substantial mass of ice that is at least twice as large as the mass of all glaciers in the European Alps ^{2,3}. The main continuous icefield covers the Cordillera Darwin mountain range and is extended by few smaller adjacent ice bodies separated by fjords (Fig. 1), such as the Mount Sarmiento Massif in the west, summing up to a total glaciated area of 2356 km² in 2022 ⁴. Glaciers in the CDI descend from up to 2500 m a.s.l. down to sea level. The large altitudinal range is possible due to the extraordinary location and the extreme climatic conditions. Tierra del Fuego, located at the southernmost end of South America (Fig. 1a), is the closest land mass to Antarctica. Being situated between the subtropical anticyclone and the subpolar low-pressure trough, the area faces strong, year-round westerly winds. Within this so-called storm track, frontal systems continuously transport moist air masses towards the continent ⁵. Orographic uplift causes abundant precipitation along the western slopes of the Cordillera Darwin while lee-side effects result in more arid conditions in the east ⁶. The strength and position of the Southern Hemisphere westerlies impacts not only on the formation of clouds and precipitation but also on the global ocean circulation ⁷. In the past few decades, the storm track has shifted poleward due to an intensification of the subtropical high in the southeast Pacific which is partly ascribed to human induced climate change ⁸. The southward shifting together with and intensification of the westerlies is projected to continue until the end of the 21st century based on high emission scenarios ⁷. Southern Patagonia is the only continental land mass disrupting the Southern Hemisphere westerly wind belt. Since glaciers are susceptible indicators of climate change, the glacier evolution of the CDI provides valuable insights into climatic changes in this region.

In the last decades, the CDI experienced strong ice loss ^{1,9,10}, contributing about 5 % of the total loss in South America between 2000 and 2011/2014 ⁹. Estimates of mass loss rates for Tierra del Fuego range between 1.02 ± 0.11 Gt yr⁻¹ (2000-2011/14) ⁹ and 1.9 ± 1.1 Gt yr⁻¹ (2000-2018) ¹⁰. Despite the general retreat pattern, individual glaciers are stable or even advancing, mostly in the central region of the CDI ¹¹. Such advance continues to this day as confirmed in glacier inventories covering the last two decades (Fig. S1). The largest advance of around 2 km was observed for Garibaldi Glacier (Fig. S1). The advancing behavior is all the more remarkable when compared to the extreme retreat of Marinelli Glacier ¹², the largest glacier of the CDI, which is located in close vicinity at the northern slope of Mount Shipton (Fig. 1c). From 1945 to 2005, it experienced an extreme recession of 12.2 km ¹³, explained by warming and fast retreat along over-deepened fjord bathymetry ¹².

Around 60 % of CDI glaciers are marine- or lake-terminating (MALT) ¹⁴. Thus, they do not only lose mass on their surface in contact with the atmosphere, but also at the ocean/lake interfaces via iceberg calving and subaqueous melting, collectively known as frontal ablation ¹⁵, which is controlled by ice dynamics and fracturing. Ice is also lost at the ocean/lake interfaces via iceberg calving and subaqueous melting, collectively known as frontal ablation ¹⁵. The ice losses and the contrasting behavior observed

for individual CDI glaciers can possibly be explained by both climatic and ice-dynamic changes. ~~However, the attribution of ice loss to climatic or ice-dynamic forcing is still missing to this day unknown. Most mass balance estimates are geodetic^{1,9,10} which includes both of these loss terms giving the total mass change. Attribution is possible for individual glaciers if one of the two ice-loss terms is computed. In a mass budgeting approach, the total mass change is quantified, along with one of the two primary loss terms controlled either by climate or ice-dynamics, leaving the other as a residual¹⁸. However, due to the harsh climatic conditions and the inaccessibility of this region, the CDI remains poorly studied^{13,14,16,17} and neither of these two loss terms cannot be quantified reliably. The climatic mass balance (CMB) in the Cordillera Darwin was conducted in the Cordon Martial (east of the study region, located in Argentina)¹⁹ and the Mount Sarmiento Massif^{20,21}, but these local efforts are insufficient and a CDI-wide estimate of the CMB is needed. The attribution of ice loss to climatic or ice-dynamic forcing is still missing to this day. It can~~ This would make attribution of the CDI possible and additionally provide give vital information on current trends and shifts in atmospheric conditions of the Southern Hemisphere's higher mid-latitudes. -

~~Attribution is possible for individual glaciers if one of the two ice-loss terms is computed. In a mass budgeting approach, the total mass change is quantified, along with one of the two primary loss terms controlled either by climate or ice-dynamics, leaving the other as a residual¹⁸. First attempts to study the climatic mass balance (CMB) in the Cordillera Darwin were conducted in the Cordon Martial (east of the study region, located in Argentina)¹⁹ and the Mount Sarmiento Massif^{20,21}. In both areas, the mass balance is negative and dominated by snowfall and surface melt, which is strongly linked to air temperature^{19,20}. Despite these local efforts to determine the mass balance of individual glaciers, a systematic approach for a CDI-wide estimate of the CMB is to this day missing. This lack of knowledge can be addressed with a well informed modelling product of CMB. Geodetic techniques in satellite remote sensing allow for operational inference of total glacier mass changes worldwide^{9,10,22}. Thus, with the climatic loss term available, the mass budget of the CDI glaciers can successfully be closed and give the dynamic loss as residual. Another common method~~ An alternative approach is the direct quantification of frontal losses with a flux gate approach^{15,23}. Here, the frontal ablation is estimated based on the ice flux through a gate upstream of the glacier front. Due to the lack of ice thickness observations in the CDI producing high uncertainties on reconstruction products (e.g., ref.^{2,3}), results will, however, remain are inaccurate. First estimates of frontal ablation in the CDI are limited to two glaciers: At Schiaparelli Glacier frontal ablation calculated with a mass budgeting approach²⁴ agrees well with inferred values from time-lapse camera observations²⁵. At Marinelli Glacier, Koppes et al.¹⁶ find a calving flux of around 0.4 Gt yr⁻¹ (2000-2005) applying an ice budget model.

The main aim of this study is to 1) quantify the unknown CMB of the CDI and 2) attribute ice loss to climatic or ice-dynamic forcing. We simulate the CMB over a 22-year period which provides first insights in climatic and glaciological trends and increase our process understanding of glacier response

Commented [NS1]: Move the preceding sentence to another section (discussion)?

~~to climate. The energy and mass balance will be examined using the physically-based CoCoupled Snowpack and Ice surface energy and mass balance model in PYthon (COSIPY)²⁶, which combines a surface energy and mass balance model with a subsurface multi-layer snow and ice model (Methods section). The fully distributed model allows an analysis of the spatial and temporal variability in the surface energy and mass fluxes across the CDI. Together with geodetic observations, we can use the CMB to close the mass budget of all glaciers in the CDI (objective 2) and provide a first estimate of their frontal losses. In this way, an unprecedented attribution of the observed mass loss to climatic and ice-dynamic forcing becomes possible. With a continuous 22-year CMB simulation, we will furthermore provide first insights in climatic and glaciological trends, and increase our process understanding of glacier response to climate. The energy and mass balance will be examined using the physically-based CoCoupled Snowpack and Ice surface energy and mass balance model in PYthon (COSIPY)²⁶, which combines a surface energy and mass balance model with a subsurface multi-layer snow and ice model (Methods section). The fully distributed model allows an analysis of the spatial and temporal variability in the surface energy and mass fluxes across the CDI. The unprecedented attribution of the observed mass loss to climatic and ice-dynamic forcing plus the fully distributed CMB model allow allowing conclusions to be drawn on the contrasting glacier behavior in the last two decades.~~

Results

Climatological characteristics of the CDI

High-resolution (200 m spatial and 3-hourly temporal) atmospheric forcing was created by observation-informed downscaling of ERA5 reanalysis data (Methods section). Our multi-method downscaling relies on quantile mapping²⁷ as well as modelling of solar radiation²⁸ and orographic precipitation²⁹⁻³¹ for the study period 04/1999-03/2022. Variables comprise air temperature, relative humidity, air pressure, wind velocities, cloud cover, incoming solar radiation and precipitation (Methods section). Evaluation of atmospheric variables shows overall good agreement with observations from automatic weather stations (Table S1, Table S2). A comparison of local precipitation with a firn core in the central Cordillera Darwin shows an overestimation while catchment-wide precipitation about 50 km downwind indicates an underestimation (Fig. 1, Fig. S2). The location of the former firn core is in an exposed saddle position where the local wind field and snowdrift are not resolved at the process level, while the latter catchment comparison suffers from neglecting water storage. As we aim at a first climatic mass balance estimate on regional scale, consistent with regional geodetic measurements, local deviations are acceptable. The CDI is divided into four subregions (Fig. 1b) to analyze spatial variability of climatic characteristics and energy and mass fluxes across the study region. Significance of trends over the study period are formulated following the IPCC guidance for communication of confidence (Table S3)³². These different significant levels are marked in italic font.

The study region is characterized by temperate maritime climate. Annual mean air temperatures close to sea level (2 m above ground) lie around 5.2 °C with moderate interannual variability (± 3 to 4 °C). Air temperatures show a positive trend (*virtually certain*) over the study period of +0.42 °C per decade with an intensification from west to east. Conditions are overall humid with annual average relative humidity showing a drying towards the east (Fig. S3b). Relative humidity is on average higher in winter than summer, with the amplitude increasing from around 7 % in the west to around 12 % in the east. Within the westerly wind belt, frontal systems move from the southern Pacific Ocean towards southern Patagonia causing high precipitation amounts due to orographic uplift. Highest amounts are reached in the westernmost edge of the Cordillera Darwin, going up to 4000 mm yr⁻¹ at Mount Sarmiento (Fig. 2a). As the air masses move eastwards over the cordillera, fallout of precipitation causes a drying effect towards the east of the CDI (Fig. 2a, Fig. S3e). Precipitation amounts peak in the summer months, related to the increased wind velocities. The seasonality is forced by a southward shift of the westerly wind belt during summer³³. Precipitation and wind also agree on an increasing trend (*likely*) over the study period (+70.8 mm yr⁻¹ per decade and +0.1 m s⁻¹ per decade, respectively). Wind velocities are high throughout the year especially in the westernmost edge of the CDI where annual averages go up to 5.5 m s⁻¹, and decrease towards the central and eastern part of the CDI (Fig. 2b, Fig. S3d). High amounts of annual precipitation are accompanied by an average cloud cover of over 84 %. Such extensive cloud cover strongly limits direct solar radiation (Fig. S3c, Fig. S8a). Regional radiation differences between north and south are mainly explained by the orientation to the sun and the aspect of the slopes.

Climatic energy and mass balance of the CDI

The climatic energy and mass balance is simulated with COSIPY²⁶, a fully-distributed surface energy and mass balance model coupled to a multi-layer subsurface snow and ice model (Methods section). For model evaluation, simulation results are compared to glacier-specific geodetic mass balances from elevation change observations (2000-2013) based on the results by Braun et al.⁹ for glaciers with no frontal losses (Methods section) (Fig. S4). Based on this performance, we infer a glacier-specific CMB uncertainty of ± 0.62 m w.e. yr⁻¹. The simulated icefield-wide average CMB of land-terminating glaciers (-0.23 m w.e. yr⁻¹) agrees well with the geodetic reference dataset of Braun et al.⁹ (-0.27 m w.e. yr⁻¹) (Fig. S5).

The CMB of the entire CDI (including marine-, lake- and land-terminating glaciers) is nearly balanced for the study period (2000-2022) with +0.02 m w.e. yr⁻¹. Temporal variability is high with annual values between -0.81 and +0.93 m w.e. yr⁻¹ (Fig. 3c). Across the CDI, glacier-specific values show a high spatial variability (Fig. 1c), ranging from -5.44 m w.e. yr⁻¹ to +2.42 m w.e. yr⁻¹. We relate this diversity to hypsometry and aspect of the respective glacier catchments. MALT glaciers show on average a more positive CMB than land-terminating glaciers. Positive CMBs dominate especially in the central high-elevated part of the CDI (e.g., Ruginor, Garibaldi, Guilcher oeste and este), whereas lower-elevated glaciers or glaciers with large outlet tongues show more pronounced mass loss (e.g., Schiaparelli,

Romanche, Alemania and many of the small, unnamed glaciers at the icefield margin) (Fig. 1c). The altitudinal gradient of the CMB is steep with strong mass losses on the glacier tongues at low elevation and high mass gain towards the mountain peaks (Fig. 2c). The largest range is present in the western part of the CDI, where the annual CMB reaches nearly $-10 \text{ m w.e. yr}^{-1}$ at the lowest point of Schiaparelli Glacier and nearly $+10 \text{ m w.e. yr}^{-1}$ at the highest peak (Fig. 2c), resulting in a gradient of $-0.92 \text{ m w.e. yr}^{-1}$ per 100 m.

Mass is mainly gained from snowfall (62 %) at the surface and from refreezing within the snowpack (37 %) (Fig. 3a). Deposition of water vapor at the glacier surface contributes around 1 % to the accumulation. Mass loss is dominated by melt at the surface (92 %) and subsurface (7 %), with sublimation contributing only around 1 % to the total ablation. The average annual CMB is positive in the southern and western part of the CDI ($+0.35$ and $+0.17 \text{ m w.e. yr}^{-1}$, respectively), while it is close to zero in the north and east ($+0.03$ and $-0.06 \text{ m w.e. yr}^{-1}$, respectively) (Fig. 3a). These differences mainly stem from the contributions of snowfall and surface melt. Snowfall amounts strongly reduce from $+1.51 \text{ m w.e. yr}^{-1}$ in the west over south and north to $+0.92 \text{ m w.e. yr}^{-1}$ in the east of the CDI (Fig. 3a). In the western part, the high accumulation is partly balanced by enhanced surface melt ($-1.95 \text{ m w.e. yr}^{-1}$). Over the remaining CDI we find very similar surface melt rates of $-1.50 \text{ m w.e. yr}^{-1}$. With the decreasing snowfall towards the east, also the resulting CMB decreases (Fig. 3a). The CMB shows a pronounced seasonality (Fig. S6). In austral summer (December, January, February), it is strongly negative with the highest losses in the western ($-0.72 \text{ m w.e. yr}^{-1}$) and the smallest losses in the southern ($-0.48 \text{ m w.e. yr}^{-1}$) part of the CDI (Fig. S6a). This spatial difference is primarily explained by spatial variability in the surface melt. In austral winter (June, July, August), the CMB is positive all over the CDI with reduced melting (Fig. S6c). Regional differences are mainly linked to the snowfall gradient.

To understand the variability in surface melt, the energy fluxes at the surface and subsurface are analyzed. On average, the largest energy input comes from the net shortwave radiation (51 %) and the sensible heat flux (32 %) (Fig. 3b). The glacier heat flux (energy generated from penetrating shortwave radiation and refreezing that is transported to the surface via heat conduction) brings on average 15 % of energy, and is mostly constrained to the higher reaches of the glaciers (Fig. S9b). The heat flux from rain is limited to the glacier tongues (Fig. S9c), contributing around 2 % in total. Energy loss at the surface is dominated by the net longwave radiation (88 %), followed by the latent heat flux (12 %), which is an energy source on the glacier tongues but a sink on the spatial average (Fig. S9a). The largest amount of energy available for melting is found in the western part of the CDI (Fig. 3b, Fig. S9d). The energy surplus mainly stems from the sensible heat flux that is more than twice as high (14.74 W m^{-2}) than compared to the northern and eastern part (around 6.30 W m^{-2}) of the CDI (Fig. 2d). Subsequently, the sensible heat flux is the primary energy source (47 %) in the west (Fig. 3b). The enhanced values there are related to the higher wind velocities prevailing over the south-western part of the CDI (Fig. 2b). The slightly reduced net shortwave radiation in the south-west is caused on the one hand by reduced

incoming radiation due to orientation and topographic shading, on the other hand by higher average surface albedo (around 0.83 in the south-west, around 0.81 in the north-east) related to the enhanced snowfall (Fig. S8). The latent heat flux is an energy sink for the south, north and east part of the CDI (Fig. 3b). Its importance is reduced in the west. Due to the moister conditions in the west (Fig. S3b), water vapor transport towards the glacier surface is more common generating deposition and condensation instead of sublimation and evaporation, which are favored in the other sub-regions (Fig. S7, Fig. S9a). In summer, the latent heat flux temporarily turns into an energy source in the west, which is not seen over the rest of the icefield (Fig. S6b). The energy balance shows a strong seasonality with highest melt energy in summer (Fig. S6b) and strongly reduced energy in winter (Fig. S6d). In winter, energy availability is limited due to the minimized solar energy together with reduced wind velocities and lower air temperatures.

Our results show that the annual CMB (Fig. 3c) is mainly controlled by air temperature (Fig. S3a) (Pearson's correlation $r = -0.71$) and by precipitation (Fig. S3e) dictating accumulation ($r = 0.72$). The air temperature strongly influences the energy available for melting ($r = 0.85$). Melt energy and CMB are also linked to the annual average incoming solar radiation ($r = 0.54$ and $r = -0.45$, respectively). The importance of wind speed over the CDI is highlighted by a significant correlation with the sensible heat flux ($r = 0.62$) as well as accumulation ($r = 0.52$).

Over the 22-year study period, we see an increasing trend (*extremely likely*) in the surface melt ($+0.18$ m w.e. yr^{-1} per decade) that translates into a *likely* decreasing trend for mass balance (-0.21 m w.e. yr^{-1} per decade) which is caused by the more negative annual CMB towards the end of the study period (Fig. 3c). The increasing surface melt is mainly caused by an *extremely likely* increasing trend in the sensible heat flux ($+0.71$ W m^{-2} per decade) and net longwave radiation ($+0.63$ W m^{-2} per decade), which we primarily relate to increasing air temperature (both variables) and wind velocity (sensible heat flux). Additionally, we observe an *extremely likely* increasing net shortwave radiation ($+0.81$ W m^{-2} per decade), which we explain with an albedo feedback (albedo decrease of -0.01 per decade, *extremely likely*). Altogether, an *extremely likely* increasing trend in available melt energy ($+1.93$ W m^{-2} per decade) is obtained. The trend in the CMB is more pronounced for lower elevation bins. Below 600 m elevation, the CMB shows an *extremely likely* decreasing trend (up to -0.47 m w.e. yr^{-1} per decade). Snowfall or accumulation trends are not significant and currently unable to compensate for the increased surface melt.

Estimation of frontal ablation in the CDI

Mass budgeting of the geodetic and climatic mass balance gives information on frontal ablation of the 39 marine- and lake-terminating (MALT) glaciers of the CDI (Methods section). This way, we are able to assess the direct ice flux into the fjords. For comparison, we also calculate frontal ablation following a flux gate approach relying on surface velocities and reconstructed ice thickness³ (Methods section).

However, these results inhibit large errors due to the uncertain modelled ice thickness data in the CDI in the absence of in-situ observations.

With the mass budgeting approach, the total frontal ablation of the CDI is estimated to $1.44 \pm 0.72 \text{ Gt yr}^{-1}$ in 2000 to 2013. About half of the total flux is channelized through Marinelli Glacier with $0.40 \pm 0.08 \text{ Gt yr}^{-1}$ and Grande Glacier with $0.30 \pm 0.08 \text{ Gt yr}^{-1}$ (Fig. 4). Another 20 % of the frontal ablation is explained by two other prominent marine-terminating glaciers: Darwin with $0.14 \pm 0.03 \text{ Gt yr}^{-1}$, and Rugidor with $0.11 \pm 0.03 \text{ Gt yr}^{-1}$. The flux-gate approach gives almost the same total frontal ablation with $1.49 \pm 0.54 \text{ Gt yr}^{-1}$ for the year 2013 and shows the overall same pattern (Fig. 4). Largest frontal losses are found for the same four glaciers, however, in this case being responsible for only around half of the total flux. For both approaches, the majority of MALT glaciers show only low frontal ablation (Fig. 4). For few glaciers, the flux-gate approach shows clearly elevated values - though with large uncertainties. In general, calving into fjords explains the main share of the frontal ablation (around 90 %) although these glaciers constitute only 75 % of the surface area of all MALT glaciers. Calving into lakes plays a secondary role.

The availability of geodetic datasets covering multiple observation periods (e.g., ref. ²²) allows an analysis of changes in frontal ablation rates over the study period (from 2000-2010 to 2010-2020). Bias-corrected estimates based on mass budgeting using these datasets and the CMB, reveal a total frontal ablation of $1.55 \pm 0.71 \text{ Gt yr}^{-1}$ (equaling $1.34 \pm 0.61 \text{ m w.e. yr}^{-1}$) for the first period, and a slightly reduced value of $1.42 \pm 0.75 \text{ Gt yr}^{-1}$ (equaling $1.25 \pm 0.66 \text{ m w.e. yr}^{-1}$) for the second period. Associated uncertainties do not allow any reliable statement to be made about changes over the study period. The derived values suggest, however, that the frontal ablation has changed less over the last 20 years ($-0.05 \text{ m w.e. yr}^{-1}$ per decade) than the CMB.

Disentangling climatical and dynamical control on observed glacier changes

With the CMB and the frontal ablation available, we are able to quantify the main contributors to the mass change in the CDI. This way, we can disentangle the glacier mass changes observed from remote sensing into climatical and dynamical forcing. For few glaciers, the mass budgeting suggests frontal ablation as the major contributor to mass loss. The frontal ablation exceeds twice the climatic ablation for glaciers Rugidor, Marinelli, Darwin, Italia and an unnamed glacier at the south-western margin of Parry fjord (CL112833070) (Fig. 1c, Table S4). For these glaciers, the mass loss is primarily dictated by ice dynamics instead of climate. For about 15% of the glaciers, the contribution of climatic and frontal ablation is rather even (e.g., Grande, Cuevas, Lovisato, Frances) (Table S4). However, for more than half of the MALT glaciers, the climatic losses at the surface and within the snowpack largely exceed the frontal ablation, meaning that these glaciers are dominated by atmospheric processes at the glacier surface rather than ice dynamics.

Glaciers that have been advancing over the study period (Garibaldi, Finlandia, Guilcher oeste and este) are in general characterized by a strongly positive CMB and by medium to low frontal ablation (Fig. 1c). These glaciers do not only advance, but also show thickening further upstream (thus, gaining mass), suggesting a primary climatic control. We ascribe these climatological favorable conditions to the glaciers' exposition, geometry and hypsometry. All four glaciers have their origin at the high-elevated central plateau of the CDI (above 2000 m) causing a steep topography, expressed in an over-average accumulation-area-ratio (between 0.72 and 0.88 compared to the CDI average of 0.61). Glaciers with high accumulation-area-ratio are often less sensitive to changes in equilibrium line altitude (ELA) - the elevation at which surface mass gain equals surface mass loss over one year - and, thus, warming due to the small and steep ablation area¹⁵. Garibaldi, Guilcher oeste and Guilcher este face southward and are, thus, topographically stronger shaded from solar radiation (Fig. S8). Altogether, these characteristics ultimately lead to high snowfall amounts together with reduced surface ablation (Fig. 2c, Fig. S9d), favoring mass gain.

Overall, mass loss in the CDI is primarily controlled by atmospheric conditions. The climatic ablation of the entire CDI on average amounts to $4.20 \pm 1.48 \text{ Gt yr}^{-1}$ (74 % of total ablation) while the frontal ablation on average amounts to $1.44 \pm 0.72 \text{ Gt yr}^{-1}$ (26 % of total ablation). Considering the MALT glaciers only, for almost half of the glaciers the mass loss is mainly controlled by ice dynamics. For six individual glaciers covering around 15 % of the CDI area, the ice-dynamical overtakes the climatical contribution to mass loss by a factor of two. While trends in CMB impose an increasing mass loss associated with climatic changes, frontal ablation remains without significant change over the study period.

Discussion

The climatic conditions during our study period (1999-2022) show strong zonal gradients across the Cordillera Darwin, specifically for precipitation, wind velocities and relative humidity (Fig. S3). These gradients were already reported for the 20th century climate¹¹. The highest precipitation amounts are located in the north-northwestern part of the CDI (Fig. 2a), as previously reported³⁴. With regard to atmospheric trends we confirm that wind velocities and precipitation are increasing over Tierra del Fuego (e.g., ref. ⁶). The warming trend for the beginning of the 21st century found in this study ($+0.42 \text{ }^\circ\text{C}$ per decade) exceeds warming rates reported for the 20th century ($+0.13$ to $+0.20 \text{ }^\circ\text{C}$ per decade)³⁵ as well as the projected warming until mid of this century under a high-emission scenario (RCP8.5) (total warming of $+0.5 \text{ }^\circ\text{C}$ until 2050)³⁶.

The moister, more rich in snow conditions in the south-western part of the CDI have been proposed to cause less thinning and retreat, while the opposite is true for the drier north-eastern part¹. Our results support the fact that climatic conditions force more negative mass balances in the north-eastern part of the CDI compared to the south-west (Fig. 3a). Regional differences are also reflected in equilibrium line

altitudes (ELAs) in the CDI. Simulated ELA values increase from the west (758 m) over the center (810 m) to the east (894 m). At Schiaparelli Glacier in the west, an average ELA of 730 ± 50 m (2000-2017) has been reported²¹ which is in very good agreement with our results (average ELA of 734 m). ELAs reported for Grande Glacier with around 640 ± 200 m¹ to 650 m (for 2011)¹⁷ based on single year end-of-summer snowline altitudes are distinctly lower than our long-term average of 790 m. For Marinelli Glacier, we calculate an average ELA around 802 m for the 22-year study period, which falls in between previous estimates ranging from around 600 m in the year 2000¹² to 1100 m in 2011¹⁷. Melkonian et al.¹ derive an ELA of 650 ± 200 m from single year end-of-summer snowlines at Garibaldi Glacier which is lower than the average ELA of 760 m we found in this study. Discrepancies between end-of-summer snowline and mean long-term ELA are explained by the possibility of snow fall events during the summer months³⁷, which do impose large uncertainties on ELA detected from satellite imagery.

Snowfall and surface melt are determined as the main contributors to the CMB at Schiaparelli Glacier (western CDI) and Martial Este (east of CDI), reflected in a strong correlation with air temperature and precipitation^{21,38}, which is in agreement with our results (Fig. 3a). Similar to our results for the western CDI, the energy input at Schiaparelli Glacier is dominated by incoming radiation and sensible heat flux, while the largest energy losses are attributed to the outgoing radiation and the energy consumed by melting^{21,20}. The importance of the sensible heat flux as an energy source has been highlighted in previous energy balance studies at the Southern Patagonian Icefield³⁹⁻⁴¹ and the Gran Campo Nevado⁴². The pronounced altitudinal mass balance gradient observed in the CDI (Fig. 2c) is a typical characteristic of Patagonian glaciers (e.g., ref. ^{39,41,43}), as is the high inter-annual variability³⁹. Periods of more negative/positive CMB (Fig. 3c) are in agreement with findings of geodetic mass balance from Dussailant et al.¹⁰ reporting stronger losses in the period 2000-2006 compared to the period 2012-2016 for the Fuegian Andes, although these values suffer from poor data coverage in that region (see supplementary material in ref. ¹⁰). Seasonality in the energy fluxes over the CDI (Fig. S6) is in general agreement with findings at Perito Moreno Glacier, located at the Southern Patagonian Icefield⁴¹: The energy input from the sensible heat flux exceeds the input from net shortwave radiation during austral winter, latent heat flux has a minimum value during austral spring, and the glacier (conductive) heat flux peaks during austral winter, when the glacier surface gets cooled by the atmosphere. Schaefer et al.⁴⁰ found latent heat flux as an energy source for two Patagonian glaciers during the ablation season, which confirms our findings for the western region of the CDI, where latent heat flux turns from energy sink to energy source during summer.

A trend analysis over the study period clearly demonstrates an *extremely likely* increase in surface melt. This development causes a *likely* trend towards more negative mass balance. Despite the *likely* positive trend in precipitation amounts, we do not find an increase in snowfall amounts. If these recent trends continue, the CDI will get further out of balance and glaciers will tread a path of accelerating mass loss as already indicated by the current trend in the CMB (Fig. 3c). In general, glaciers that are covering a

lower elevation range (< 600 m) already experience more pronounced mass losses with an average CMB trend of $-0.41 \text{ m w.e. yr}^{-1}$ per decade (*extremely likely*). At Perito Moreno Glacier, Minowa et al. ⁴¹ found a *virtually certain* decreasing trend in the surface mass balance over the ablation zone of $-0.90 \pm 0.3 \text{ m w.e. yr}^{-1}$ per decade (1996-2020), which is in general agreement with our findings for areas of similar elevation in the CDI ($-0.47 \text{ m w.e. yr}^{-1}$ per decade).

Frontal ablation is to this day unexplored in the CDI, except for Marinelli Glacier. Koppes et al. ¹⁶ found 0.40 Gt yr^{-1} for the beginning of the 21st century, which is in agreement with our estimates of $0.40 \pm 0.08 \text{ Gt yr}^{-1}$ by mass budgeting. Frontal ablation at the CDI is in a similar order of magnitude as for the Northern Patagonian Icefield ($2.5 \pm 0.5 \text{ Gt yr}^{-1}$ in 2000-2019), which covers almost twice the area ¹⁵. Alike the Northern Patagonian Icefield, ice-dynamic losses are also channelized through only a few prominent outlet glaciers (Fig. 4). Yet, the fraction of frontal ablation to total ablation (26 %) for the entire CDI is substantial. Considering the MALT glaciers only, this fraction (48 %) is similar to the MALT glaciers of the Southern Patagonian Icefield (48 %) ¹⁵. Apart from these few glaciers, however, mass loss is primarily controlled by atmospheric conditions. Frontal ablation calculated by two different methods in this study overall agrees well (Fig. 4). Discrepancies between the two approaches are within the error ranges for most glaciers. For a few glaciers (e.g., Finlandia, Garibaldi) the flux gate method produces higher frontal ablation, which can be explained by a possible overestimation of ice thickness and/or velocity, or underestimation of modelled CMB, as well as frontal advances that are neglected in the flux gate approach.

Major limitations of this study are the scarce observations, making model calibration and validation a challenge. Combining in-situ and remotely sensed observations, **we accomplished to satisfyingly constrain the CMB**. However, **we want to note that** most energy fluxes are not directly verifiably due to a lack of observations. Thus, absolute values should be handled **with a certain care**, while relative comparison and trends are considered more reliable. During model calibration and analysis, we found a strong sensitivity of modelled latent heat flux to relative humidity, a model input variable which is prone to uncertainties. Furthermore, **we want to point out that** the considerable refreezing rates predicted by the model are a surprising result for a supposedly **temperature** icefield in a maritime climate setting. Using a 3-hourly timestep, we assume that the sub-daily melt-refreeze cycle is resolved properly. Thus, refreezing rates are expected to be higher than in studies using a daily resolution⁴⁴. Veldhuijsen et al. ⁴⁵ found that reducing the temporal model resolution from hourly to daily causes a strong underestimation of refreezing (84 %). Due to the high amounts of rainfall and melt water, refreezing is expected to be important over parts of the CDI. Our results suggest that about 23 % of percolated water (rainfall plus melt water) refreezes within the snowpack with largest proportion in spring and autumn. These values lie in-between numbers for mid-latitude glaciers ($\sim 10 \%$)^{46,47} and the Antarctic (Peninsula) ($\sim 70\text{-}95 \%$)^{48,49}. To better quantify the refreezing in this region, we recommend the acquisition and investigation of firn cores in future studies. A minor limitation is related to the mass budgeting approach: Geodetic elevation change products cannot measure subaqueous ice losses when glaciers are retreating.

Field Code Changed

However, an assessment of these losses demonstrates that they are distinctly smaller compared to iceberg calving and lie within the reported uncertainty. For Marinelli Glacier, who experienced the strongest retreat over the study period, we estimate a maximum subaqueous ice loss of roughly 0.05 Gt yr^{-1} , constituting about 10 % of the total frontal ablation.

The CDI is one of the largest ice bodies in the Southern Hemisphere. We present the first simulation of the climatic energy and mass balance as well as frontal ablation. Our results allow an attribution of the observed mass loss to climatic or ice-dynamic forcing. Furthermore, these simulations shed light on the climatic imprint of glacier response at a unique location in the higher mid-latitudes of the Southern Hemisphere. Results reveal strong climatic gradients across the CDI that cause regional differences in the energy and mass fluxes. Overall, we show that the CDI has been climatically balanced in the recent two decades, but is entering a state of accelerated mass loss due to increasing surface melt. The melt increase is associated with an intense warming rate that exceeds projections for the early 21st century. The current melt trend is more pronounced at lower elevations. If recent warming trends continue, glaciers in the CDI will follow a trajectory of mass loss acceleration. Frontal ablation accounts for a significant share of the total ablation (26 %), but is only important for a minority of glaciers. These few glaciers dominate the CDI-wide frontal losses of $1.44 \pm 0.72 \text{ Gt yr}^{-1}$. We conclude that climatic warming has reached the CDI glaciers and that atmospheric conditions exert main control on the current glacier evolution in Tierra del Fuego.

Data & Methods

Data

Meteorological observations in the Cordillera Darwin are sparse (Figure 1c) due to the harsh conditions and the inaccessibility of the region. Details of all automatic weather stations (AWSs) used in this study are listed in Table S1. Operators are the Chilean Water Directorate (Dirección General de Aguas, short DGA) or individual researchers installing stations within the framework of different research projects in the Cordillera Darwin. Measured variables range from air temperature via relative humidity, global radiation, wind, air pressure to precipitation (Table S1). However, the station network is located close to sea level, lacking information at higher elevation. This deficit is especially problematic for precipitation, which is strongly influenced by orographic effects. At one AWS (Río Betbeder) a gauging station is installed, providing valuable information on precipitation amounts over the entire river catchment (Fig. 1c). All stations have been quality checked including a screening for outliers or drift in the data. In March 2020, the COrdillera Darwin Ice CorE Survey (CODICES) project drilled a 3.25 m firn core in a flat, northwest-southeast oriented 150 x 150 m saddle (54.6814°S, 69.6394°W, 2324 m a.s.l), one of the highest flat areas in the Cordillera Darwin. Seasonal variability in major ions and insoluble microparticles were used to estimate annual layers. The firn core record extends from March 2020 to austral spring 2016.

Measurements of surface ablation or mass balance are limited in the Cordillera Darwin. Ablation stakes have been installed between 2013 and 2020 at Schiaparelli Glacier located within the Mount Sarmiento Massif at the western edge of the CDI. Stakes are limited to the lowest part of the ablation area, delivering information about surface melt only²⁰. A 21-year long record of annual, winter and summer mass balance exists at the Martial Este Glacier located east of the main body of the CDI⁵⁰. The glacier is located outside of the direct study region but we extended the domain for model validation.

We use atmospheric variables from the ERA5 reanalysis product (the latest global product of the European Centre for Medium-Range Weather Forecasts, ECMWF) to generate the climatic forcing for the COSIPY model over the study site (54.25-55.00°S, 71.00-68.25°W). ERA5 provides high temporal (hourly) and spatial (31x31 km) resolution⁵¹. For Southern Patagonia, ERA5 and its previous versions have proven reliable in several modeling studies (e.g., ref. 20,21,31,52-54). Required variables comprise air temperature, relative humidity, air pressure, wind speed, cloud cover fraction and total precipitation over the study site. For downscaling of precipitation, we need upstream (53.75-55.50°S, 74.50-73.25°W) information about air temperature, relative humidity, wind vectors and geopotential height between 850 and 500 hPa.

Glacier outlines are taken from the DGA glacier inventory^{55,56} since comparison with the Randolph Glacier Inventory (V6) has revealed strong weaknesses of the latter for the southern Andes, especially for smaller glaciers⁵⁷. Glacier outlines available comprise two time stamps: 2000-2003 (in the following

denoted as 2000) and 2019. Since glacier catchments in the more recent inventory had been partially upgraded, catchments in the 2000 inventory had to be homogenized with the 2019 inventory for consistency. Furthermore, it was shown that neglecting the temporal evolution of the glacial extent by relying on constant glacier outlines can bias the comparison between surface and geodetic mass balance^{58,59}. Since elevation changes cover the period 2000 to 2013, we manually produced outlines for 2013, which have been missing so far. Thus, inventories available in this study cover years 2000, 2013 and 2019 oriented on inventory availability and geodetic data.

Geodetic mass changes of glaciers of the Cordillera Darwin are derived from interferometric Synthetic Aperture Radar (SAR) digital elevation models (DEMs) of the Shuttle Radar Topography Mission (SRTM) of the National Aeronautics and Space Administration (NASA) and the TerraSAR-X add-on for Digital Elevation Measurement satellite mission (TanDEM-X) of the German Aerospace Center (DLR). The SRTM C-band DEM has been acquired in February 2000 at a spatial resolution of 1 arcsec⁶⁰. We use the void-filled LP DAAC NASA SRTM DEM⁶¹. Bistatic SAR acquisitions of the TanDEM-X mission are available since 2011⁶². Here, we use Co-registered Single look Slant range Complex (CoSSC) data of the Southern Hemisphere ablation periods of the years 2012-2014. To further minimize elevation offsets due to differences in ice accumulation or time-varying depths of SAR signal penetration into the glacier volume, we use TanDEM-X acquisition dates which are close to the mean SRTM acquisition date (2000-02-16) whenever possible.

Ice flow velocities and reconstructed ice thickness had been taken from Millan et al.³. These datasets are used for calculation of frontal ablation based on a flux gate approach. Furthermore, surface velocities close to the glacier fronts are used for a classification of non-calving glaciers. A reliable classification is essential for a comparison between climatic and geodetic mass balance. Glaciers exceeding a velocity threshold of 65 m yr⁻¹ are classified as marine- or lake-terminating (MALT) glaciers, glaciers below the threshold velocity are classified as glaciers without significant frontal ablation and treated as land-terminating in this study.

Atmospheric forcing

Atmospheric input data required for the COSIPY simulation includes air temperature, relative humidity, incoming shortwave radiation, wind speed, air pressure, cloud cover and precipitation. To extend the climatic data beyond the respective measurement periods, we apply a downscaling scheme where we combine statistical downscaling with the application of a radiation model and a model of orographic precipitation.

Following previous studies in southern Patagonia (e.g., ref. ^{20,21,39}), we apply quantile mapping for statistical downscaling of air temperature, relative humidity and air pressure. Quantile mapping is a method of statistical bias correction, where the cumulative distribution function of the model is transferred to the cumulative distribution function of the observation^{27,63}. Statistically downscaled air

temperature and pressure are adjusted to sea level conditions, interpolated between the available station points via Ordinary Kriging ⁶⁴, and subsequently spatially extrapolated over the topography using a linear temperature lapse rate of $-0.6 \text{ K}/100 \text{ m}$ ²⁰ and the barometric equation, respectively. Relative humidity is likewise interpolated between the recording AWSs with Ordinary Kriging. Wind speed and cloud cover fraction are taken directly from ERA5 and interpolated to the model resolution.

A radiation model ⁶⁵ is applied over the study site to calculate global radiation over the glacier surface, following the methodology of Temme et al. ²⁰ at the Mount Sarmiento Massif. The model calculates both the direct and diffuse component of the solar radiation based on cloud cover, temperature, humidity and pressure. Corrections are applied for the slope and aspect of the respective grid cell. Shaded grid cells, either from the terrain or self-shaded, exclusively receive the diffuse solar radiation component ^{28,65}.

Due to the small-scale and episodic character of precipitation events, statistical techniques often fail to infer reliable distributions over complex terrain from coarse global data sets. Furthermore, strong winds limit the reliability of observations in southern Patagonia ^{42,66}. An orographic precipitation model showed improved performance as compared to extrapolation of observational data using altitudinal lapse rates ⁶⁷. The model calculates the orographic portion of precipitation resulting from forced orographic uplift over a mountain ³⁹. It is grounded on the linear steady-state theory of orographic precipitation, considering airflow dynamics, cloud timescales and processes of advection and downslope evaporation ^{29,30}. Since the model assumes stable and saturated conditions with unblocked air flow crossing the CDI from west to east ^{29,39}, time intervals that do not fulfill these constraints are excluded. Thresholds and parameter settings are taken from Temme et al. ²⁰. The total precipitation is calculated by adding the orographic precipitation calculated in the model to the large-scale precipitation, which is obtained by removing the orographic component from the ERA5 precipitation. Recent elevation- and bias-corrected precipitation products (W5E5, WFDE5) with lower spatial resolution and shorter temporal coverage indicate, an ERA5 overestimation over Tierra del Fuego ⁶⁸. Comparison of ERA5 daily precipitation with observations supports this finding ⁶⁹. To guarantee that the simulated total precipitation at the AWS locations agrees with the observed amounts, we constrain the large-scale precipitation from ERA5 to the annual measurements.

To derive snowfall from precipitation, a logistic transfer function is applied scaling around a threshold temperature of $1.0 \text{ }^\circ\text{C}$. A snow drift parametrization is included in the modelling framework to account for snow redistribution caused by the strong westerly winds over the CDI. Locations sheltered from or exposed to wind are identified by a topographic analysis and solid precipitation is redistributed accordingly ⁷⁰. Parameters and adjustments to the model are transferred from the Mount Sarmiento Massif ²⁰.

Climatic forcing data are validated on a daily basis with meteorological observations from AWSs that have not been used in the downscaling (Table S1) based on a statistical analysis of mean model bias,

Formatted: Spanish (Chile)

Field Code Changed

root mean square error and correlation. Overall, the performance of downscaled and modelled climate variables is satisfying (Table S2). The agreement of downscaled variables with measurements is improved compared to the raw ERA5 input, confirming the success of the downscaling approach. For further information on precipitation amounts, we compare annual precipitation over the river catchment of Río Betbeder with observed stream flow at the gauging station there, and snowfall with results of a firn core in the central CDI. The former river catchment is located in the northeast of the CDI covering a total area of 146 km² (Fig. 1c). The comparison indicates an underestimation (Fig. S2). However, considering the simplified approach (e.g., neglecting water storage and glaciers in the system), results are satisfying. The firn core site is located at an exposed saddle where we assume important wind erosion. The small-scale local wind field and snowdrift are, however, not fully resolved in our modeling approach, which explains an overestimation in the modelled snowfall.

COSIPY model

The open-source ‘COupled Snowpack and Ice surface energy and mass balance model in PYthon’ (COSIPY) ²⁶ is a physically based model grounding on the concept of energy and mass conservation. It couples a surface energy and mass balance model with a multi-layer subsurface snow and ice model, with the calculated surface meltwater serving as input to the subsurface model ²⁶. The energy balance model solves all energy fluxes F at the glacier surface:

$$F = SW_{in}(1 - \alpha) + LW_{in} + LW_{out} + Q_{sen} + Q_{lat} + Q_g + Q_{RRR}$$

where SW_{in} is the incoming shortwave radiation taken from the radiation model, α is the surface albedo, LW_{in} and LW_{out} are the incoming and outgoing longwave radiation, Q_{sen} and Q_{lat} are the turbulent sensible and latent heat flux, Q_g is the glacier heat flux and Q_{RRR} the rain heat flux. Melt can occur if the surface temperature is at the melting point (0.0 °C) and F is positive. Under this condition, the available energy for surface melt Q_M equals F . Rain and meltwater can percolate the snowpack and cause refreezing in the snow layers. Subsurface melting is possible by penetration of shortwave radiation in the upper snow layers. Solving the surface plus the internal mass balance in the snowpack, COSIPY gives the climatic mass balance (CMB) ⁷¹. The total ablation includes surface melting, sublimation and subsurface melting. Accumulation is the sum of snowfall, deposition and refreezing.

Albedo values are differentiated between snow, firn and ice surfaces. The decay of surface albedo due to snow aging is parameterized following the scheme of Oerlemans and Knap ⁷². The albedo depends on the time since the last snowfall and the snow depth. A bulk approach is applied to parameterize the turbulent heat fluxes. COSIPY offers the option to correct the flux-profile relationship by a stability correction using the Richardson-Number or the Monin-Obukhov similarity theory ²⁶.

We apply COSIPY version 1.4 in the period 04/1999-03/2022 with a 200 m spatial and a 3-hourly temporal resolution. All parameter settings follow the COSIPY set-up in the Mount Sarmiento Massif ²⁰ and are summarized in Table S5. The model performance of COSIPY was positively evaluated in the

Mount Sarmiento Massif, where four surface mass balance models of varying complexity were compared. COSIPY results agreed well with the other models as well as with observations of ablation stakes and geodetic mass balance ²⁰.

Geodetic mass balance processing

Elevation changes are calculated by DEM-differencing of SRTM and TanDEM-X. TanDEM-X DEMs are created based on differential interferometry following an established workflow ⁹. First, interferograms are computed from concatenated overlapping acquisitions, phase-unwrapped based on a minimum cost flow algorithm and converted to elevation values using the SRTM DEM as reference surface. Thereafter, the 'raw' TanDEM-X DEMs are iteratively co-registered to the SRTM DEM in the vertical and horizontal plane. Therefore, the 3D offset of each DEM is estimated based on all stable terrain with less than 25° surface slope excluding water and glacier areas. Finally, a regional elevation mosaic is created by merging all co-registered DEMs in the order of the relative deviation between the SRTM mean acquisition date (February 16th) and the tile-specific TanDEM-X date. The cell-specific TanDEM-X dates are stored with the DEM mosaic and subsequently used to calculate the respective elevation change rate during the SRTM and TanDEM-X DEM-differencing. The mean regional observation period of the elevation change rate measurement is 12.97 years.

To extract glacier-specific mass changes within the geodetic observation period, the elevation change map is masked to the glacier outlines of the 2000 inventory (see Data section). The mean elevation change rate is extracted for each glacier geometry and converted to volume and mass change based on the respective glacierized area and an approximate ice density of $900\pm 60 \text{ kg m}^{-3}$. Since glacier area changes during the observation period can bias the derived mass budgets ^{73,74}, the specific mass change rate of each glacier is calculated using the mean glacier area of the 2000 and 2013 inventories following the UNESCO definitions ⁷¹.

Model calibration and validation

Due to the limited in-situ observations in the Cordillera Darwin, calibration and validation of the CMB are a major challenge. With the large model domain and the high temporal and spatial resolution, resulting in a massive computational effort, intense model calibration is not feasible. Instead, the downscaling procedure and optimal parameter setting are grounded on the expertise gained at the Mount Sarmiento Massif, located at the western edge of the CDI ²⁰. Sensitivity runs are applied for further optimization. Temme et al. ²⁴ conclude that calibrating against regional satellite observations of mass change significantly improves the performance of CMB models. Following this approach, we rank the sensitivity runs based on the highest agreement with regional specific mass balance, as observed with satellite remote sensing, for all glaciers with no frontal ablation. The highest ranked run is presented in this study.

For model validation, we compare the climatic with the geodetic mass balance of each land-terminating glacier on a catchment level (catchment information not used during calibration). MALT glaciers are excluded because they also lose mass at the calving front due to ice dynamics. To reduce uncertainties, we limit the comparison to glaciers exceeding an area of 3 km². This gives a validation dataset of glaciers covering ~37% of the glaciated area of the CDI. Model performance is quantified by the root mean square error between the glacier-specific climatic and geodetic mass balance for the individual catchments. Stake measurements at Schiaparelli Glacier²⁰ and Martial Este Glacier⁷⁵ serve as additional validation of melt on the western and eastern edges of the CDI (Table S6).

Frontal ablation

In this study, we apply two different methods to determine frontal ablation for the entire CDI and the individual marine- and lake-terminating glaciers in the Cordillera Darwin. Firstly, we apply a mass budgeting, where the residual of the total glacier mass balance (ΔM_{tot}) from geodetic observations and the CMB (\dot{B}) simulated with COSIPY provides the frontal ablation (A_f): $A_f = \Delta M_{tot} - \dot{B}$ ¹⁸. Uncertainties of the glacier-specific CMB, constrained by model validation, directly translate into the uncertainties of frontal ablation estimations.

Secondly, we apply a flux gate approach^{15,23}. Here, frontal ablation is calculated based on the discharge (D) at a flux gate located upstream of the glacier front and the CMB downstream of the flux gate (\dot{B}_{FG}): $A_f = -D - \dot{B}_{FG}$. D is calculated by integrating the product of ice thickness and surface velocity perpendicular to the gate. For our study site, the lack of ice thickness measurements in the CDI makes ice thickness highly speculative. This directly translates into elevated uncertainties for frontal ablation.

Data availability

Average annual fields of the simulation results are available via this/is/a/dummy/link. Temporally higher resolved model data is available from the corresponding author on request.

References

1. Melkonian, A. K. *et al.* Satellite-derived volume loss rates and glacier speeds for the Cordillera Darwin Icefield, Chile. *The Cryosphere* **7**, 823–839 (2013).
2. Farinotti, D. *et al.* A consensus estimate for the ice thickness distribution of all glaciers on Earth. *Nature Geoscience* **12**, 168–173 (2019).
3. Millan, R., Mouginot, J., Rabatel, A. & Morlighem, M. Ice velocity and thickness of the world's glaciers. *Nature Geoscience* **15**, 124–129 (2022).
4. Izagirre, E. *et al.* The glacial geomorphology of the Cordillera Darwin Icefield, Tierra del Fuego, southernmost South America. *Journal of Maps* **20**, 2378000 (2024).
5. Garreaud, R. D., Vuille, M., Compagnucci, R. & Marengo, J. Present-day South American climate. *Palaeogeography, Palaeoclimatology, Palaeoecology* **281**, 180–195 (2009).
6. Garreaud, R., Lopez, P., Minvielle, M. & Rojas, M. Large-scale control on the Patagonian climate. *Journal of Climate* **26**, 215–230 (2013).
7. Goyal, R., Sen Gupta, A., Jucker, M. & England, M. H. Historical and Projected Changes in the Southern Hemisphere Surface Westerlies. *Geophysical Research Letters* **48**, e2020GL090849 (2021).
8. Garreaud, R. D., Clem, K. & Veloso, J. V. The South Pacific Pressure Trend Dipole and the Southern Blob. *Journal of Climate* **34**, 7661–7676 (2021).
9. Braun, M. H. *et al.* Constraining glacier elevation and mass changes in South America. *Nature Climate Change* **9**, 130–136 (2019).
10. Dussailant, I. *et al.* Two decades of glacier mass loss along the Andes. *Nature Geoscience* **12**, 802–808 (2019).
11. Holmlund, P. & Fuenzalida, H. Anomalous glacier responses to 20th century climatic changes in Darwin Cordillera, southern Chile. *J. Glaciol.* **41**, 465–473 (1995).
12. Porter, C. & Santana, A. Rapid 20th century retreat of Ventisquero Marinelli in the Cordillera Darwin Icefield. *Anales del Instituto de la Patagonia* **31**, 17–26 (2003).
13. Lopez, P. *et al.* A regional view of fluctuations in glacier length in southern South America. *Global and Planetary Change* **71**, 85–108 (2010).
14. Meier, W. *et al.* Late Holocene Glacial Fluctuations of Schiaparelli Glacier at Monte Sarmiento Massif, Tierra del Fuego (54°24'S). *Geosciences* **9**, 340 (2019).
15. Minowa, M., Schaefer, M., Sugiyama, S., Sakakibara, D. & Skvarca, P. Frontal ablation and mass loss of the Patagonian icefields. *Earth and Planetary Science Letters* **561**, 116811 (2021).
16. Koppes, M., Hallet, B. & Anderson, J. Synchronous acceleration of ice loss and glacial erosion, Glaciar Marinelli, Chilean Tierra del Fuego. *Journal of Glaciology* **55**, 207–220 (2009).
17. Bown, F., Rivera, A., Zenteno, P., Bravo, C. & Cawkwell, F. First Glacier Inventory and Recent Glacier Variation on Isla Grande de Tierra Del Fuego and Adjacent Islands in Southern Chile. in *Global Land Ice Measurements from Space* (eds. Kargel, J. S., Leonard, G. J., Bishop, M. P., Käab, A. & Raup, B. H.) 661–674 (Springer Berlin Heidelberg, Berlin, Heidelberg, 2014). doi:10.1007/978-3-540-79818-7_28.
18. Schaefer, M., Machguth, H., Falvey, M. & Casassa, G. Modeling past and future surface mass balance of the Northern Patagonia Icefield. *Journal of Geophysical Research: Earth Surface* **118**, 571–588 (2013).
19. Buttstädt, M., Möller, M., Iturraspe, R. & Schneider, C. Mass balance evolution of Martial Este Glacier, Tierra del Fuego (Argentina) for the period 1960–2009. *Advances in Geosciences* **22**, 117–124 (2009).
20. Temme, F. *et al.* Strategies for regional modeling of surface mass balance at the Monte Sarmiento Massif, Tierra del Fuego. *The Cryosphere* **17**, 2343–2365 (2023).
21. Weidemann, S. S. *et al.* Recent Climatic Mass Balance of the Schiaparelli Glacier at the Monte Sarmiento Massif and Reconstruction of Little Ice Age Climate by Simulating Steady-State Glacier Conditions. *Geosciences* **10**, 272 (2020).
22. Hugonnet, R. *et al.* Accelerated global glacier mass loss in the early twenty-first century. *Nature* **592**, 726–731 (2021).
23. Fürst, J. J. *et al.* The foundations of the Patagonian icefields. *Commun Earth Environ* **5**, 142 (2024).
24. Temme, F. *et al.* Strategies for Regional Modelling of Surface Mass Balance at the Monte

Formatted: Spanish (Chile)

Sarmiento Massif, Tierra Del Fuego.
<https://egusphere.copernicus.org/preprints/2022/egusphere-2022-1036/> (2022)
doi:10.5194/egusphere-2022-1036.

25. Langhamer, L. *et al.* Response of lacustrine glacier dynamics to atmospheric forcing in the Cordillera Darwin. *J. Glaciol.* 1–19 (2024) doi:10.1017/jog.2024.14.
26. Sauter, T., Arndt, A. & Schneider, C. COSIPY v1.3 – an open-source coupled snowpack and ice surface energy and mass balance model. *Geoscientific Model Development* **13**, 5645–5662 (2020).
27. Gudmundsson, L., Bremnes, J. B., Haugen, J. E. & Engen-Skaugen, T. Technical Note: Downscaling RCM precipitation to the station scale using statistical transformations – A comparison of methods. *Hydrology and Earth System Sciences* **16**, 3383–3390 (2012).
28. Mól, T., Cullen, N. J., Hardy, D. R., Winkler, M. & Kaser, G. Quantifying climate change in the tropical midtroposphere over East Africa from glacier shrinkage on Kilimanjaro. *Journal of Climate* **22**, 4162–4181 (2009).
29. Smith, R. B. & Barstad, I. A linear theory of orographic precipitation. *Journal of the Atmospheric Sciences* **61**, 1377–1391 (2004).
30. Barstad, I. & Smith, R. B. Evaluation of an orographic precipitation model. *Journal of Hydrometeorology* **6**, 85–99 (2005).
31. Sauter, T. Revisiting extreme precipitation amounts over southern South America and implications for the Patagonian Icefields. *Hydrology and Earth System Sciences* **24**, 2003–2016 (2020).
32. IPCC. *The Ocean and Cryosphere in a Changing Climate: Special Report of the Intergovernmental Panel on Climate Change.* (Cambridge University Press, 2022). doi:10.1017/9781009157964.
33. Garreaud, R. D. The Andes climate and weather. *Advances in Geosciences* **22**, 3–11 (2009).
34. Fernandez, R. A., Anderson, J. B., Wellner, J. S. & Hallet, B. Timescale dependence of glacial erosion rates: A case study of Marinelli Glacier, Cordillera Darwin, southern Patagonia. *J. Geophys. Res.* **116**, F01020 (2011).
35. Rosenblüth, B., Fuenzalida, H. A. & Aceituno, P. RECENT TEMPERATURE VARIATIONS IN SOUTHERN SOUTH AMERICA. *Int. J. Climatol.* **17**, 67–85 (1997).
36. Giesecke, R. *et al.* General Hydrography of the Beagle Channel, a Subantarctic Inter-oceanic Passage at the Southern Tip of South America. *Front. Mar. Sci.* **8**, 621822 (2021).
37. Bravo, C., Bozkurt, D., Ross, A. N. & Quincey, D. J. Projected increases in surface melt and ice loss for the Northern and Southern Patagonian Icefields. *Sci Rep* **11**, 16847 (2021).
38. Mutz, S. G. & Aschauer, J. Empirical glacier mass-balance models for South America. *J. Glaciol.* 1–15 (2022) doi:10.1017/jog.2022.6.
39. Weidemann, S. S. *et al.* Glacier Mass Changes of Lake-Terminating Grey and Tyndall Glaciers at the Southern Patagonia Icefield Derived From Geodetic Observations and Energy and Mass Balance Modeling. *Frontiers in Earth Science* **6**, 1–16 (2018).
40. Schaefer, M., Fonseca-Gallardo, D., Fariás-Barahona, D. & Casassa, G. Surface energy fluxes on Chilean glaciers: Measurements and models. *The Cryosphere* **14**, 2545–2565 (2020).
41. Minowa, M., Skvarca, P. & Fujita, K. Climate and Surface Mass Balance at Glaciar Perito Moreno, Southern Patagonia. *Journal of Climate* **36**, 625–641 (2023).
42. Schneider, C., Kilian, R. & Glaser, M. Energy balance in the ablation zone during the summer season at the Gran Campo Nevado Ice Cap in the Southern Andes. *Global and Planetary Change* **59**, 175–188 (2007).
43. Schaefer, M., MacHugth, H., Falvey, M., Casassa, G. & Rignot, E. Quantifying mass balance processes on the Southern Patagonia Icefield. *The Cryosphere* **9**, 25–35 (2015).
44. Arndt, A. & Schneider, C. Spatial pattern of glacier mass balance sensitivity to atmospheric forcing in High Mountain Asia. *J. Glaciol.* 1–18 (2023) doi:10.1017/jog.2023.46.
45. Veldhuijsen, S. B. M. *et al.* Spatial and temporal patterns of snowmelt refreezing in a Himalayan catchment. *J. Glaciol.* **68**, 369–389 (2022).
46. Krampe, D., Arndt, A. & Schneider, C. Energy and glacier mass balance of Fürkeleferner, Italy: past, present, and future. *Front. Earth Sci.* **10**, 814027 (2022).
47. Abraham, B. N., Cullen, N. J., Conway, J. P. & Sirguey, P. Applying a distributed mass-balance model to identify uncertainties in glaciological mass balance on Brewster Glacier, New Zealand.

- J. Glaciol.* **69**, 1030–1046 (2023).
48. Van Wessem, J. M. *et al.* Modelling the climate and surface mass balance of polar ice sheets using RACMO2 – Part 2: Antarctica (1979–2016). *The Cryosphere* **12**, 1479–1498 (2018).
 49. Hansen, N. *et al.* Downscaled surface mass balance in Antarctica: impacts of subsurface processes and large-scale atmospheric circulation. *The Cryosphere* **15**, 4315–4333 (2021).
 50. WGMS. FLUCTUATIONS OF GLACIERS DATABASE. 35 MB World Glacier Monitoring Service (WGMS) <https://doi.org/10.5904/WGMS-FOG-2023-09> (2023).
 51. Hersbach, H. *et al.* The ERA5 global reanalysis. *Quarterly Journal of the Royal Meteorological Society* **146**, 1999–2049 (2020).
 52. Lenaerts, J. T. M. *et al.* Extreme precipitation and climate gradients in Patagonia revealed by high-resolution regional atmospheric climate modeling. *Journal of Climate* **27**, 4607–4621 (2014).
 53. Bravo, C. *et al.* Assessing snow accumulation patterns and changes on the Patagonian Icefields. *Frontiers in Environmental Science* **7**, 1–18 (2019).
 54. Temme, F., Turton, J. V., Mölg, T. & Sauter, T. Flow regimes and Föhn types characterize the local climate of Southern Patagonia. *Atmosphere* **11**, (2020).
 55. Barcaza, G. *et al.* Glacier inventory and recent glacier variations in the Andes of Chile, South America. *Ann. Glaciol.* **58**, 166–180 (2017).
 56. DGA. Metodología de inventario público de glaciares, SDT No. 447, Ministerio de Obras Públicas, Dirección General de Aguas Unidad de Glaciología y Nieves, realizado por: Casassa, G., Espinoza, A., Segovia, A., and Huenante, J. (2022).
 57. Zalazar, L. *et al.* Spatial distribution and characteristics of Andean ice masses in Argentina: results from the first National Glacier Inventory. *J. Glaciol.* **66**, 938–949 (2020).
 58. Elsberg, D. H., Harrison, W. D., Echelmeyer, K. A. & Krimmel, R. M. Quantifying the effects of climate and surface change on glacier mass balance. *J. Glaciol.* **47**, 649–658 (2001).
 59. Mukherjee, K. *et al.* Evaluation of surface mass-balance records using geodetic data and physically-based modelling, Place and Peyto glaciers, western Canada. *J. Glaciol.* **69**, 665–682 (2023).
 60. Farr, T. G. *et al.* The Shuttle Radar Topography Mission. *Reviews of Geophysics* **45**, 2005RG000183 (2007).
 61. Earth Resources Observation And Science (EROS) Center. Shuttle Radar Topography Mission (SRTM) 1 Arc-Second Global. U.S. Geological Survey <https://doi.org/10.5066/F7PR7TFT> (2017).
 62. Zink, M. *et al.* TanDEM-X mission status: The complete new topography of the Earth. in *2016 IEEE International Geoscience and Remote Sensing Symposium (IGARSS)* 317–320 (IEEE, Beijing, 2016). doi:10.1109/IGARSS.2016.7729075.
 63. Cannon, A. J., Sobie, S. R. & Murdock, T. Q. Bias correction of GCM precipitation by quantile mapping: How well do methods preserve changes in quantiles and extremes? *Journal of Climate* **28**, 6938–6959 (2015).
 64. Murphy, B., Yurchak, R. & Müller, S. GeoStat-Framework/PyKrige: v1.7.0. Zenodo <https://doi.org/10.5281/ZENODO.7008206> (2022).
 65. Mölg, T., Cullen, N. J. & Kaser, G. Solar radiation, cloudiness and longwave radiation over low-latitude glaciers: Implications for mass-balance modelling. *Journal of Glaciology* **55**, 292–302 (2009).
 66. Schneider, C. *et al.* Weather Observations Across the Southern Andes at 53°S. *Physical Geography* **24**, 97–119 (2003).
 67. Jarosch, A. H., Anslow, F. S. & Clarke, G. K. C. High-resolution precipitation and temperature downscaling for glacier models. *Climate Dynamics* **38**, 391–409 (2012).
 68. Cucchi, M. *et al.* WFDE5: bias-adjusted ERA5 reanalysis data for impact studies. *Earth Syst. Sci. Data* **12**, 2097–2120 (2020).
 69. Lavers, D. A., Simmons, A., Vamborg, F. & Rodwell, M. J. An evaluation of ERA5 precipitation for climate monitoring. *Quart J Royal Meteor Soc* **148**, 3152–3165 (2022).
 70. Warscher, M. *et al.* Performance of complex snow cover descriptions in a distributed hydrological model system: A case study for the high Alpine terrain of the Berchtesgaden Alps. *Water Resources Research* **49**, 2619–2637 (2013).
 71. Cogley, J. C. *et al.* Glossary of Glacier Mass Balance and Related Terms. *IACS Contribution*

Formatted: Spanish (Chile)

No. 2 (2011).

72. Oerlemans, J. & Knap, W. H. A 1 year record of global radiation and albedo in the ablation zone of Morteratschgletscher, Switzerland. *Journal of Glaciology* **44**, 231–238 (1998).
73. Florentine, C., Sass, L., McNeil, C., Baker, E. & O’Neel, S. How to handle glacier area change in geodetic mass balance. *J. Glaciol.* 1–7 (2023) doi:10.1017/jog.2023.86.
74. Sommer, C. *et al.* Rapid glacier retreat and downwasting throughout the European Alps in the early 21st century. *Nat Commun* **11**, 3209 (2020).
75. Strelin, J. & Iturraspe, R. Recent evolution and mass balance of Cordón Martial glaciers, Cordillera Fuegoina Oriental. *Global and Planetary Change* **59**, 17–26 (2007).

Acknowledgements

FT was funded by the German Research Foundation (DFG) within the MAGIC project (FU 1032/5-1) and the RESPONSE project (TA 1719/2-1). JJF has received funding from the European Union's Horizon 2020 research and innovation programme via the European Research Council (ERC) as a Starting Grant (StG) under grant agreement No 948290. JA has received funding by the CNPq project (308831/2022-5) and the FAPERGS project (21/2551-0002034-2). EI was funded by the University of the Basque Country (UPV/EHU) under Grant PIF17/182 and by the Basque Government under the Consolidated Research Group IT1029-16 and IT1678-22. The authors are grateful for the scientific support and resources provided by the Erlangen National High Performance Computing (HPC) Center (NHR@FAU) of the Friedrich-Alexander-Universität ErlangenNürnberg (FAU). NHR funding is provided by federal and Bavarian state authorities. NHR@FAU hardware is partially funded by the DFG – 440719683. TanDEM-X data were kindly provided free of charge by the German Aerospace Center (DLR) under AO mabra_XTI_GLAC0264. The authors want to thank the Chilean National Forest Corporation (CONAF) for enabling and supporting the field work in the Cordillera Darwin, Parque Nacional Alberto de Agostini.

Author contribution

The concept of this study was developed by JJF, FT and MS. FT implemented the simulations with the support of JJF. In situ observational data were collected and provided by EI, RG, RJ, JAN, IG and DT. Elevation change rates and geodetic mass balances were processed by CS. FT led the writing process with the support of all the authors.

Competing interests

The contact author has declared that none of the authors has any competing interests.

Figure 1: Climatic mass balance of the Cordillera Darwin Icefield (CDI). *a* Overview panel of the study site. *b* Subregions of the CDI defined within this study. *c* Specific climatic mass balance (color scheme), termination type of the glaciers (outline style, outlines mark 2000 extent), observations from automatic weather stations and ablation stakes in the region, and specific mass fluxes from accumulation, ablation and calving (2000-2013) for selected glaciers. Triangles mark Mount Shipton (2568 m) and Mount Sarmiento (2207 m). The river catchment of Río Betbeder (eastern edge) is shown in yellow.

Figure 2: Climatological and mass and energy balance characteristics. Panels show mean annual **a** precipitation, **b** wind speed, **c** climatic mass balance and **d** sensible heat flux over the CDI (2000-2022). Black outlines display the glacier extent in 2000 (Barcaza et al. 2017).

Figure 3: Climatic energy and mass fluxes together with the resulting mass balance. CDI-wide (land-, lake- and marine-terminating glaciers) average annual climatic **a** mass and **b** energy balance components for the four subdomains: Snowfall (SNOW), deposition (DEPO), refreezing (REFR), surface melt (surfM), subsurface melt (subM), sublimation (SUBL), net shortwave radiation (SWnet), sensible (H), latent (LE) and glacier heat flux (B), heat flux from rain (QRR) and net longwave radiation (LWnet). The black diamonds give the resulting **a** climatic mass balance and **b** energy available for melting, respectively. The lower panel **c** displays the CDI-wide annual average climatic mass balance with shading indicating positive (blue) and negative (red) years. Dashed lines give 5-year averages.

Figure 4: Frontal ablation. Mean annual frontal ablation for marine- and lake-terminating glaciers, calculated with a mass budgeting approach (2000-2013) (dark green) and a flux gate approach (2013) (light green). The respective uncertainty is given in grey caps.